# Longitudinal molecular profiling elucidates immunometabolism dynamics in breast cancer

Kang Wang [1], Ioannis Zerdes [1,2], Henrik J. Johansson [3], Dhifaf Sarhan [4], Yizhe Sun[4], Dimitris C. Kanellis [5], Emmanouil G. Sifakis [1], Artur Mezheyeuski[6,7], Xingrong Liu[1], Niklas Loman[8,9], Ingrid Hedenfalk [9], Jonas Bergh [1,10], Jiri Bartek [5,11], Thomas Hatschek[1,10], Janne Lehtiö [3,12], Alexios Matikas [1,10] & Theodoros Foukakis [1,10] ✉

Although metabolic reprogramming within tumor cells and tumor micro-environment (TME) is well described in breast cancer, little is known about how the interplay of immune state and cancer metabolism evolves during treatment. Here, we characterize the immunometabolic profiles of tumor tissue samples longitudinally collected from individuals with breast cancer before, during and after neoadjuvant chemotherapy (NAC) using proteomics, genomics and histopathology. We show that the pre-, on-treatment and dynamic changes of the immune state, tumor metabolic proteins and tumor cell gene expression profiling-based metabolic phenotype are associated with treatment response. Single-cell/nucleus RNA sequencing revealed distinct tumor and immune cell states in metabolism between cold and hot tumors. Potential drivers of NAC based on above analyses were validated in vitro. In summary, the study shows that the interaction of tumor-intrinsic metabolic states and TME is associated with treatment outcome, supporting the concept of targeting tumor metabolism for immunoregulation.

Breast cancer is a highly heterogeneous disease and represents an ecosystem of extrinsic factors from the tumor microenvironment (TME) and intrinsic parameters from the cancer cells[1]. Clinical trials in the neoadjuvant setting are accelerating the evaluation process of novel breast cancer drugs[2–5], using pathologic complete response (pCR) as the primary endpoint, which is strongly associated with long-term outcome[6]. We have previously shown that immune state profiles

measured in sequential samples from breast cancer patients receiving neoadjuvant chemotherapy (NAC), especially from on-treatment samples, could better predict therapeutic response and long-term survival[7], but dynamic functional interplay within the intricate eco-system is not clear. As our understanding of various strategies employed by cancer cells to evade immune surveillance deepens, several key steps of immune state modulation are increasingly

[1]Department of Oncology-Pathology, Karolinska Institutet, Stockholm, Sweden. [2]Theme Cancer, Karolinska University Hospital and Karolinska Comprehensive Cancer Center, Stockholm, Sweden. [3]Department of Oncology-Pathology, Karolinska Institutet, and Science for Life Laboratory, Stockholm, Sweden. [4]Department of Laboratory Medicine, Division of Pathology, Karolinska Institutet, Stockholm, Sweden. [5]Department of Medical Biochemistry and Biophysics, Karolinska Institutet, Stockholm, Sweden. [6]Department of Immunology, Genetics and Pathology, Uppsala University, Rudbeck Laboratory, Uppsala, Sweden. [7]Molecular Oncology Group, Vall d'Hebron Institute of Oncology (VHIO), Barcelona, Spain. [8]Department of Hematology, Oncology and Radiation Physics, Lund University Hospital, Lund, Sweden. [9]Division of Oncology, Department of Clinical Sciences, Lund University, Lund, Sweden. [10]Breast Center, Theme Cancer, Karolinska University Hospital and Karolinska Comprehensive Cancer Center, Stockholm, Sweden. [11]Danish Cancer Institute, DK-2100 Copenhagen, Denmark. [12]Division of Pathology, Karolinska University Hospital and Karolinska Comprehensive Cancer Center, Stockholm, Sweden. ✉e-mail: theodoros.foukakis@ki.se

becoming targets for boosting the antitumor immunity, such as innate immune-sensing machinery, cellular metabolism, genetic alterations of oncogenic signaling, and epigenetic regulators[8].

Metabolic reprogramming allows tumors to acquire metabolic properties that support cell survival, evasion of immune surveillance, and hyperplastic growth[9], which not only meets specific demands for increased energy, biomass, redox maintenance, and cellular communication, but also interacts with the complex TME[10–12]. Previous studies unraveled the plasticity of cancer metabolism in vitro and identified alterations in several metabolic enzymes, fluxes and mediators[13–16]. Moreover, immunometabolic interplay exists in the TME, where complex metabolic networks of tumor, stromal and immune cells are dictated by cell-intrinsic and environmental factors[17]. This diverse milieu of immune, tumor and stromal cells creates a complex and dynamic ecosystem that can be influenced by cancer type and treatment[18,19]. There is emerging interest in the cancer immunometabolic phenotype, which can have clinical implications[17,20], as a source of prognostic biomarkers[21–23] and for therapeutic targeting[24–26]. However, the immunometabolic remodeling that occurs during treatment is not well understood. The neoadjuvant setting provides a unique window to assess predictive biomarkers and response to therapy in vivo[27]. Using single-nucleus DNA and RNA sequencing in longitudinal tumor samples, we previously demonstrated that breast cancer-resistant genotypes are pre-existing and adaptively selected by NAC[28]. In addition, modern mass spectrometry (MS) enables us to measure the actual druggable molecular phenotype, paving the way for precision cancer medicine[29]. A series of translational studies including treatment-naive triple-negative breast cancer (TNBC) samples have indicated metabolite-immune crosstalk, where metabolites are expected to represent a potential therapeutic strategy to promote the efficacy of immunotherapy[24–26,30]. Therefore, proteogenomic analysis of neoadjuvant trials enables us to better understand the metabolic communication between tumor cells and TME with a dynamic perspective, which holds great therapeutic value on shaping the TME for effective antitumor immunity.

In this study, we comprehensively characterize the genomic and proteomic landscape of HER2-negative breast cancer longitudinally, using mass spectrometry-based proteomics, bulk RNA microarray, single nucleus RNA sequencing (snRNA-seq), blood-tumor paired whole-exome sequencing (WES), histopathology, and in vitro validation. The main objectives were to (i) evaluate the interaction of immune and metabolic molecular phenotypes and prognostic relevance, (ii) quantify how immune-metabolism interplay evolves during NAC, (iii) reveal immunometabolic biomarkers and cellular phenotypes correlated to NAC response, (iv) delineate metabolic heterogeneity by immune state in single-cell resolution, and (v) explore the effect of clonal evolution on shaping TME switch.

## Results

### Longitudinal proteogenomic breast cancer cohort

The PROMIX trial is described in detail in Methods. After receiving NAC, 13.4% of patients attained pCR, including 10 (26.3%) triple-negative and 10 (9%) luminal tumors. After a median follow-up of 80 months, 56 (37.3%) first events had occurred. There were 55 (36.7%) recurrences and 46 (30.7%) all-causes deaths. Detailed clinical and pathologic patient characteristics are provided in Supplementary Table 1 and 2. Longitudinal tissue biopsies (pretreatment and after two cycles) and surgical specimens were collected in the PROMIX trial (Fig. 1a), thus a multi-omics cohort of 150 HER2-negative breast cancer patients was established (Fig. 1b): bulk microarray gene expression profiling (GEP) (122 patients, 275 samples), single nucleus RNA-seq (snRNA-seq) (8 patients, 20 samples), whole-exome sequencing (WES) (20 patients, 50 tumor samples), multiplex fluorescent (mf) immuno-histochemistry (IHC) (6 patients, 16 tumor samples), and MS-based proteomics (29 patients, 53 samples). The intersection of the

proteomics data with GEP and WES is shown in Fig. 1c. Herein, multi-omics data enabled comprehensive analyses on the correlation of immunometabolic phenotype and treatment response/long-term survival (GEP), the interaction of immune state and tumor metabolism (GEP), potential metabolic targets that modulated TME (paired GEP and proteomics), metabolic characteristics of breast epithelial, immune and stromal cells per immune state (snRNA-seq), tumor and immune state co-evolution under NAC (WES and GEP).

### Immune landscape and its prognostic significance

To better delineate TME evolution during NAC, we analyzed GEP data across the three time points (pre/on/post-NAC) and validated our findings both at the protein level using mf IHC and MS-based proteomics, as well as in an external cohort. Using GEP, unsupervised immune state clusters were modeled by integrating comprehensive immune signatures representing seven immune components with quanTIseq-based immune cell fractions, which classified all samples into three distinct immune states: cold ($n = 100$), warm ($n = 118$), and hot ($n = 57$) (Fig. 2a and Supplementary Fig. 1a). Overall, hot tumors with the highest immune score calculated by ESTIMATE (Supplementary Fig. 1b) also had upregulated global immune-related signature scores, and vice versa for cold tumors (Fig. 2a), whereas KI67 mRNA was comparable across the three groups (Supplementary Fig. 1c). A series of clinicopathologic characteristics and GEP-based biomarkers were compared between the three groups to validate the generated computational immune states internally (shown as Supplementary Table. 3). In general, TNBC (IHC-based)/basal-like (PAM50 intrinsic subtype) tumors were more often labeled as hot (both 38.6%), and luminal A/B tumors were immunologically cold (71%) ($P < 0.001$, see Supplementary Fig. 1d and Supplementary Table. 3). Interestingly, when the samples were annotated with Thorsson's PanCancer immune subtypes[31], >80% of the hot tumors were classified as IFN−γ dominant while TGF−ß dominant subtype that exerts systemic immune suppression and inhibits host immunosurveillance[32] only appeared in cold tumors (Fig. 2a and Supplementary Table. 3). Hot tumors were more infiltrated with higher intermediate-(10–50%) and high-density (>50%) TIL infiltration (65.6%) than warm (49.4%) or cold (31.3%) tumors ($P = 0.006$). Neither timepoint ($P = 0.5$) nor tumor cellularity ($P = 0.69$) distributed differently between the three immune states (Fig. 2b). Moreover, quanTIseq-based immune cell composition indicated hot tumors had the highest proportion of activated cells such as CD8 + T cells, B cells, and M1 macrophages. In contrast, suppressor cells like M2 Macrophages were elevated in cold tumors (Supplementary Table. 3). In addition, we employed multiplex fluorescent immunohistochemistry (mfIHC) of whole-section slides to support above immune cell deconvolution results, and cell classes were assigned using the binarized marker (co)expression patterns as illustrated in Supplementary Fig. 1e. mfIHC indicated that hot tumors have a higher density of B cells ($P = 0.03$), CD8 + T cells ($P = 0.06$) but a lower density of macrophage M2 cells ($P = 0.03$) than cold tumors (Fig. 2c and Supplementary Data. 1). To investigate protein level differences between the immune state subtypes, we conducted differential protein abundance analysis using mass spectrometry (MS) based proteomics data (Fig. 2d). Protein levels of biomarkers of immune activation (such as GZMK, CD8A, HLA-A, CD48) were upregulated within hot/warm tumors ($n = 30$) compared with cold tumors ($n = 23$) (Fig. 2d).

Next, we aimed to identify association of tumor immune states with clinical outcome (Fig. 2e and Supplementary Table. 4). Hot tumors were more likely to reach pCR compared with cold tumors, both pre-treatment (multivariable adjusted OR = 1.24; 95% CI, 1.04–1.48, $P = 0.02$) and on-treatment (multivariable adjusted OR = 1.39; 95% CI, 1.11–1.73, $P = 0.005$) after adjusting for breast cancer subtype, tumor size and lymph node status. Similar associations (hot vs cold tumors) were observed within the group of luminal tumors (pre-treatment: multivariable OR = 1.22; 95% CI, 1.03–1.46, $P = 0.03$;

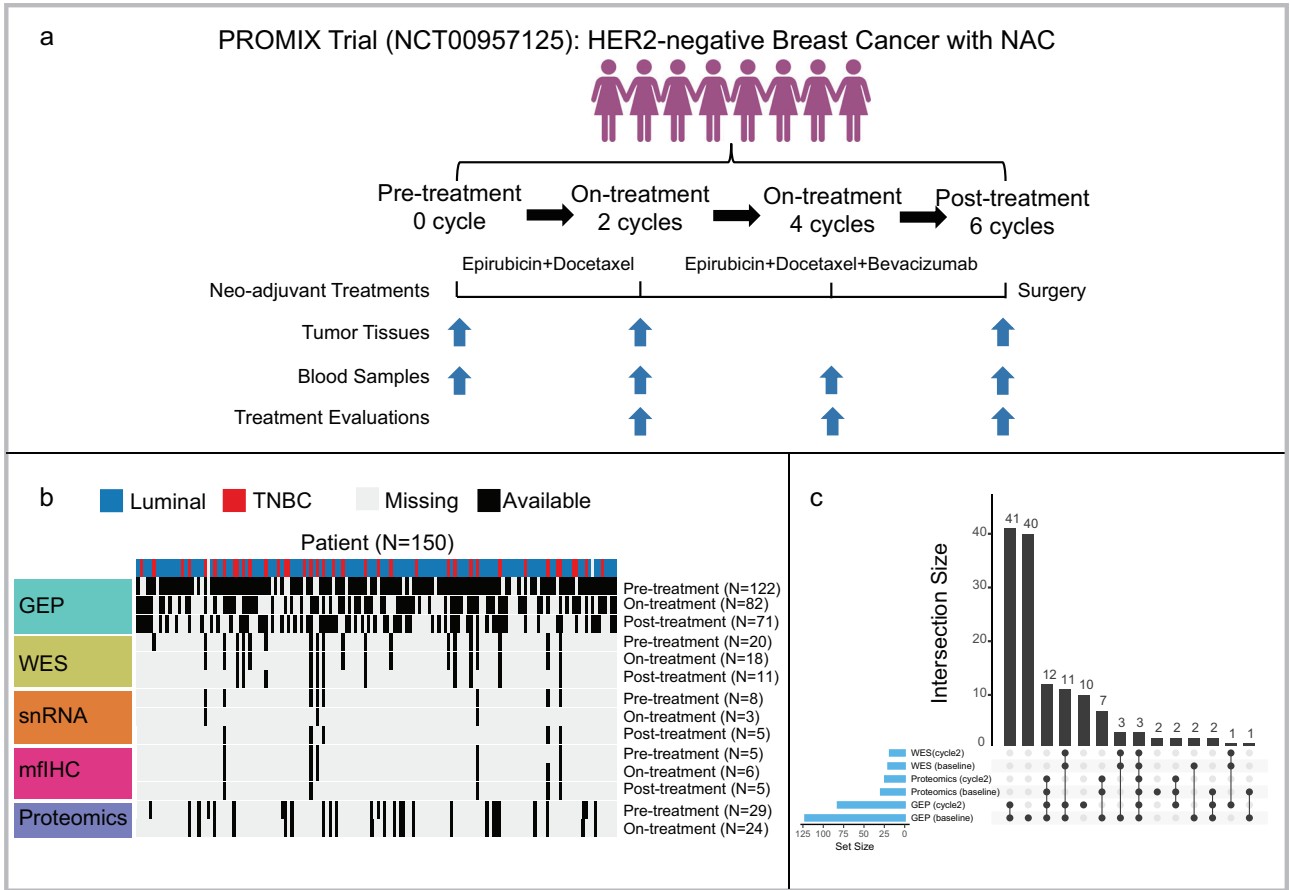

**Fig. 1 | Overview of the study design and multi-omics data collection. a** Pre/on/ post-treatment breast cancer tissue were collected from a phase II PROMIX trial (ClinicalTrials.gov identifier NCT00957125). Enrolled patients with locally advanced (tumor size > 20 mm) HER2-negative breast cancer, who were scheduled to receive six cycles of NAC with a combination of epirubicin and docetaxel. Bevacizumab was added during cycles 3–6 for those patients who did not achieve a clinical complete response (cCR) after the second cycle of NAC. For details on clinicopathologic characteristics, see Supplementary Table1. Fig1.a created using Biorender (https:// biorender.com/). **b** Heatmap showing longitudinally in-depth MS-based proteomics, transcriptomics, genetic and mfIHC/IF profiling from PROMIX trial. **c** UpSet plot showing multi-omics data intersection. HER2 Human epidermal growth factor receptor-2, mfIHC multiplex fluorescent immunohistochemistry; MS mass spectrometry.

on-treatment: multivariable OR = 1.64, 95% CI, 1.33–2.02, $P < 0.001$), while the limited number of TNBC cases precluded any conclusions in this subgroup. Regarding the association between immune state and long-term DFS, we observed that pre-treatment warm tumors tended to have inferior DFS than cold tumors (pre-treatment: multivariable HR = 2.18; 95% CI, 0.92–5.15, $P = 0.08$).

Furthermore, a machine learning based algorithm using GSVA score of TCGA PanCancerAtlas (https://cri-iatlas.org/) immune gene set[31,33], was applied to screen prognostic role on DFS by immune state. Natural killer (NK) cell and its subpopulation like CD56[bright] NK cells were identified as a promising biomarker across different immune states (Supplementary Fig. 2a). Kaplan-Meier survival analysis stratified by treatment timepoint verified that patients with higher signature score of NK cells exhibited longer DFS, especially on post-treatment samples (Supplementary Fig. 2b, c). Specifically, we observed that difference in CD56[bright] NK cells signature score between disease-free patients and those with relapse became more and more pronounced during NAC (Supplementary Fig. 2d), but no difference was found between pCR and non-pCR group (Supplementary Fig. 2e). Gene set enrichment analysis (GSEA) suggested that a series of metabolic pathways (i.e. glycolysis, fatty acid, oxidative phosphorylation (OXPHOS)) were enriched in patients with low NK cells signature score (Supplementary Fig. 2f). Furthermore, representative post-treatment slides were stained by NK cell mIF panel, which included NK cells (*CD3-CD56 +*), NKT cells (*CD3 + CD56 +*), adaptive NK cells

(*CD3-CD56 + CD57 + NKG2C + FcɛRγ-*) and conventional NK cells (*CD3-CD56 + CD57 + FcɛRγ +*) (Supplementary Fig. 2g). As illustrative examples, patients 617 (pCR) and 213 (non-pCR) from PROMIX who had high abundance of NK and adaptive NK cells remained event-free at 10 years postoperatively. Conversely, patient 314 (pCR) had few NK cells and recurred within 1 year after the operation (Supplementary Fig. 2g).

For external validation, we repeated our analysis pipeline on the Korean validation cohort (HER2-negative subset, 86 patients, 144 tumors) and corroborated the above findings. Tumors with GEP were classified into cold ($n = 46$), warm ($n = 62$) and hot ($n = 36$) immune state (Supplementary Fig. 3a). TIL density gradually increased from cold to hot tumors (Supplementary Fig. 3b), and triple negative tumors were significantly enriched in the hot group (Supplementary Fig. 3c). Compared with immunologically cold tumors, hot tumors were more likely to reach pCR (Supplementary Fig. 3d), (pre-treatment: multivariable OR = 1.56; 95% CI, 1.10 to 2.2, $P = 0.02$; on-treatment: multivariable OR = 1.29; 95% CI, 0.92 to 1.83, $P = 0.15$); and similar correlation was seen in TNBC subsets (pre-treatment: multivariable OR = 1.64; 95% CI, 0.99 to 2.72, $P = 0.06$; on-treatment: multivariable OR = 1.51; 95% CI, 0.83 to 2.74, $P = 0.19$).

Taken together, our genomic and proteomic analyses revealed and validated distinct immune clusters. Immunologically hot tumors associated with better treatment response, whereas post-NAC NK cell abundance correlated with improved long-term survival.

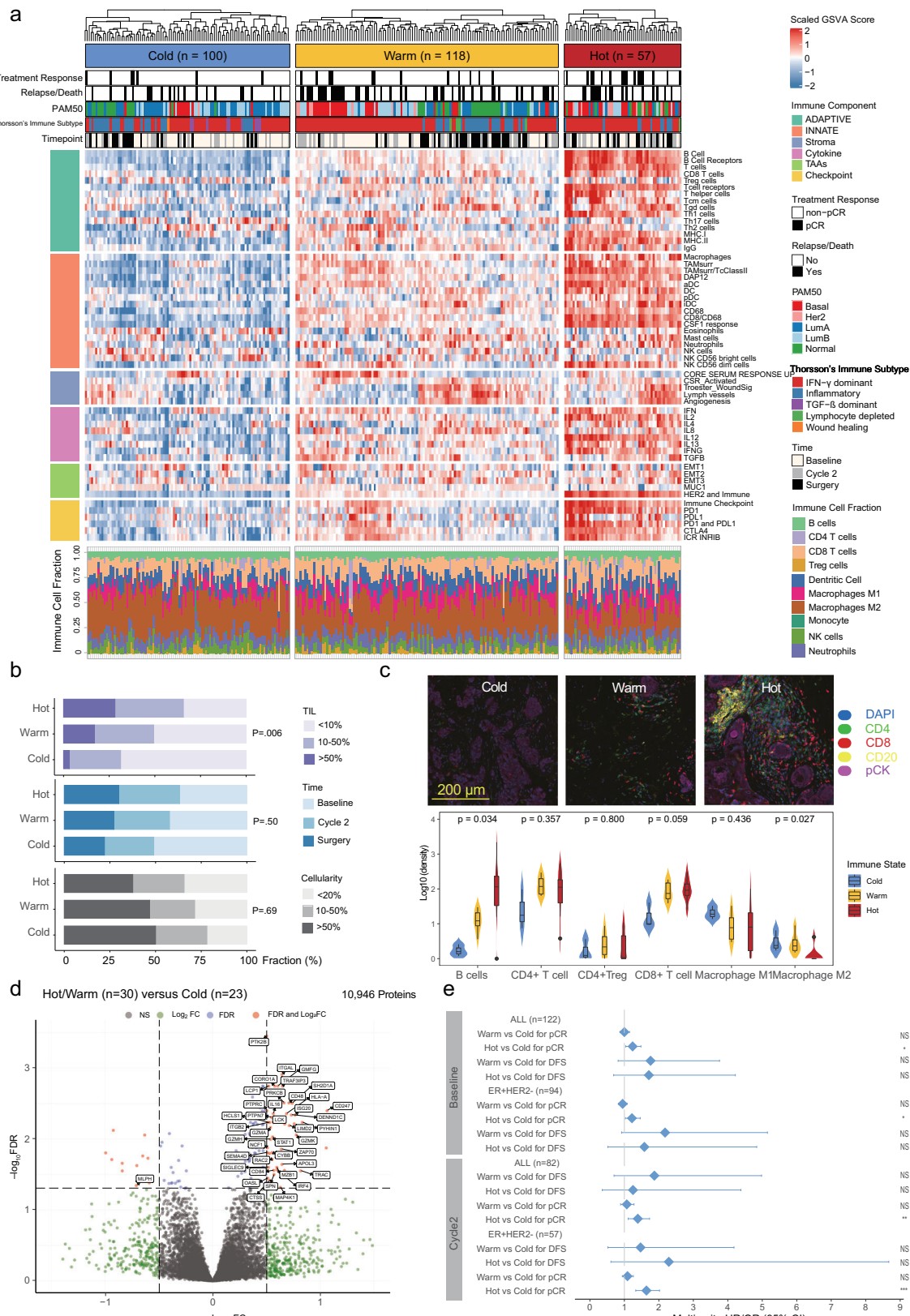

## Tumor metabolic phenotype interacts with immune state

Given the complexity of metabolites in bulk tumor tissues[10] and positive correlation between actual metabolite abundance and expression of corresponding metabolic pathway genes[23,24], we explored the interaction between the derived immune states and metabolic phenotypes using GEP and confirmed in MS proteomics. First, we classified

tumors into three metabolic states (i.e downregulated, neutral, upregulated) for each of the seven predefined metabolic pathways using GEP of tumor cells only (Fig. 3a). We applied ISOpureR to 270 samples with tumor cells, with estimated mean tumor purity of 0.58, 0.50, 0.31 at baseline, cycle 2 and surgery, respectively ($P < 0.001$, ANOVA), and extracted mRNA expression for tumor cells only and tumor-adjacent

**Fig. 2 | Immune state association with clinical outcomes. a** Unsupervised clustering using a joint latent variable model based on immune gene signatures and immune cell composition reveals distinct immune states of breast cancer (Supplementary Table2). **b** Distribution of TILs, sampling timepoints, and cellularity among the immune states, which is tested by two-sided chi-squared test or Fisher's exact test (frequencies < 5). **c** Representative mfIHC images (stained for lymphocytic, macrophage and epithelial markers, i.e CD4, CD8, CD20, CD163, CD68, FoxP3 and Cytokeratin), for three immune states, and immune cells density (number of positive cells normalized to tissue area) between cold ($n = 3$), warm ($n = 6$) and hot ($n = 7$) tumors. Box plot wrapped in violin plot bounds the interquartile range divided by the median, with the whiskers extending to a maximum of 1.5 times the interquartile range beyond the box. Statistical significance ($P$-value) was

determined using Kruskal-Wallis test. **d** Volcano plot showing differentially abundant proteins between hot/warm ($n = 30$) and cold tumors ($n = 23$), where the arrow indicates immune-related proteins. **e** Forest plot of multivariable Cox and Logistic regression analysis ($n = 204$) for DFS (HR with 95% CI) and pCR (OR with 95% CI), respectively, adjusting for IHC-based breast cancer subtype, tumor size and lymph node status (KI67 mRNA was additionally adjusted for DFS) (Supplementary Table3). The hazard ratios and odds ratios are shown with 95% confidence intervals. ***$p < 0.001$; **$p < 0.01$; *$p < 0.05$; NS $p > 0.05$. TILs tumor-infiltrating lymphocytes, mfIHC multiplex immunofluorescence immunohistochemistry, DFS disease-free survival, pCR pathologic complete response, ER estrogen receptor, HER2 Human epidermal growth factor receptor-2, OR odds ratio, HR hazard ratio, CI confidence interval.

cells (Supplementary Data. 2). As expected, $t$-statistic Stochastic Neighbor Embedding ($t$-SNE)[34] revealed seven GEP-based metabolic pathways distinct between tumor-adjacent cells versus tumor cells (Supplementary Fig. 4). Then, we classified those 270 samples into the three metabolic states (downregulated, neutral, upregulated) using bulk and tumor cell-based gene expression (Supplementary Data. 2). Figure 3b shows the proportions of bulk/tumor cell-derived metabolic states within immune states, for all patients and separately per breast cancer subtype (luminal and triple negative). Differential metabolic tumor-cell GEP between each metabolic state (FDR < 0.1) are shown in Supplementary Fig. 5, where the Kolmogorov-Smirnov test indicated that FDR values (derived from ANOVA tests) of each metabolic pathway are lower than those from other coding genes (all $P < 0.05$) (Supplementary Data. 3). Furthermore, representative metabolic protein abundance (Supplementary Fig. 6a-d) as well as KEGG metabolism signatures (Supplementary Fig. 6e–k) were compared between groups (all $P \leq 0.1$), which also showed a good concordance with tumor-cell based metabolic group.

Next, we investigated the interaction between metabolism and immune states and the prognostic implications of the different metabolic subtypes at the gene expression level. Overall, a linear mixed-effects model (LMEM) adjusted for tumor subtype, showed that metabolic gene sets states were associated with immune states. Upregulated lipid (coefficient, −0.25; $P = 0.002$), amino acid (coefficient, −0.23; $P = 0.0007$), TCA cycle (coefficient, −0.21; $P = 0.0007$) and vitamin/cofactors (coefficient, −0.19; $P = 0.03$) metabolic pathways were inversely correlated with immunologically hot tumors (Fig. 3b, c). We further assessed pair-wise correlations of the seven different metabolic pathways (tumor cell/bulk GEP-based) and immune state (Fig. 3d). TCA cycle subtype shared strongly positive correlations (Spearman's Rho > 0.4, $P < .001$) with other metabolic subtypes other than carbohydrate subtype, and both bulk and tumor cell GEP-based metabolic states were negatively associated with immune state (Spearman's Rho < −0.1, $P < 0.1$). When we combined pre-treatment and on-treatment samples, more prognostic metabolic phenotypes derived from tumor cell GEP were identified than those from bulk GEP (Fig. 3d and Supplementary Table. 5). Specifically, upregulated tumor cell GEP-based metabolic subtypes in carbohydrate (multivariable-adjusted HR = 2.62, 95% CI, 1.07–6.44, $P = 0.04$) and TCA cycle pathways (multivariable-adjusted HR = 2.89, 95% CI, 1.16–7.21, $P = 0.02$) were associated with worse DFS compared with the downregulated group. Likewise, patients with upregulated amino acid (multivariable-adjusted OR = 0.87, 95% CI, 0.76–1.01, $P = 0.07$), TCA cycle (multivariable-adjusted OR = 0.87, 95% CI, 0.74 to 1.03, $P = 0.1$) and nucleotide (multivariable-adjusted OR = 0.77, 95% CI, 0.61 to 0.98, $P = 0.04$) pathways-based subtype were less likely to attain pCR than those with downregulated pathways.

The reproducibility of the interaction between metabolic pathway-based subtype and immune state was externally validated by the expression profiles of the Korean cohort (Supplementary Fig. 7a–c). Although we failed to extract TC GEP due to lack of RNA-seq data of post-treatment samples that reached pCR, the tumor cellularity

was added into LMEM (Supplementary Fig. 7b) and multivariate logistic regression. Metabolic-pathway based phenotype like TCA cycle (coefficient, −0.22; $P = 0.003$) and nucleotide (coefficient, −0.26; $P = 0.03$) were negatively correlated with immune states, but upregulated vitamin/co-factors was associated with hot immune state (coefficient, 0.37; $P = 0.03$) (Supplementary Fig. 7b). Moreover, patients with pre-treatment upregulated amino acid phenotype were less likely to reach pCR compared to downregulated group (multivariable-adjusted OR = 0.71, 95% CI, 0.54–0.94, $P = 0.02$) (Supplementary Fig. 7c).

Overall, here we uncovered a critical interplay between tumor cell GEP-based metabolic phenotype (i.e. TCA cycle) and immune state subtype, and highlighted the prognostic role of cellular metabolism.

## MS-based proteomic landscape of immunometabolic phenotype and pathways

Protein abundance data were generated based on a subset of pre/on-treatment samples ($N = 53$) using mass spectrometry-based proteomics. No significant batch effect between six TMT sets was detected (Supplementary Fig. 8). In total, we identified 10,946 proteins (median (interquartile range): 8300 (8090-8948)), of which 7357 proteins were quantified in each of the 53 tumors (29 samples on pretreatment and 24 samples after two cycles of chemotherapy) (Supplementary Fig. 9a and Supplementary Data. 4). Gene based correlation of mRNA and protein data for 42 tumors showed positive correlations for 4585/8,290 (55%) proteins (median r: 0.37, Fig. 4a), which is comparable to previous proteogenomic studies[35,36]. Moreover, correlation between protein interactions of known complex members from Biogrid or CORUM differed between mRNA-mRNA and protein-protein correlations (Fig. 4b, Supplementary Fig. 9b and Supplementary Data. 5). The higher correlations at the protein level compared to the mRNA level demonstrated that those biological processes are tightly regulated at the protein level. Proteins related to metabolic and immune functions showed varying correlations with their respective transcript abundances (Fig. 4c), where immune proteins (mean r: 0.58) were significantly more correlated with corresponding mRNA than those of metabolism (mean r: 0.50) ($P = 0.002$) (Supplementary Fig. 9c).

After excluding protein pairs with weak correlations (absolute Pearson's $r < 0.3$), we mapped the protein correlation network using immunometabolic and breast cancer-specific (PAM50) proteins. The network snapshot demonstrated that proteins with similar biological functions were highly connected (Fig. 4d). Interestingly, lipid proteins were enriched in the immune-related module (Fig. 4d), supporting the importance of lipid metabolism in immune functions such as antigen presentation and T cell activation[37]. The mean abundance of immunometabolic proteins differed across the three immune states (Fig. 4e), which highlighted the role of the TME in shaping the tumoral metabolic landscapes. Hot/warm tumors were likely to have higher mean protein abundance in amino acid ($P = 0.03$, Fig. 4e) and nucleotide ($P = 0.07$, Fig. 4e) metabolism compared with cold tumors, probably owing to extra nutrition demands from functional immune cells. Interestingly, though hot tumors were associated with increased

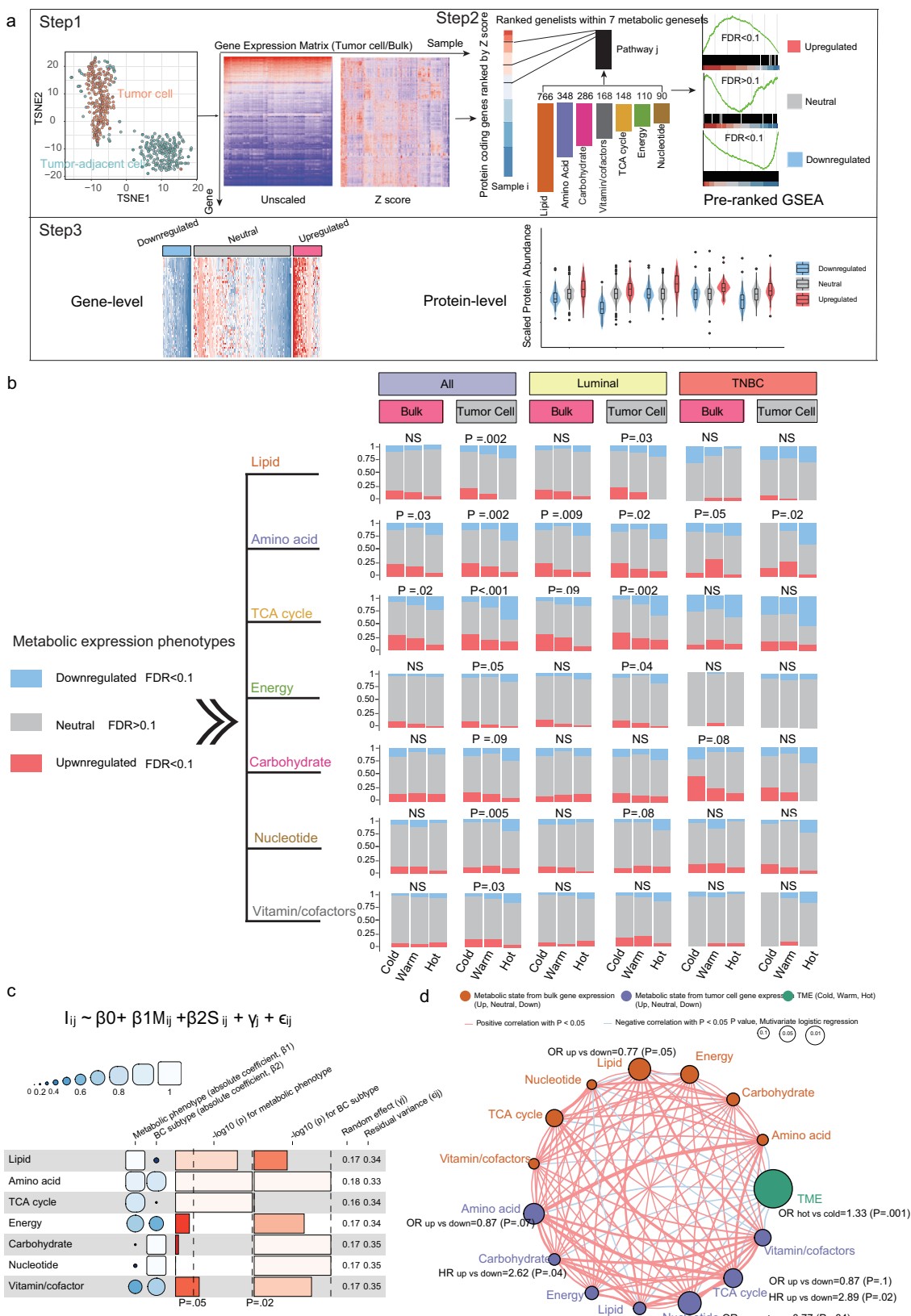

proliferation (bulk MKI67 protein) (Supplementary Fig. 9d), MKI67 percentage according to IHC that only counts tumor cells was not different between the three immune states at baseline (Supplementary Fig. 9e). Therefore, we speculated that this upregulated proliferation/ nucleotide signaling in hot tumors observed from proteomics might be derived from immune cells. Furthermore, we identified metabolic proteins sustaining proliferative signaling through protein-protein correlation analyses, and proteins involved in glycolysis (e.g. SLC16A1, SLC2A3, HK2), glutamine (e.g. GLS, HMGCS1) and one-carbon metabolism (e.g. PHGDH, PSAT1, SHMT2, MTHFD2, MTHFD1L), showing positive correlations with cell proliferation proteins (e.g. MKI67, CCNB1, KIF2C) (Supplementary Fig. 9f).

**Fig. 3 | Tumor cell GEP-based metabolic states interaction with immune states.**
**a** Bioinformatics approach named PureMeta (https://github.com/WangKang-Leo/PureMeta) classified each sample into one of three GEP-based metabolic states (upregulated, neutral and downregulated) in seven metabolic pathways. Step 1, tumor cell gene expression profiling was extracted from bulk RNA microarray data using ISOpureR (1.1.3)[92], and GEP of surgical samples (*n* = 5) without tumor cells (T0, N0) were regarded as reference. Step 2, GSEA pre-ranked analysis based on the gene set of each specific metabolic pathway was conducted, and the phenotype was determined based on FDR and Z scores. Step 3, tumor cell GEP-based metabolic phenotype was further validated on gene and protein levels. Box plot wrapped in violin plot bounds the interquartile range divided by the median, with the whiskers extending to a maximum of 1.5 times the interquartile range beyond the box. **b** Percentage of metabolic states in seven metabolic pathways for the three immune subtypes/states (from Fig. 2a), based on tumor cell or bulk GEP. Two-sided *P*-values were derived from the Chi-Square test or Fisher's exact test. **c** Funky heatmap depicting the coefficient, random term, residual variance and *p*-value of LMEM, which was conducted within all samples to identify interaction effects between immune state (I) and tumor metabolic phenotypes (M), adjusting for the breast cancer subtype (S). **d** Interaction of metabolic phenotypes derived from bulk or tumor cell gene expression profiling and the immune states. Orange bubble, metabolic phenotype from bulk gene expression; Purple bubble, metabolic phenotype from tumor cell gene expression; Green bubble, immune state. The size of each cell represents pCR prediction of each phenotype, the calculation used the formula $\log_{10}$(multivariable logistic regression two-sided *P*-values) (for details, see Supplementary Table 4). The lines connecting bubbles represent immunometabolic interactions. The thickness of the line represents the strength of correlation estimated by Spearman correlation analysis. A positive correlation is indicated in red and a negative in blue. GEP, gene expression profiling; GSEA, gene set enrichment analysis; LMEM, linear mixed-effects model; pCR, pathologic complete response.

Metabolic reprogramming may provide novel therapeutic opportunities, and several metabolic enzymes have been found to be valuable drug targets for cancer therapy and modulation of supporting immune cells[24,38,39]. We mapped differentially expressed mRNA and proteins between hot and cold tumors on major metabolic pathways (Fig. 4f), to find potential targets with FDA-approved drugs that appeared to synergize with TME improvement in stopping tumor growth. Metabolic druggable proteome upregulated in cold tumors such as FASN, whose inhibitor (cerulenin) is a potential candidate for inhibiting tumor growth and simultaneously boosting TME function (Supplementary Fig. 9g). Likewise, other potential drug targets with experimental evidence like ACACA (inhibitor: KD-023), ALDOA and HMGCS1 were also identified (Supplementary Fig. 9h–j). To validate the above findings, we replicated our analyses using the Oslo2 proteogenomic cohort that showed similar results (Supplementary Fig. 10).

These results further emphasized the interaction of immune and metabolic phenotype in bulk transcriptomic and proteome levels, and systematically identified potential metabolic targets as TME modulation. Although our bulk MS-based proteomics revealed distinct metabolic characteristics across the three immune states, our data also suggest that single-cell level metabolomics or proteome will be more informative due to metabolic heterogeneity in TME[20].

**Longitudinal pairwise analyses on immunometabolism**
Longitudinal GEP and proteomic data from PROMIX trial provide a unique opportunity to assess the correlation between tumor biological characteristics (seen in Supplementary results, Supplementary Fig. 11) involving immunometabolism profiles, and the response to NAC. We employed a linear mixed-effects model (LMEM) and identified four consensus clusters using pairwise differentially expressed genes (DGEs) across treatment (on-treatment vs. pre-treatment, *n* = 137; post-treatment vs. on-treatment, *n* = 344; post-treatment vs. pre-treatment, *n* = 740) (Supplementary Fig. 11a and Supplementary Data. 6). These were enriched in immune response (C1), metabolism (C2), extracellular matrix (ECM) (C3), or tumor proliferation (C4) pathways (Supplementary Fig. 11b). Corresponding pathway scores were calculated and compared between sampling timepoints, where representative and top enriched pathways within each cluster are shown in Supplementary Fig. 11b, c. In tumor purity and subtype-adjusted LMEM, continuously downregulated interferons (IFNs)-based antitumor response was observed during NAC, while antigen-presenting cell (APC) signaling was elevated from baseline to 2-cycle NAC then decreased to lower levels in residual tumors/normal breast tissues at surgery (Supplementary Fig. 11c). Conversely, fatty acid related metabolic pathways such as triglyceride catabolism tended to be upregulated by NAC (Supplementary Fig. 11c). Other tumor-intrinsic signaling pathways, including ECM and cell proliferation, presented completely reverse profiles during NAC (Supplementary Fig. 11c). As

reported previously[40], we found that NAC led to a notable downregulation of proliferation. Given the strong correlation of ECM and stroma score (Rho = 0.8, *P* < 0.001) (Supplementary Fig. 11d), we identified upregulated ECM pathways score with reduced tumor cellularity during NAC (Supplementary Fig. 11c and Supplementary Fig. 11e).

We then limited analyses to correlation of dynamic immune or/and metabolic phenotype changes and treatment response within 69 patients with both pre- and on-treatment tumors (patients' characteristics shown in Supplementary Table 6). Pairwise comparisons of immune states indicated that 40% of cold tumors were converted into warm or hot states after two cycles of NAC (Fig. 5a, b). Likewise, 28% of the warm tumors became hot, but fewer tumors went from hot to cold (1/10) immune states under NAC. Interestingly, this effect on immune state was persistent during the last four cycles of NAC. Moreover, we defined as a negative immune state change if tumors kept a cold immune state or turned to a colder immune state after NAC, and vice versa for positive immune state change (Fig. 5c). Patients with positive immune state change (OR = 1.2, 95% CI, 1 to 1.45; *P* = 0.05) (Fig.5d) were more likely to achieve pCR after adjusting for tumor size, lymph node status and breast cancer subtype. The same strategies were also applied to define tumor-cell-GEP based metabolic phenotype profile (i.e. positive change: patients maintained downregulated metabolic phenotype under NAC, or with changed metabolic phenotype (i.e. from upregulated/neutral to downregulated)); all others are defined as negative change. Accordingly, we revealed that positive changes of TCA cycle (OR = 1.28, 95% CI, 1.03 to 1.58; *P* = 0.03) and nucleotide (OR = 1.41, 95% CI, 1.09 to 1.82; *P* = 0.01) metabolisms were independently associated with increased pCR. Similarly, patients with positive energy metabolism change showed a trend towards better treatment response (OR = 1.36, 95% CI, 0.96 to 1.94; *P* = 0.09). In addition, we evaluated the correlation between integrated immunometabolism profiles (Group1-4 shown in Fig. 5e) and radiologic response (response group, *N* = 28; no response group, *N* = 41) after two treatment cycles. We found that patients with both positive immune state and metabolic phenotype profile were more likely to respond to NAC compared with other groups (Fig. 5e and Supplementary Table 7).

To identify potential tumor drivers and biomarkers in immunometabolism during NAC, we conducted longitudinally differential mRNA and protein expression analyses separately, according to objective response status (partial response versus stable disease or disease progression) (Fig. 5f, g). Several immune-related proteins, including IL32, CCL18, CD247, and CD8A, showed a strong positive correlation with NAC response. Conversely, proteins on the TCA cycle and nucleotide metabolism were downregulated (Fig. 5f). Interestingly, in the "no response" group, we identified upregulated carbohydrate (SLC6A8, HS6ST2, SLC5A1) and exhausted CD8 + T cells (CD244) biomarkers exclusively at the protein level (Fig. 5g).

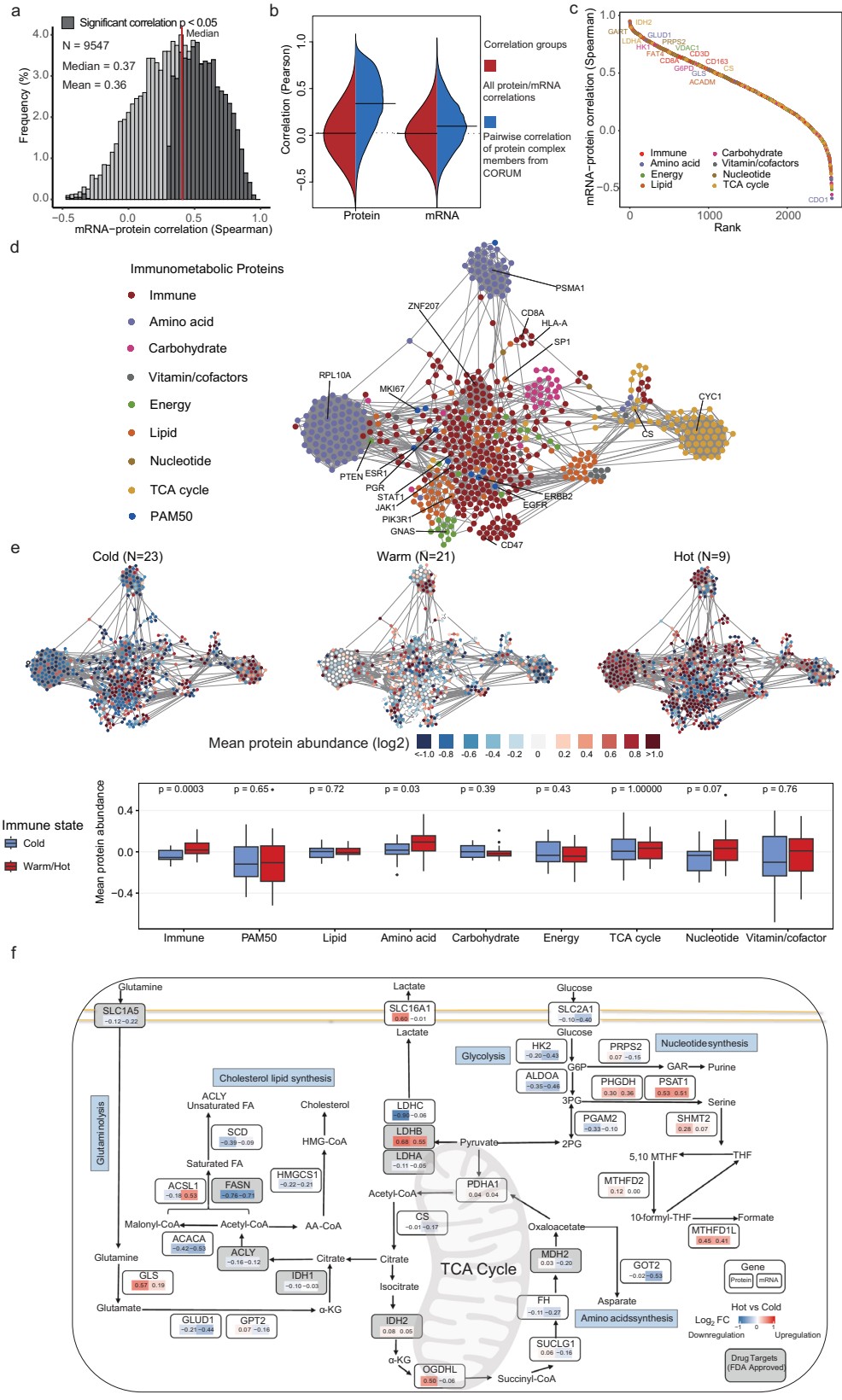

In summary, we demonstrate that both immune states and metabolic phenotypes in HER2-negative breast cancer are not stable but are dynamically shaped by NAC, with these early changes having important prognostic implications regarding both radiologic and pathologic response to treatment.

## Single-cell transcriptional analysis of metabolic states and reprogramming in TME

To better characterize metabolic heterogeneity and cellular composition in TME at single-cell resolution, we utilized longitudinal snRNA-seq data from the PROMIX trial (8 TNBC patients, 16 samples) and a

**Fig. 4 | MS-based proteomic landscape of immunometabolic phenotype and pathways. a** Correlation between protein and mRNA quantitative values (Spearman) of individual genes. Correlation coefficient is considered statistically significant if two-sided $P < 0.05$ (dark gray bars). **b** Comparison of all pairwise correlations to correlations from known interaction pairs from CORUM database, using quantitative protein and RNA levels across 53 tumors (see Supplementary Fig. 8b for the same analysis using Biogrid interactions). **c** Ranked mRNA–protein correlations. **d** Breast cancer protein correlation network based on immunometabolic and PAM50 proteins ($n = 2837$ in total) using > 0.3 Pearson correlation and KCore > 2 cutoff. **e** Visualization of average proteome quantification of the three immune subtypes/states (from Fig. 2a) in the correlation network. Boxplot showing difference of mean protein abundance of each module between cold ($n = 23$) and

warm/hot tumors ($n = 30$). Box plot bounds the interquartile range divided by the median, with the whiskers extending to a maximum of 1.5 times the interquartile range beyond the box. Outliers are shown as dots. Statistical significance (two-sided $P$-value) was determined using Wilcoxon rank-sum test. **f** Pathway diagram summarizing metabolic genes involved in the TCA cycle, glycolysis, nucleotide, amino acid, and cholesterol lipid synthesis and metabolism. Alterations are defined by significant upregulation or downregulation of protein abundance (left) and mRNA expression (right) between hot and cold tumors (expressed as $\log_2$(fold-change)). Red, upregulated genes/proteins in immunological hot tumor; blue, downregulated genes in the immunological hot tumor. The gray panel highlighted the FDA-approved drug targets. Figure4f created using Biorender (https://biorender.com/).

large-scale, well-annotated, treatment-naïve scRNA-seq cohort (21 HER2-negative breast cancers)[41].

In the PROMIX snRNA-seq dataset, we distinguished cancer cells from normal cell types including immune, stromal, normal epithelial cells using estimated genomic copy number profiles at an average genomic resolution of 5 Mb (Supplementary Fig. 12). Then, immune, stromal, normal and tumor epithelial cells compartments were classified by classic biomarkers (Supplementary Fig. 13). Cell subpopulations were further identified within each compartment (Supplementary Fig. 14). To evaluate compartment-specific metabolic heterogeneity, we conducted single-sample gene set enrichment analysis (ssGSEA) analyses of metabolic pathways in the PROMIX snRNA dataset, indicating that bioenergetics of normal breast cancer epithelial cells differed from cancer cells. Tumor epithelial cells shared global metabolic superiority to other compartments except for fatty acid and drug metabolism by cytochrome P450 (Supplementary Fig. 15a, b), and consistent results remained within each cell type (Supplementary Fig. 15c).

We next characterized metabolic profiles of breast epithelial cells under the pressure of NAC treatment. Five epithelial clusters in metabolism (MC1-MC5) were identified across longitudinally collected TNBC biopsies and surgical specimens (Fig. 6a, b). The composition of the metabolism-specific epithelial clusters changed between pre-treatment and on/post-treatment samples, which was significantly associated with immune state change (negative or positive) (Fig. 6c). Epithelial MC1 (PDK4, DCN, ACTA2, FABP4) was characterized by normal epithelial cells, antigen presentation (chemokines) and glycerolipid metabolism (Fig. 6d, e), and MC1 proportion increased after NAC in the group with positive immune state change. Within MC2 (FABP5, MTHFD2, TPI1, GAPDH), we identified a highly proliferated (purine, folate one carbon metabolism) epithelial cell subset with active bioenergetic metabolism features (i.e. OXPHOS, glycolysis) (Fig. 6d, e). MC2 proportion increased after treatment in the negative immune change group while it disappeared in the positive change group, indicating shrinkage of this cell subtype was associated with the good response to NAC (Fig. 6c). MC2 showed enrichment of glycolysis and hypoxia, while MC3 (COL1A1, MMADHC, PGK1, RAN, GLS, OAT) was mainly associated with OXPHOS and glutathione metabolism rather than glycolysis (Fig. 6d, e). Furthermore, the transition of MC2 through the other metabolic clusters was strongly supported by the trajectory analysis (Fig. 6f). Pseudo-time ordering demonstrated an ordered, progressive, stepwise transition from normal breast epithelial cells (MC1) to malignant hypoxic and glycolytic phenotype (MC2) (Fig. 6f). MC3 was probably converted into MC2 in the negative immune state change group (Fig. 6c). The composition of MC4 (FAU, ATP5G2, COX8A, COX5B), depending on amino acid and cholesterol (retinol metabolism, steroid hormone metabolism) (Fig. 6d, e), was relatively stable during NAC in either positive or negative immune state change group (Fig. 6c). Interestingly, MC5 (PIK3R1, ITPR2, LRP2) was mainly seen in pre-treatment tumor cells of positive immune state change group (Fig. 6c). The disappearance of MC5 on/post-treatment

in positive immune state change group could be explained by highly expressed chemokines, which directed the migration of immune cells into tumor tissue[42]. We performed differential expression analyses between pre-treatment and on/post-treatment by the immune state change group, to identify differential expression patterns in immunometabolism (Fig. 6g, h). In the negative change group, downregulated genes following NAC were associated with antigen presentation/major histocompatibility complex (MHC) (HLA − DRA, HLA-DPAI, HLA − B, HLA − C, HLA − DQB1, CXCL9/10/11) and metabolic genes were upregulated (RPL5, GAPDH, TPI1, DCXR, ATP5G2), and vice versa for patients in the positive change group (Fig. 6g, h). High gene expression of genes RPL5, GAPDH and TPI1 were potential therapy targets based on CRISPR screen (DepMap 21Q2) data (high gene expression associated with high gene dependency) (Supplementary Fig. 16a−c). Furthermore, we calculated GSVA scores of cancer antigen presentation and metabolisms for epithelial cells, demonstrating that dynamic metabolism changes of breast epithelial cells during treatment were associated with the immune state switch (all $P$-values of Two-way ANOVA test with test for interaction < 0.05) (Fig. 6i).

We applied the same strategies to immune and stromal cells to assess the metabolic heterogeneity of immune state. Interestingly, gene set of the seven metabolic pathways provided clear clusters by time or immune states (Supplementary Fig. 17a). Given the limited number of immune and stromal cells, we just focused on the cluster (baseline immune cells in the hot immune state) containing a high fraction of B cells and CD8 + T cells (>85%) (Supplementary Fig. 17a), with high expression of CD44 that is a receptor for extracellular matrix component hyaluronan and biomarker of activated and memory T cells[43] (Supplementary Fig. 17b). In addition, lysosomal acid lipase A (LIPA), which mobilizes fatty acids for FAO for memory CD8 T cell development[44], was enriched in this cluster (Supplementary Fig. 17b), which was correlated with CD8A based on bulk GEP (Supplementary Fig. 17c). Other critical metabolic genes like SDHD (TCA cycle) and ASAH1 (lipid metabolism), were also found to be upregulated (Supplementary Fig. 17b). To capture special metabolic features for hot tumors, we compared metabolic GSVA score between hot and warm/cold tumors. Interestingly, immune cells within cold/warm TME characterized hypoxia, glycolysis, and glutathione (Supplementary Fig. 17d). Upregulated kynurenine and tryptophan pathway signaling were identified among immune cells within the hot immune state group. Interestingly, we noted an upregulation of indoleamine 2, 3-dioxygenase 1 (IDO1), a protein recently identified as immune checkpoint target, characterized as a rate-limiting metabolic enzyme that converts tryptophan (Trp) into downstream kynurenines[45]. Markedly upregulated IDO1 was seen on both immune and tumor cells in hot/warm vs. cold immune state groups (Supplementary Fig. 17e).

As complement to inherent limitations of the used snRNA-seq from PROMIX trial method[46] for depicting immune and stromal cells (seen in Supplementary results, Supplementary Fig. 17), we conducted additional analyses mainly on metabolic profiles within immune and stromal cells using HER2-negative subset of breast cancer single-cell

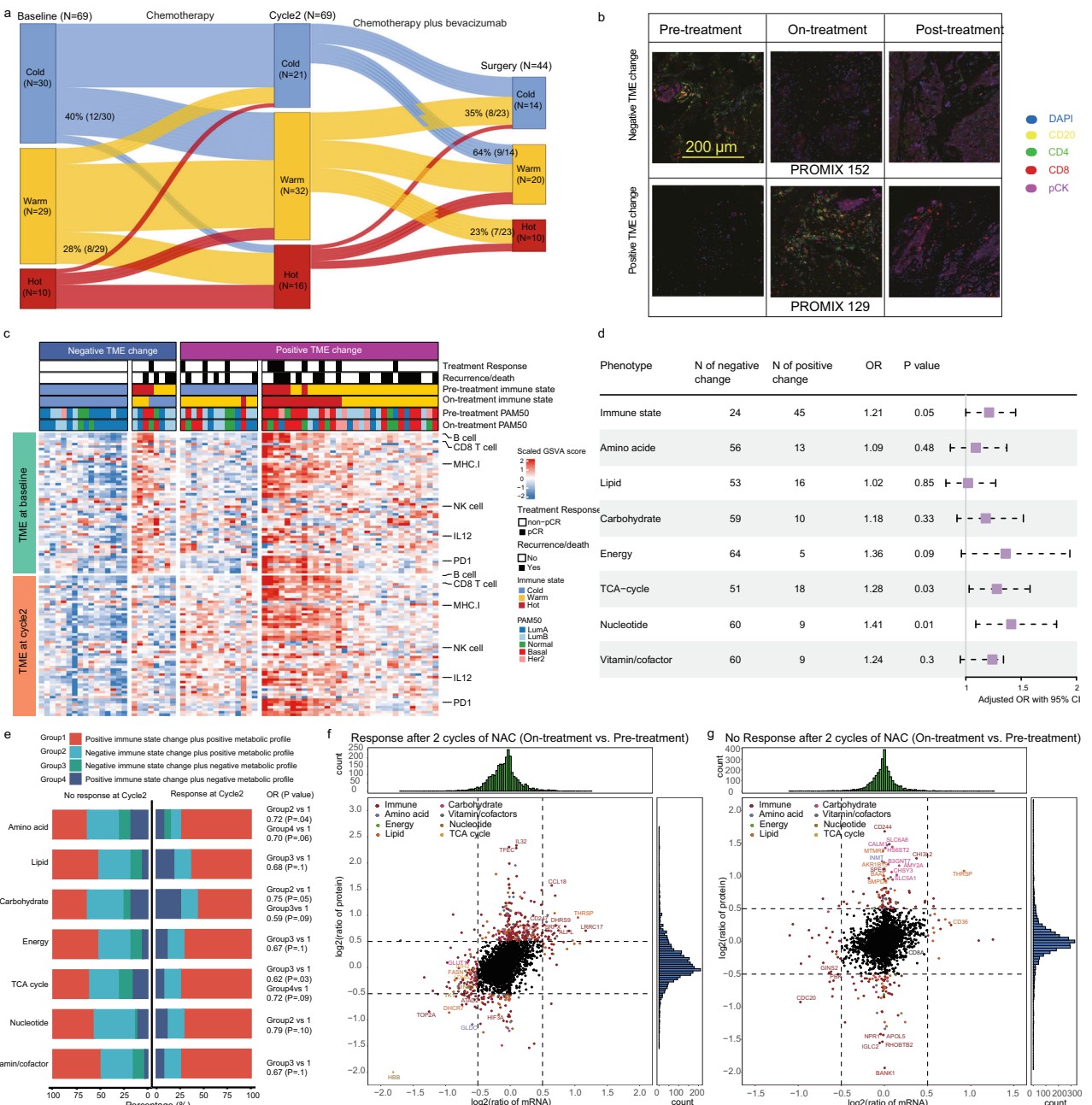

**Fig. 5 | Proteogenomic analyses of paired pre/on-treatment samples revealed an association between changes in immunometabolism and response to NAC.** **a** Sankey plot of immune state changes from pre-treatment (baseline) to on-treatment (cycle2) and from on-treatment to post-treatment (surgery) (Supplementary Table 6). Numbers denote the number of samples in each immune state and the percentage of samples that switch from one immune state to another. **b** Representative mfIHC images from 16 independent experiments for patients with positive and negative immune state change, respectively. PROMIX patient 152: with negative immune state change, had early distant metastasis after 10 months of diagnosis occurred. PROMIX patient 129: with positive immune state change, had long-term DFS (66 months). **c** Immune signatures for 69 paired pre/on-treatment samples. Negative immune state change was defined if tumors conserved cold immune state or turned to a colder immune state after NAC, and vice versa for

positive immune state change. **d** Forest plot depicting multivariable the logistic regression model (OR with 95% CI) adjusting for tumor size, lymph node status, and IHC subtype that assess the association between immunometabolism profiles and treatment response (pCR). **e** Bar plots showing the distribution of integrated immune state with tumor cell GEP based metabolic state change among response and no-response groups after two cycles of NAC (partial response versus stable disease or disease progression). Multivariable logistic regression was fitted, adjusting for tumor size, lymph node status, and IHC subtype (Supplementary Table 7). **f, g** Scatterplot showing pair-wise differential protein (y-axis) and mRNA (x-axis) expression between on-treatment and pre-treatment samples in the response group and no response group (partial response versus stable disease or disease progression), respectively. The x/y axis shows the log₂ (fold change). OR, odds ratio; CI, confidence interval.

atlas dataset (GSE176078) (Fig. 7a)[41]. Interestingly, although breast epithelial cells (Supplementary Fig. 18a), immune cells (Fig. 7b), or stromal cells (Fig. 7c) predominantly depended on OXPHOS, metabolic flexibility and variation within the TME were identified (Supplementary Data. 7). Specifically, in epithelial cells, glycolysis was the most

critical metabolic pathway besides OXPHOS, and was enriched in basal-like and cycling cancer cells that were highly proliferative (all GSEA FDR < 0.05) (Supplementary Fig. 18a and Supplementary Data 7). Likewise, glycolysis was also enriched in immune effector cells including memory B cells, CD8 + T cells, cycling T cells, monocyte and

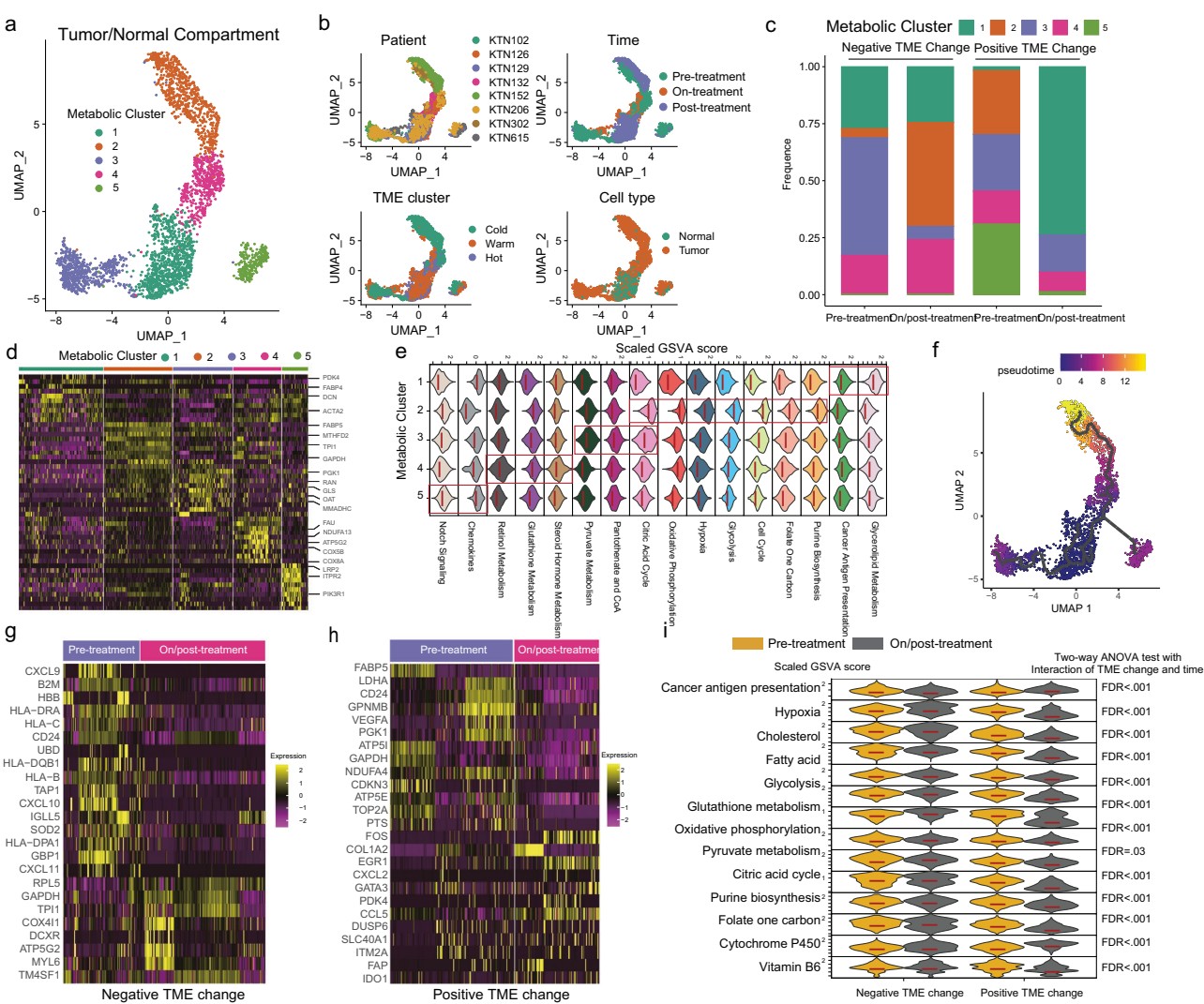

**Fig. 6 | Identification of metabolic reprogramming during immune state evolution by longitudinal single-nucleus RNA-seq. a** Single-nucleus RNA-seq using gene sets of the seven metabolic pathways identify five metabolic clusters (MC1-5) of breast epithelial cells pre/on/post-treatment breast cancer in PROMIX trial (*n* of sample = 16, *n* of cell = 3,039). **b** Feature plots showing sample ID, sampling timepoint, TME subtype, and cell type in each metabolic breast epithelial cell subcluster. **c** The percentage of metabolic epithelial cell cluster in each sampling timepoint by immune state change. **d** Heatmap of the top 10 differentially expressed genes compared to all other clusters in (**a**), and the arrow indicated metabolic-related genes. **e** The metabolic characteristics of each breast epithelial cell cluster (MC) were analyzed and quantified based on metabolic signature scores: MC1 (glycerolipid metabolism, log$_2$FC = 7.4; cancer antigen presentation, log$_2$FC = 2.5), MC2 (purine biosynthesis, log$_2$FC = 2.5; folate one carbon, log$_2$FC = 2.7; cell cycle,

log$_2$FC = 3.0; glycolysis, log$_2$FC = 3.2; hypoxia, log$_2$FC = 3.1; oxidative phosphorylation, log$_2$FC = 3.0; citric acid cycle, log$_2$FC = 2.7), MC3 (citric acid cycle, log$_2$FC = 0.8; pantothenate and CoA, log$_2$FC = 1.6; pyruvate metabolism, log$_2$FC = 1.8), MC4 (steroid hormone metabolism, log$_2$FC = 2.5; glutathione metabolism, log$_2$FC = 0.8; retinol metabolism, log$_2$FC = 2.7), MC5 (chemokines, log$_2$FC = 1.7; notch signaling, log$_2$FC = 0.8). All FDR < 0.05. **f** UMAP of metabolic breast epithelial cell clusters, colored by pseudotime, calculated using Monocle3 (1.3.4). **g**, **h** Heatmap of the expression levels of differentially expressed genes between pre- and post-treatment breast epithelial cells by immune state change. **i** Stacked violin plots of metabolic gene signature scores for pre- and post-treatment breast epithelial cells that significantly interacted with immune state change, where the FDR values derived from a Two-way ANOVA test with Interaction of TME change and time.

cycling myeloid (Fig. 7b and Supplementary Data 7), which was revealed by a series of nuanced models[47–56] investigating the metabolism of T cell expansion and CD8+ effector differentiation. Cancer-associated fibroblasts (CAFs) (myofibroblast-like CAFs and inflammatory-like CAFs) and endothelial cells, serving as major components of tumor stroma and ECM, showed metabolic plasticity and shared similar metabolic activity (OXPHOS, glycolysis, glutathione, cytochrome P450) (Fig. 7c and Supplementary Data 7). The 21 tumors were further classified into immunologically cold (*n* = 13) and hot (*n* = 8) based on mean CD8 + T cell proportion as cut-off value (Supplementary Fig. 18b). Then we calculated represented metabolic

pathway GSVA score using pseudo-bulk gene profiles for immune and stromal cells for each sample, which were compared between hot and cold tumors (Fig. 7d). Immune effector cells from hot tumors harbored higher metabolic activity than counterpart cells from cold tumors, including pyruvate (CD4 + T cells and NK cells), glycolysis (NK cells), citric acid cycle (CD8 + T cells and NK cells), and fatty acid metabolism (CD4 + /CD8 + T cells) (Fig. 7d). Conversely, amino acid (glutathione) metabolism signature score derived from CD4 + T cells was higher in cold tumors compared with hot tumors (Fig. 7d), and we observed similar tendencies for cycling T−cells and memory B cells (both *P* = 0.1) (Fig. 7d).

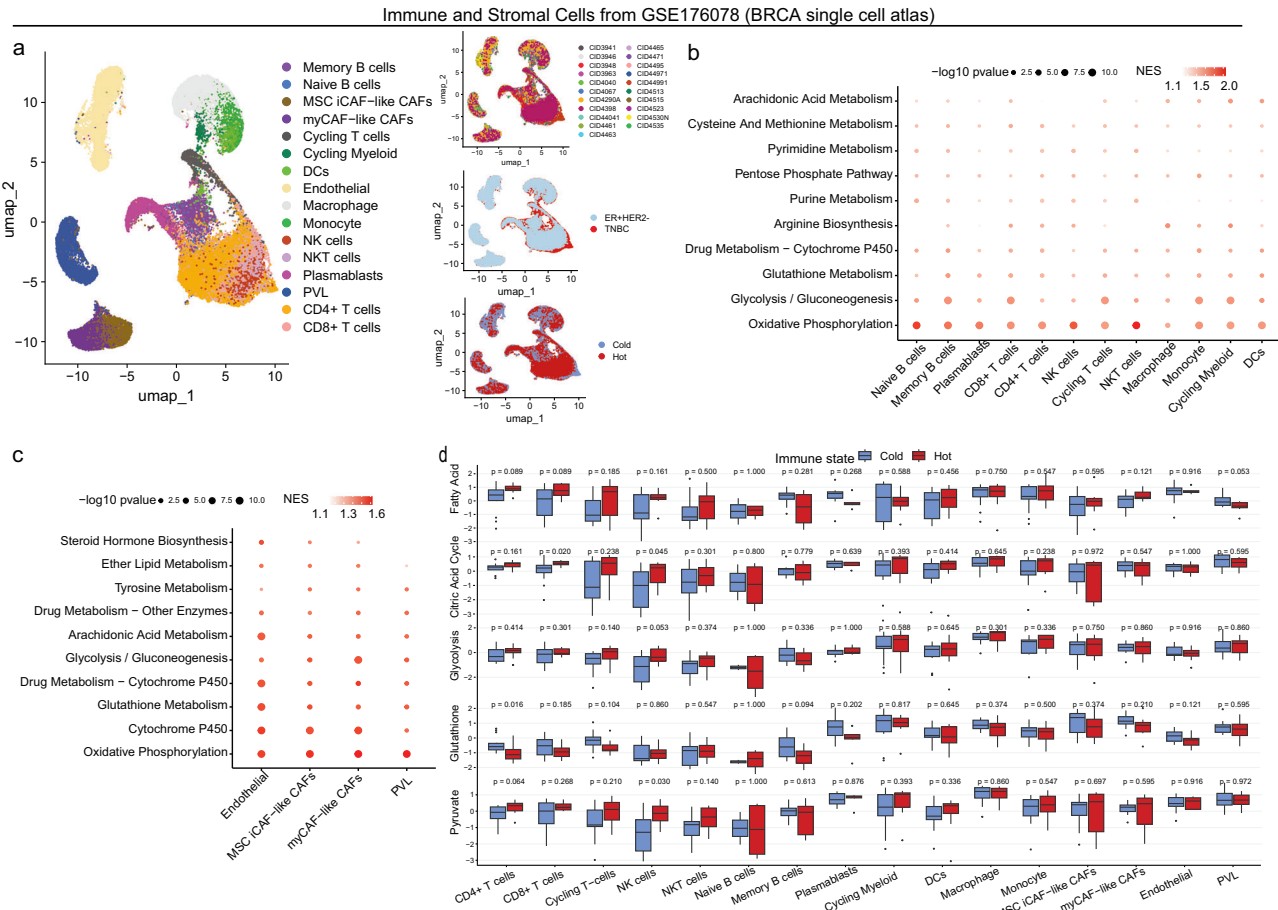

**Fig. 7 | Metabolic heterogeneity of immune and stromal cells. a** UMAP dimensionality reduction diagram showing immune and stromal cell from HER2-negative subset of breast cancer single cell atlas (GSE176078) by sampleID, subtype, and immune state (*n* of sample = 21, *n* of cell = 54,652). Cell type was previously well annotated as follows: B cells Memory (memory B cells (CD79A, MS4A1, CD27), naive B cells (CD79A, MS4A1, IGHD), plasmablasts (IGKC and IGLC2), CD8 + T cells (CD3, CD8), CD4 + T cells (CD3, CD4), NK cells (KLRC1, KLRB1, NKG7, AREG), cycling T cells (CD3, MKI67), NKT cells (KLRC1, KLRB1, NKG7, FCGR3A), macrophage (CD86), monocyte (CD127), cycling myeloid (KI67), DCs (CLEC9A or CD1C), endothelial (PECAM1, CD34 and VWF), MSC iCAF−like CAFs (ALDH1A1, KLF4 and LEPR), myCAF−like CAFs (ACTA2 (αSMA), TAGLN, FAP and COL1A1), PVL (ACTA 2, PDGFRB and MCAM)). Metabolic pathways enriched in genes with highest contribution to the metabolic heterogeneities among immune cells (**b**) and stromal cells (**c**). The metabolic pathways with GSEA nominal *p*-value < 0.05 were considered as significant. **d** Represented metabolic pathways score of immune and stromal cells was compared between cold (*n* = 8) and hot (*n* = 7) tumors. Box plot bounds the interquartile range divided by the median, with the whiskers extending to a maximum of 1.5 times the interquartile range beyond the box. Outliers are shown as dots. TME tumor micro-environment, FC fold change, FDR false discovery rate, CAF cancer associated fibroblast, MSC mesenchymal stem cells, iCAFs inflammatory-like CAFs, DCs dendritic cell, PVL perivascular-like.

In summary, using snRNA-seq, we here characterized the metabolic states and their dynamic evolution under the pressure of therapy in the different cell types within breast tumors. As in the bulk tumor analyses described above, significant associations between the tumor metabolic activity and the immune state of the tumors were described. More importantly, we also implied the metabolic differences that existed in immune effector cells from cold and hot tumors.

**In vitro validation of immunometabolic targets**
Following extensive analyses of bulk GEP, proteomic data and snRNA-seq, we demonstrated that various metabolic-related genes were upregulated in cold tumors (i.e. *FASN, ALDOA, HMGCS1, ACACA,* seen in Fig. 4f) or presented in on/post-treatment tumors with negative immune state change (i.e. *RLP5, GAPDH, TPI1, DCXR,* seen in Fig. 6g). Therefore, to further substantiate these findings and gain functional insights into the immunometabolic interplay, we performed in vitro studies targeting these genes. In order to establish our experimental set of HER2-negative human breast cancer cell lines, we used publicly available cell line transcriptomic data (Supplementary Fig. 19a), and observed that baseline expression of three of these metabolic-related

genes (i.e. *RLP5, TPI1, ALDOA*) was most commonly upregulated in the following breast cancer cell lines: MDA-MB-231 & BT549 (basal-like subtype), MCF7 and T47D (luminal subtype). By using siRNA transient transection technology, we successfully performed knockdown of the three genes in all 4 cell lines (Supplementary Fig. 19b). Upon knockdown of the metabolic-related genes, we observed a decreased cell viability (evaluated by XTT cytotoxicity assay, Fig. 8a, b) and increased apoptosis (evaluated by caspase 3/7 assay, Fig. 8c, d and Supplementary Fig. 19c) of the tumor cells with the knocked-down genes compared to the control cells.

Given the importance of the anti-tumor activity of T cells in the TME to eliminate tumor cells, we next conducted co-cultures of T-cells with the aforementioned cancer cell lines (both control and gene-silenced) and assessed for tumor killing over the course of a 24 h live cell-imaging. Upon knockdown of the metabolic-related genes, direct tumor cell killing was observed for most cell lines and target genes when cultured with T-cells, especially for *ALDOA* in both luminal and basal-like cells, and *TPI1* in luminal cells (Fig. 8e). Furthermore, tumor cell growth was arrested when cultured with T cells and upon gene silencing, as demonstrated by a decreased confluence compared to

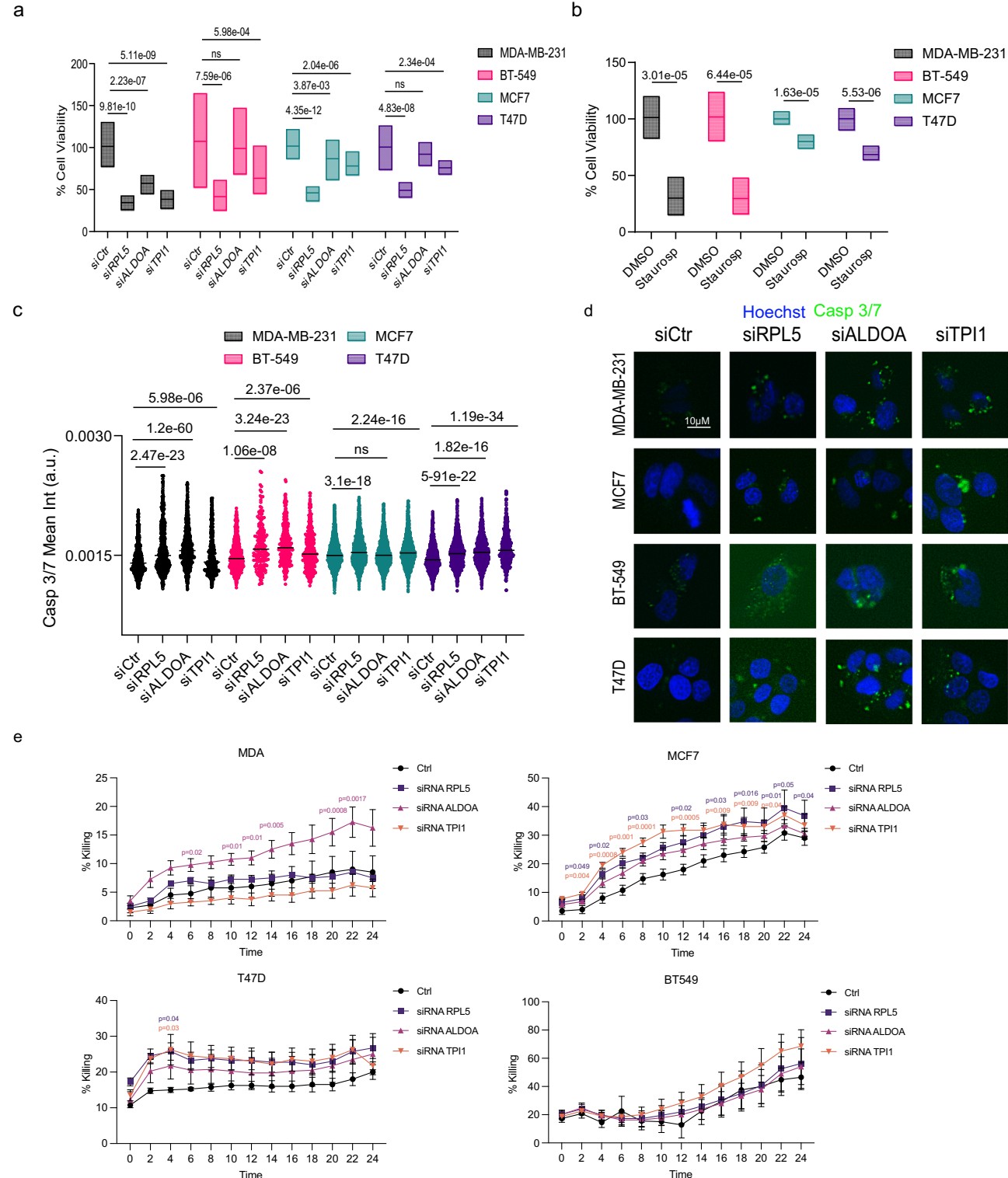

**Fig. 8 | In vitro validation of TME-related metabolic targets. a**, **b** XTT assay of two luminal (MCF7, T47D) and two basal-like (MDA-MB-231, BT-549) breast cancer cell lines following a. the knock-down of RPL5, ALDOA or TPI1 or (**b**). treatment with Staurosporine. All growth assays were performed twice in sextuplicates (*N* = 12, knock-down) or triplicates (*N* = 6, Staurosporine). Data are shown as mean ± SD, Students *t*-test (treatments compared to siCtr or DMSO), ns: non-significant. The percentage of viable cells is shown inside the bards. **c** Cell apoptosis is shown as mean intensity (arbitrary units) of Caspase 3/7 following a 72-h knockdown of the same genes in the cell lines mentioned in (**a**). All apoptotic assays were performed twice with a total of *N* = 1000–2000 cells counted per experimental condition. Data

are shown as mean ± SD, Students *t*-test (two-sided) (treatments compared to siCtr), ns: non-significant. **d** Representative immunofluorescence images used for the analysis presented in (**c**). Scale bar = 10 μm. **e** Tumor cell killing percentage following T-cell co-culturing in control cell lines and upon knock-down of RPL5, ALDOA or TPI1 for 24 h and 2 h time-lapse using live cell imaging. Cumulative data (*n* = 4) are shown as mean ± SEM, experiments were repeated twice with three technical replicates and eight biological replicates. Statistical analyses were performed using Dunnett's multiple comparisons test presenting adjusted *p*-values (two-sided). TME tumor microenvironment.

control cells (Supplementary Fig. 19d). Taken together, our in vitro data support the hypothesis that targeting metabolic genes may lead to immune cell-mediated tumor cell killing and tumor growth inhibition, thereby provide insights for future studies in this field.

## Co-evolution of tumor and immune state during NAC

To gain insights on how tumor cell and immune states co-evolve under TME, we conducted clonality analyses on WES data[28] of longitudinal tumor samples from 20 TNBC patients in the PROMIX trial. Median sequencing depth across all samples was 132X (range 97-191), with a 99.9% coverage rate (mean), and median tumor composition calculated by PureCN was 31%, 16%, and 15% at pre/on/post-NAC, respectively. We identified 4142 unique somatic mutations based on two callers in this cohort, including 3218 single nucleotide variants (SNVs) and 1345 insertion/deletion variants (Indels) (Supplementary Data 8). The most common somatic mutations (cancer drivers) included TP53 (61%) and PIK3CA (20%) (Supplementary Fig. 20a and Supplementary Data. 8), and were seen more often in warm/hot than cold tumors. Furthermore, we identified additional breast cancer genes deleted more frequently in warm/hot tumors than in cold tumors (BAP1, PBRM1, NOTCH1, CCND1, RB1) (Supplementary Fig. 20b, c).

We observed a tendency for baseline hot/warm tumors to be more heterogeneous than those with a cold immune phenotype (mean subclone percentage (hot/warm vs. cold): 36.7% vs. 28.5%, $P = 0.26$) (Fig. 9a). Tumor mass contains clones of different fitness, and heterogeneous clones co-existed in the absence of selection pressure. However, chemotherapy-induced extinction of weaker clones produced tumors dominant by chemo-resistant clones[57]. Accordingly, we identified a lower number of subclones in on/post-treatment hot/warm tumors than cold tumors (mean subclone percentage (hot/warm vs. cold): 21.3% vs. 36.1%, $P = 0.08$) (Fig. 9a). Post-treatment cold tumors also acquired more cancer-specific mutations (i.e., *BRCA2, TPR, OMD, RANBP2, EP3OO*) (Supplementary Data. 8). Mutations were filtered to include only coding non-synonymous SNV and Indels, and were then used to classify clonal evolution status. The status was defined as clonal extinction if >90% of mutations present at baseline disappeared following NAC or <10 mutations existed in post-treatment samples (Supplementary Fig. 20d, e), and all other cases were defined as clonal persistence. Somatic copy number alteration (SCNA) profiles were in line with the changes of mutation number in each clonal evolution group, respectively (Supplementary Fig. 20f, g). Interestingly, we found that immune state change was correlated with clonal evolution during treatment ($P = 0.002$) (Fig. 9b). Tumors with positive immune state change were more often associated with clonal extinction (7/9), whereas negative immune state change was associated with clonal persistence (11/11).

To further uncover therapeutic vulnerabilities based on the clonal evolution of tumors, we employed PhylogenicNDT[58] to calculate the differences in growth rate ($\Delta GR$) (Prob[$\Delta GR > 0$] > 0.95, limited to $P < 0.05$) between child and parent clones, where putative drivers were defined using cancer consensus gene[59]. Here we focused on subclones detected in 20 TNBC patients receiving NAC (Fig. 9c, d and Supplementary Figs. 21, 22). Indeed, two subclones from 2 patients (patients 152 and 310) who both had negative immune phenotype change and clonal persistence contained known metabolic drivers and their growth was significantly higher than their parent clone/subclone (all *P*-value < 0.05, Fig. 9c). The strongest accelerations were associated with second hits in lipid metabolism drivers, such as Acyl-CoA Synthetase Long Chain Family Member 3 (*ACSL3*) and Cyclin C (*CCNC*) ($\Delta GR$ of 128% and 320% for subclone 3 of patient 152 and subclone 2 of patient 310, respectively). We further saw strong growth acceleration in patient 206 who had positive immune state change and clonal extinction, with mutation of the mediator complex subunit 12 (*MED12*) that was related to chemoresistance[60]. Conversely, those subclones

with known breast cancer drivers (patients 155, 115 and 612, Fig. 9d and Supplementary Fig. 22) showed no growth rate advantage compared with their parent clones. Overall, these analyses provided evidence of TME and tumor co-evolution, highlighting that treatment might result in the emergence of immune-suppressed TME and selection of resistant subclones.

## Discussion

Using integrated temporal proteomic and genomic profiling of breast cancer during NAC, this study highlights the evolution of therapeutic response biomarkers in metabolism under the pressure of chemotherapy. We have shown that mutual metabolic requirements of tumor and immune cells contribute to immunosuppression in TME, where patients with immunogenic tumors and downregulated cancer metabolism are more likely to respond to NAC. Potential therapeutic vulnerabilities in immunometabolism were screened, including TME related metabolic targets that inhibit tumor growth while enhancing anti-tumor immunity, and immunometabolic targets exposed after two cycles of NAC. Breast epithelial cellular heterogeneity in metabolism was revealed through unsupervised clustering on snRNA-seq, uncovering different metabolic dependencies, whose changeable composition correlated with immune state switch and treatment response over time. The relationship of the intra-tumoral heterogeneity with the immune state differed by treatment timepoint, and metabolism drivers were associated with an accelerated growth rate of relevant subclones. Those findings are supported by our in vitro validation and previous functional research studies[11,19,61,62], demonstrating that metabolism within TME is associated with immune infiltration of tumors, and that targeting metabolism has a potential dual effect of tumor suppression and TME modulation.

A deeper analysis of the immune state subtype may help identify therapy-predictive biomarkers[63]. Although advanced techniques, such as spatially resolved transcriptomics, multiplex flow cytometry, T cell receptor abundance and cytometry by time of flight (CyTOF), offer a higher resolution in analyzing TME, they are still unavailable for routine clinical use due to high associated costs. Deconvolution algorithms can estimate the immune infiltrate composition from bulk gene expression data, achieving moderate resolution[64,65], but have generally failed to uncover the heterogeneity in immunological composition, spatial distribution, and function[63]. Here, we demonstrated that immune state (cold, warm and hot) was independent of tumor purity, and was in good accordance with other measures (i.e TILs by routine pathology assessment, protein abundance related to immune activators/suppressors). Nevertheless, patients with warm or hot tumors did not seem to present with significantly improved DFS. Though, hot tumors in this study showed higher immune gene expression and corresponded to an infiltrated-inflamed immune state subtype (with an abundance of PD-L1, CTLA-4 expression on tumor and myeloid cells and highly activated CTLs characterized by expression of Grzb, IFNγ and PD-1)[63]. Therefore, besides abundance of immune cells, measuring immune cell function in vivo is of utmost importance. Previous studies indicated that NAC could shape the immune state of breast cancer[66–68], and we found that TME had an early positive response to NAC (after two cycles) such as elevated IFN and APC signaling but residual tumors post-NAC tended to be immune suppressed. Interestingly, upregulation of the T cell exhaustion biomarker (CD244) in protein level on-treatment was demonstrated in patients who did not respond to NAC. Given the recent approval of the addition of pembrolizumab to neoadjuvant chemotherapy[3,69], this phenomenon also raises the question on window-phase of immune checkpoint inhibitors added to NAC[70]. However, it is even more complex -as indicated by final analysis of the neoadjuvant neoMono trial (NCT04770272)[71,72], demonstrating a trend towards higher pCR rate from the addition of an atezolizumab monotherapy window phase, prior to combination of chemotherapy with atezolizumab in PD-L1 positive TNBC patients.

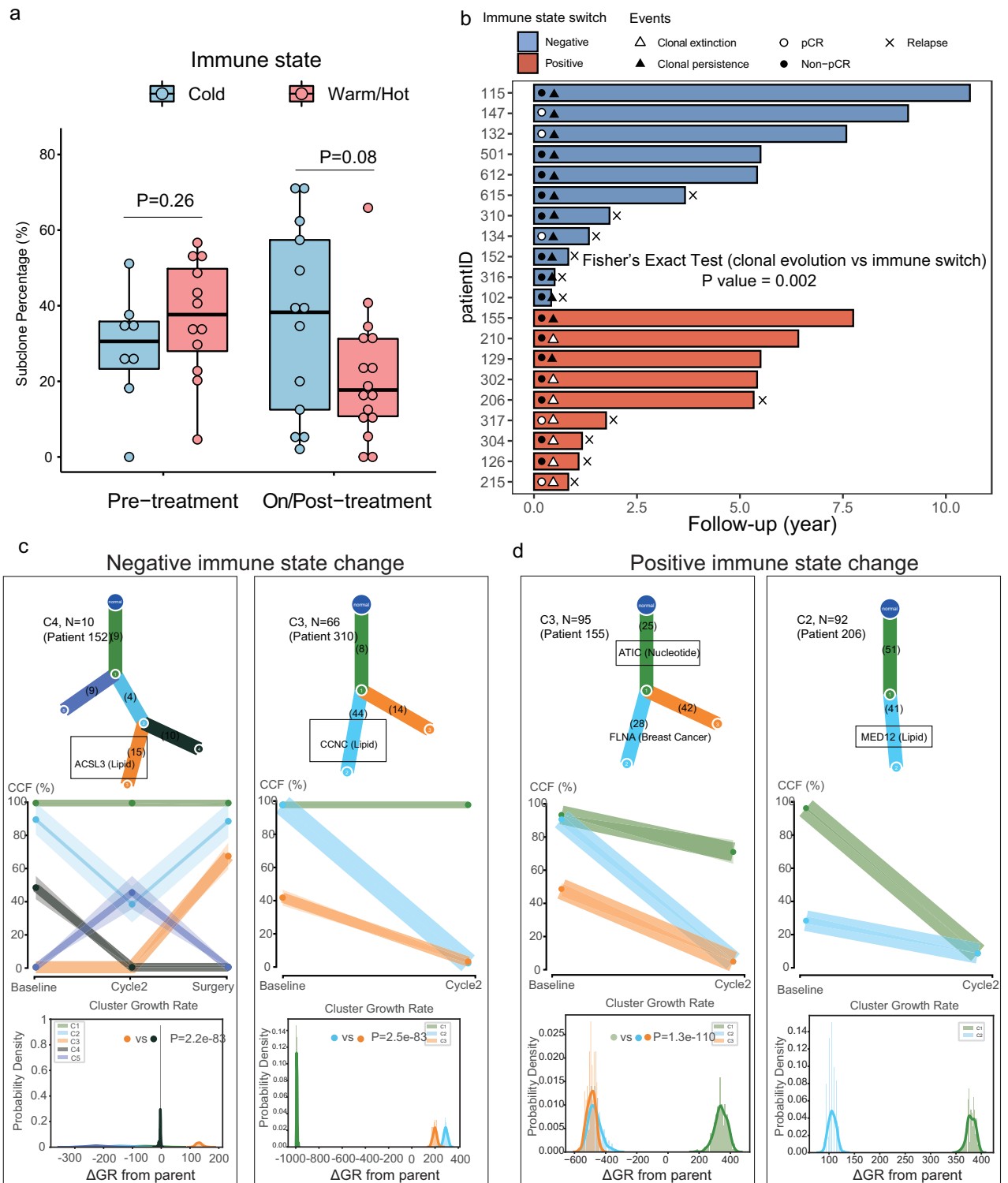

**Fig. 9 | Co-evolution of tumor clonality and immune state subtype under neoadjuvant chemotherapy. a** Subclone percentage difference (*n* = 49) between immune states by sampling timepoint. Box plot bounds the interquartile range divided by the median, with the whiskers extending to a maximum of 1.5 times the interquartile range beyond the box. Two-side Wilcoxon rank sum tests were performed. **b** Swimming chart showing the treatment results (*n* = 20), the length of each bar represents the duration of DFS of each patient in the PROMIX trial. The patients were grouped by immune state change, and a chi-squared test (*P* = 0.008) was conducted based on immune state change (negative, positive) x clonal evolution (persistence, extinction) contingency table. **c,d** Examples of subclones with

significant growth advantage relative to their parent that contain known metabolic drivers in patients with negative and positive immune state change. Results of PhylogicNDT analysis: most likely phylogenetic tree (top); permutations of sSNVs during tree construction yielding posterior CCFs of the clusters (with 95% credible intervals) (middle); and growth rates relative to parental clones (bottom). Significance of the differential growth rate (ΔGR > 0) was estimated based on the Markov Chain Monte Carlo (MCMC). Linear mixed model (two-sided *p*-values) was used to compare ΔGR between interest clones. TME tumor micro-environment, pCR pathologic complete response, DFS disease-free survival, CCF cancer cell fraction.

Efforts into metabolic subtyping have been made by previous cancer studies[23,24], although without extracting actual tumoral metabolic gene expression data. We systematically synthesized metabolic-pathway-based subtypes using tumor cell GEP and demonstrated that the tumor intrinsic metabolic subtype is a stronger predictor of immune state subtype and clinical outcomes than the bulk tumor metabolic phenotype. Upregulated bioenergetic metabolism features (i.e., TCA cycle, carbohydrate), lipid, and nucleotide-related metabolic phenotype were independently associated with lower pCR and worse DFS, confirming prior pan-cancer findings[23]. Among those important metabolic modules, we revealed that tumors with upregulated TCA cycle were more likely to be immunologically cold. By observing close protein(lipid)-protein(immune) interactions (Fig. 4d), immune effector cells (CD4/8 + T cell) from hot tumors harboring higher fat acid signature score (Fig. 7d), and also driver mutations (ACSL3 and CCNC) in lipid metabolism associated with an accelerated growth rate of relevant subclones (Fig. 9c), we could demonstrate that lipid metabolism plays a crucial role on immune cells. To infer potential metabolic competitions within the TME, we revealed metabolic dependencies of different cellular subtypes, including OXPHOS (all the cells), glycolysis (basal-like and cycling cancer cells, memory B cells, CD8 + T cells, cycling T cells, monocyte and cycling myeloid), glutathione (CAFs and endothelial cells), and cytochrome P450 (CAFs, endothelial cells, and normal breast epithelial cell). As a proof-of-concept study, Edwards et al. have previously reported that glutamine transporter inhibitor V-9302 selectively blocked glutamine uptake by TNBC cells but not CD8 + T cells, driving glutathione synthesis to improve CD8 + T cell effector function[73]. Interestingly, we identified the metabolic flexibility of TNBC epithelial cell clusters from a dynamic perspective, whose composition during NAC confers predictive implications for immune state switch and treatment outcome. One of them, epithelial cells MC2 characterized by high proliferation and multiple bioenergetic metabolism features, were more likely to present within cold tumors and harbor a powerful ability to compete with immune cells. Of note, this cellular phenotype was comparable with the metabolic subtype (MPS2) proposed by Gong et al., and inhibiting lactate dehydrogenase could improve response to anti–PD-1 therapy and increase the anti-tumor immune response within this subtype[24].

Another question that this study attempted to answer is the effect of NAC on TME. Firstly, the heterogeneity of treatment-naïve tumors was associated with immune status, which was supported by the fact that tumors with high proliferation and intra-tumoral heterogeneity exhibited a hot immune state[31]. NAC was found to be able to reshape TME by inducing immune cell infiltration and priming immune cell functions[74,75], since 40% of cold tumors changed during NAC into warm/hot status. Meanwhile, we observed that the immune state and tumor metabolic phenotype switch was consistent, and the alteration of tumor metabolism due to NAC influenced the immune state through nutrient and energy competition[76,77]. Moreover, post-treatment cold tumors with clonal persistence or expansion exhibited increased intra-tumoral heterogeneity due to pre-existing or acquired drug-resistant clones[28,78]. Importantly, our pair-wise analyses suggest the integrated immunometabolic phenotype switch from baseline to on-treatment as a novel biomarker for pCR, highlighting the importance of immunometabolism dynamics on prognosis.

In conclusion, this study demonstrates the feasibility of proteogenomic profiling in longitudinal breast cancer biopsies during NAC, systematically ascertaining unbiased biomarkers and phenotypes present on-treatment. Additionally, our findings advance the understanding of the dynamic nature of tumor-TME-metabolism interactions and suggest prognostic immunometabolic biomarkers and potential immunomodulating candidates as treatment targets. Future mechanistic experiments and clinical trials are needed to assess the mechanisms of response and efficacy of those immunometabolic targets and regimens, respectively.

## Methods

### Patients and materials

The clinical trial and patients in this study have been previously described in detail by *Kimbung* et al., summarized in Fig. 1a[79]. PROMIX trial (ClinicalTrials.gov identifier NCT00957125) enrolled patients with locally advanced (tumor size > 20 mm) HER2-negative breast cancer, who were scheduled to receive six cycles of NAC with a combination of epirubicin and docetaxel. Bevacizumab was added during cycles 3–6 for patients who did not achieve a clinical complete response (cCR) after the second cycle of NAC. Following surgery, patients received adjuvant therapy according to Swedish national guidelines and local clinical practice. Therapeutic response to NAC was evaluated by physical examination and breast imaging (mammography and ultrasound) after two, four and six courses of NAC. The study's primary endpoints were the early objective response rate and the pCR rate, which was defined as the absence of invasive cancer in the breast and lymph nodes at surgery; presence of residual non-invasive DCIS was allowed. Disease-free survival (DFS) was regarded as a secondary endpoint in this study. All patients underwent core needle biopsies at baseline and after two cycles of NAC, and post-treatment surgical specimens were also collected. Hormone receptor status was determined by immunohistochemistry (IHC) and was considered positive if ≥10% of cancer cells stained positive for estrogen receptor (ER) or progesterone receptor (PR) at the baseline core biopsy, in accordance with Swedish national guidelines.

The clinical study and correlative analyses were approved by the Ethics Committee at Karolinska University Hospital, 2007/1529–31/2 and patients provided written informed consent for their participation in the clinical trial and for translational research.

### Sample preparation for MS-based proteomics

Tumor samples for MS-proteomics were prepared as previously described[35]. The protein fraction from the Allprep kit (Qiagen) was prepared for mass spectrometry-based proteomics using a modified version of the filter assisted sample preparation method (FASP)[80]. Samples were mixed with 1 mM DTT, 8 M urea, 25 mM HEPES, pH 7.6 in a centrifugation filtering unit, 10 kDa cutoff (Nanosep® Centrifugal Devices with Omega™ Membrane, 10 k), and centrifuged for 15 min at 14.000 g, followed by another addition of the 8 M urea buffer and centrifugation. Proteins were alkylated by 55 mM IAA, in 8 M urea, 25 mM HEPES, pH 7.6 for 10 min, centrifuged, followed by two more additions and centrifugations with 8 M urea, 25 mM HEPES pH 7.6. Trypsin (Promega). 1:50, trypsin:protein, was added to the samples in 0.250 M urea, 25 mM HEPES and digested overnight at 37 °C. The filter units were centrifuged for 15 min at 14.000 g, followed by another centrifugation with MQ. Flow-through of peptides was collected and TMT10 labeled according to manufacturer's instructions (Thermo). A TMT tag with pool of all samples was used as denominator in each TMT10 to connect the different sets. TMT labeled peptides were pooled and cleaned by a strata-X-C-cartridge (Phenomenex).

### IPG-IEF of peptides

TMT labeled peptides were separated by immobilized pH gradient−isoelectric focusing (IPG-IEF) on pH 3–10 strips as described by Branca et al.[80]. Peptides were extracted from the strips by a prototype liquid handling robot, supplied by GE Healthcare Bio-Sciences AB. A plastic device with 72 wells was put onto each strip and 50 μl of MQ was added to each well. After 30 min of incubation, the liquid was transferred to a 96-well plate and the extraction was repeated two more times. The extracted peptides were dried in a speed vac for storage and dissolved in 3% acetonitrile (ACN), 0.1 % formic acid before MS analysis.

### Q exactive analysis

Before analysis on the Q Exactive (Thermo Fisher Scientific, San Jose, CA, USA), peptides were separated using an Ultimate 3000 RSLCnano

system. Samples were trapped on an Acclaim PepMap nanotrap column (C18, 3 μm, 100 Å, 75 μm x 20 mm), and separated on an Acclaim PepMap RSLC column (C18, 2 μm, 100 Å, 75 μm x 50 cm), (Thermo Scientific). Peptides were separated using a gradient of A (5% DMSO, 0.1% FA) and B (90% ACN, 5% DMSO, 0.1% FA), ranging from 6–37% B in 30–90 min (depending on IPG-IEF fraction complexity) with a flow of 0.25 μl/min. The Q Exactive was operated in a data-dependent manner, selecting the top 10 precursors for fragmentation by HCD. The survey scan was performed at 70,000 resolution from 400–1600 m/z, with a max injection time of 100 ms and a target of $1 \times 10^6$ ions. For the generation of HCD fragmentation spectra, a max ion injection time of 140 ms and AGC of $1 \times 10^5$ were used before fragmentation at 30% normalized collision energy, 35,000 resolution. Precursors were isolated with a width of 2 m/z and put on the exclusion list for 70 s. Single and unassigned charge states were rejected from precursor selection.

### Peptide and protein identification

Orbitrap raw MS/MS files were converted to mzML format using msConvert from the ProteoWizard tool suite. Spectra were then searched using MSGF+ (2020.03.14) and Percolator (v3.04.0), where search results from 8 subsequent IPG-IEF fractions were grouped for Percolator target/decoy analysis. All searches were performed against the human protein subset of Ensembl 103 in Nextflow (v20.01.0). MSGF+ settings included precursor mass tolerance of 10 ppm, fully tryptic peptides, maximum peptide length of 50 amino acids and a maximum charge of 6. Fixed modifications were TMT6plex on lysines and peptide N-termini, and carbamidomethylation on cysteine residues; a variable modification was used for oxidation on methionine residues. Quantification of TMT6plex reporter ions was done using OpenMS project's IsobaricAnalyzer (v2.5). PSMs found at 1% false discovery rate (FDR) were used to infer gene identities. Protein false discovery rates were calculated using the picked-FDR method using gene symbols as protein groups and limited to 1% FDR.

Protein quantification by TMT10plex reporter ions was calculated using TMT PSM ratios to the tissue sample pool and each tumor was normalized to its median ratio. The median PSM TMT reporter ratio from peptides unique to a gene symbol was used for quantification. The normalized protein $\log_2$ ratios are denoted as protein abundance or protein expression levels in figures and text. The output from all quantitative MS experiments is available in Supplementary Data. 4. The mass spectrometry proteomics data have been deposited to the ProteomeXchange Consortium via the JPOST partner repository with the data set identifier PXD039529 (URL: https://repository.jpostdb.org/entry/JPST001987).

### Tumor-Infiltrating Lymphocytes (TILs) score and cellularity

Hematoxylin-eosin (H&E) slides of 4 μm thickness were prepared from formalin-fixed paraffin-embedded (FFPE) tissue blocks. Two pathologists performed the blinded evaluations of cellularity and TIL score, and discordant cases were reviewed in order consensus to be reached. TIL score was defined as the estimated proportion of area with TIL infiltration within the tumor and adjacent stroma, which was categorized as low (<10%), intermediate (10–50%), and high (>50%) based on international consensus guidelines[81,82]. Tumor cellularity was estimated as the percentage of tumor cells among all cells (tumor cells, lymphocytes, stromal cells). Tumor purity was also computationally inferred using four different tools, namely ESTIMATE[83] THetA[84], Control-FREEC[85], and PureCN[86].

### Multiplex Fluorescent Immunohistochemistry (mfIHC)/Immunofluorescence (mIF) and multispectral image analysis

Whole tissue FFPE sections (4-μm thickness) were prepared and stained for mfIHC using the Leica Bond RX™ (Leica Biosystems, Buffalo Grove, IL, USA) autostainer. The 7-color IHC kit (Opal™ 7 Solid Tumor Immunology Kit, Akoya Biosciences, Malborough, MA, USA) was

modified to include the following immune (lymphocyte & macrophage) markers: CD4, CD8a, CD163, CD20, FoxP3, CD68, cytokeratin (Supplementary Table. 7). 4′,6-diamidino-2 phenylindole (DAPI) was used for nuclei staining and the tissue sections were subsequently mounted with the Prolong Diamond Antifade Mountant (ThermoFisher, Waltham, MA, USA). A detailed list of antibodies and experimental conditions is provided in Supplementary Table. 7. Whole-slide image acquisition was performed using the Vectra® Polaris™ Automated Quantitative Pathology Imaging System (Akoya Biosciences) by scanning multiple areas of the same tissue biopsy at 10x magnification. Algorithm training for tissue (tumor, stroma, blank) and cell segmentation, thresholding and image analysis were performed using the Phenochart® and inForm® image analysis software (Akoya Biosciences) as previously described[87–90]. Tissue curation was then performed to exclude staining artifacts, necrotic areas and/or intraglandular structures by a trained researcher (IZ) and reviewed by a certified pathologist (Ar.M). Spectral unmixing and cut-offs for marker positivity were applied as previously described[87] to obtain the cell densities for each marker (number of positive cells normalized to tissue area).

Regarding the NK cell panel (Supplementary Table. 8), the FFPE tissue sections were treated with antigen retrieval using the 8–10 ml citrate acid-based antigen unmasking solution (Vector Laboratories, Oxfordshire, UK) at the boiling status followed by PBS washing. Then the blocking process was conducted with 5% goat serum (Agilent Dako, Santa Clara, CA, USA) in PBS. The tissue sections were subsequently stained with the primary antibody CD3 (Abcam, Cambridge, UK) at 1:500 dilution for 1 h followed by AF594-conjugated goat anti-rabbit secondary antibody at the ratio of 1:5000 (Thermofisher Scientific, Waltham, MA, USA) staining for 30 min. Next, all the other antibodies with direct fluorophore conjugation including CD56 (AF532) (Biotechne, Minneapolis, USA), CD57(APC) (Biolegend, San Diego, USA), FcεRγ (FITC) (Merck Millipore, Darmstadt, Germany) NKG2C(PE) (Miltenyi Biotech, Bergisch Gladbach, Germany) and PanCK (AF405) (Bio-techne, Minneapolis, USA) were used to stain the slides for 1 h without light exposure at room temperature. Following the antibodies staining, the nucleus dye Hochest (Thermofisher Scientific, Waltham, MA, USA) was used to stain the slides, PBS washing was performed between the abovementioned sequential steps. The images were acquired by AIR+ Confocal Microscope (Nikon, Tokyo, Japan). We used Image J (Fiji ImageJ 2.9.0) to achieve the calculations of multiple fluorescent intensities for each cell and extract XY coordinates, followed by counting for different immune cells in the slides.

### Microarray-based bulk GEP

RNA was extracted from serial biopsies at baseline and cycle2 and surgical specimens (275 samples from 141 patients) and was then profiled on Illumina Human HT-12 v4.0 Expression BeadChip (Illumina Inc., San Diego, CA), as described previously (GSE87455)[79]. We calculated breast cancer PAM50 subtype for each sample using Genefu package[91], taking the official centroids with traditional scaling of the gene expressions as input.

### Immune state modeling

To identify the immune state for each sample, we performed immunogenomic analyses by utilizing GEP across the three time points. Considering the heterogeneity of TME in BC[92], we jointly included immune cell fraction and immune-related signatures to establish the immune classification. QuanTIseq (1.6.0)[93] wrapped in immunedeconv[65] (2.1.0) was employed estimate the relative proportion of ten primary immune cells types (B cells, CD8 T cells, CD4 T cells, Treg cells, NK cells, monocytes, M1 macrophages, M2 macrophages, mast cells, dendritic cells, and neutrophils) from bulk GEP. We used the TCGA PanCancerAtlas (https://cri-iatlas.org/) in-house immune gene set[31,33] (Supplementary Data. 11) kindly provided by Dr. David L Gibbs. The gene signature score was calculated through ssGSEA[94], and

representative immune gene signatures within six distinct TME components including adaptive/innate immune cells, tumor stroma, cytokine, tumor-associated antigens, and immune checkpoints, were extracted. A joint latent variable model[95] involving immune cell fraction and immune signatures was fitted to cluster immune state, whose optimal number of clusters was determined based on the Bayesian information criterion (n.lambda = 233, cpus = 4). The immune score calculated by the ESTIMATE[83] and the pan-cancer immune subtype (leukocyte infiltration, macrophages, TGF-beta, IFN-gamma, Wound healing)[31] deconvoluted by the XGBoost classifier, were acquired to evaluate coherence with our immune state. Moreover, correlations of immune phenotypes with pathologic features (cellularity and TIL score) were analyzed to evaluate the robustness of the computational immune state.

## Deconvolution of tumor metabolic phenotype

Due to the complexity of metabolic phenotypes and given that bulk GEP measurements are not distinguishing tumor cells from tumor-adjacent cells[20,96], we constructed a computational pipeline (https://github.com/WangKang-Leo/PureMeta) for the deconvolution of metabolic subtypes based on the tumor cell-based expression of seven metabolic pathways (as shown in Fig. 3a). In the first step, the tumor cells' mRNA abundance, as well as tumor purity, was concurrently estimated using the R package ISOpureR (1.1.3)[96], and GEP of surgical samples (n = 5) without tumor cells (T0, N0) were regarded as reference. Tumor- cells' mRNA profiles were calculated using:

$$B = p \times t + (1 - p) \times s, \qquad (1)$$

where B is the bulk GEP, t is ISOpureR estimated tumor cells' mRNA abundance we want to estimate, p is ISOpureR's estimate of the proportion of tumor and s is the tumor-adjacent cells' mRNA abundance. Bulk/tumor cells' GEPs were normalized across samples by Z scores to obtain a rank within ~16,000 coding genes in each sample. T-Distributed stochastic neighbor embedding (t-SNE)[97] was applied on tumor cell and tumor-adjacent cell mRNA profiles of curated metabolic pathway-based gene sets derived from the Reactome annotations[98] (Supplementary Data 12). Then, as described in a previous pan-cancer study on metabolic expression subtypes[23], we conducted GSEA pre-ranked analysis based on the gene set of each specific metabolic pathway, including amino acid metabolism, carbohydrate metabolism, integration of energy, lipid metabolism, nucleotide metabolism, tricarboxylic acid cycle (TCA cycle) and vitamin &cofactor metabolism. The following criteria determined the bulk/tumor cells GEP-based subtype on each metabolic pathway: (1) Upregulated subtype: Samples with FDR < 0.1 and higher (positive) Z scores; (2) Downregulated subtype: Samples with FDR < 0.1 and lower (negative) Z scores; (3) Neutral subtype: Samples with FDR > 0.1. Lastly, each tumor sample was labeled with seven bulk/tumor cell GEP-based metabolic subtypes, which was further confirmed by differential metabolic gene analyses with one-way analysis of variance (ANOVA) and Kolmogorov-Smirnov tests.

## Immunometabolic protein network

We assessed mRNA-protein, mRNA-mRNA and protein-protein pair-wise correlations using the Pearson correlation test, and immunometabolic gene/protein was annotated by a gene list that we used for immunometabolic subtypes. Protein core complex information was downloaded from CORUM 3.0 released on 03/09/2018 (http://mips.helmholtz-muenchen.de/corum/#download), and another database of protein-protein interactions was downloaded from BioGrid (BIOGRID-4.4.201, https://downloads.thebiogrid.org/BioGRID). Genes (n = 9547) in the above two databases that were found in both GEP and proteomic data were extracted for the following analyses: immunometabolic protein correlation network was generated by calculating the Pearson

correlation matrix based on immunometabolic as well as PAM50 proteins (n = 2837 in total) and filtering away protein pairs with weak correlations (absolute Pearson's r < 0.3). Then, data was exported into Gephi 9.2, which removed outlier nodes (KCore < 2) and employed the ForceAtlas2 algorithm (Scaling 1, Gravity 5, PreventOverlap) to layout network structures. To visualize protein levels across immune states in the correlation network, the average protein level was calculated for each immunologically cold, warm, and hot group.

The key proteins/genes were extracted from metabolic pathways (https://metabolicatlas.org/), whose magnitudes in mRNA and protein levels were compared between hot and cold tumors. The gene list with FDA-approved drug targets was downloaded from ProteinAtlas (http://www.proteinatlas.org/humanproteome/druggable).

## Longitudinal Gene Expression Profiling (GEP) and pathway analysis

To systematically discover biological clusters changed during NAC, firstly, we collected pair-wise DGEs across three sampling timepoints (i.e., on- vs. pre-treatment; post- vs. pre-treatment; post- vs. on-treatment). Accordingly, the linear mixed-effects models (LMEM)[99] were respectively fitted, adjusting for the BC subtype (ER+ or triple-negative (TN)) and tumor purity derived from ESTIMATE[83]. Then, the aggregated list of DGEs with false discovery rate (FDR)[100] <0.05 and absolute log 2-fold change (FC) >0.5, was collected to detect unsupervised consensus clusters on their GEP[101]. Then, genes that belong to each cluster were fed to conduct gene set enrichment analysis (GSEA)[102] based on the MSigDB[103] resource, including the HALLMARK, KEGG, and ImmuneSigDB gene sets. Additionally, the GSVA algorithm[104] was employed to quantify the enriched pathways score (PS) for each sample. Lastly, due to non-independence distributions of the PS longitudinally calculated, the LMEM was fitted for differential GSVA score analyses, which also involved three pair-wise comparisons like DGE analyses, respectively. More specifically, we constructed the model with individual PS (y) as the response, tissues collection time (T) as the predictor, and potential confounding factors such as tumor purity (P) and breast cancer subtype (S) as follows:

$$y_{ij} \sim \beta 0 + \beta 1 P_{ij} + \beta 2 S_{ij} + \beta 3 T_{ij} + \gamma_j + \in_{ij} \qquad (2)$$

where samples (i) were contributed by the patient (j), β0 is the overall intercept, β1 is the purity effect on PS, β2 estimates GEP due to IHC subtype, β3 describes the treatment time effects on GEP, γ_j is the intercept that is allowed to vary across patients (random effect term), and ∈_ij is the residual variations, respectively. To identify the significantly changed PS over time during NAC, we inferred the statistical relevance between the time (T) covariate and the LMEM goodness-of-fit using likelihood ratio tests. A differentially expressed pathway mapped to the DEG clusters was defined based on GSEA (FDR < 0.1) and differential GSVA score (FDR < 0.1).

## Single-nucleus RNA-seq (snRNA-seq) data analysis

snRNA-seq data (NCBI Sequence Read Archive under accession: SRP114962, URL: https://www.ebi.ac.uk/ena/browser/view/PRJNA 396019) were acquired for 8 TNBC patients with 16 samples from the PROMIX trial, as previously described[28]. The log-transformed TPM (Transcript per million) (log(TPM/10 + 1)) value was used to conduct downstream analyses, which was filtered to include genes that were expressed in at least 30% of cells. The data of 16 libraries were integrated using "Harmony"[105] from Seurat (4.0.1)[106] according to sample ID.

Variable genes were detected through a mean-variance inspection, and the top principal components were identified using an elbow plot and used for the UMAP dimensionality reduction[107]. To extract the immune, stromal, normal epithelial and tumor compartments from cluster identification, an integrated two-step based strategy was applied: (1) CopyKAT[108] was employed to distinguish normal cell types

(immune, stromal, normal epithelial cells) in the TME from malignant cells; (2) mean expression of the following gene markers for each cluster was calculated: (a) immune compartment: PTPRC, (b) stromal compartment: FN1, COL1A1, ACTA2, (c) tumor compartment: EPCAM, (d) normal epithelial cells: ACTA2, KRT18, KRT19. Subsequently, the cluster identity was determined at a mean expression cutoff >0.8. (3) Immune and stromal cell subpopulations were further defined based on the following gene markers: (a) plasma cell: SDC1, (b) B cell: MS4A1, CD79A, (c) CD8 + T cell: CD8A/B, CD40LG, CD3E, (d) CD4 + T cell: CD4, CD40LG, CD3E, (e) myeloid cell: CD14, CSF1R, LIRA4, (f) macrophage M1: CD68, (g) macrophage M2, CD163, (h) dendritic cell: IRF7, CXCR3, (i) endothelial cell: PECAM1, (j) adipocyte cell: ADIPOR1/2, (k) fibroblast cell: FN1, DCN, C1R, PDGFRA, OGN.

To quantify cellular immunometabolism under NAC treatment, we applied different bioinformatic strategies. Unsupervised clustering of all metabolism-associated genes was conducted for the immune/stromal and tumor/normal compartment, respectively, using the K-nearest neighbor (KNN) graph (No. of neighbors = 20) based on Euclidean distance in "harmony" space. Then, We employed the FindALLMarkers (from Seurat 4.0.1) function with the Kruskal-Wallis test to perform a DGE analysis of different cellular metabolic states or immune state switch groups (min.pct = 0.25, logfc.threshold = 0.5). Likewise, we also calculated single cell-based GSVA score as bulk GEP, and differential expression on GSVA score across groups was identified. Single-cell trajectory analysis within metabolic epithelial cell clusters was performed using Monocle3 (1.3.4)[109], which uses an algorithm to learn the sequence of gene expression changes each cell. Lastly, the computational framework for characterizing metabolism using single-cell RNA-seq (scRNA-seq) data, developed by *Xiao* et al. (https://github.com/LocasaleLab/Single-Cell-Metabolic-Landscape)[22], was employed to evaluate metabolism of TME using an external validation scRNA-seq dataset.

## Whole-exome Sequencing (WES) data preprocessing

The bulk WES data (100 bp paired-end on the HiSeq2000/HiSeq4000 Illumina systems) of 20 TNBC patients in the PROMIX trial, including 20 normal blood and 49 paired tumor samples collected longitudinally, were generated as previously described (NCBI Sequence Read Archive under accession: SRP114962, URL: https://www.ebi.ac.uk/ena/browser/view/PRJNA396019)[28]. In this study, we have re-analyzed the WES data using the nf-core/sarek[110] (v.2.7.1), an analysis pipeline built using Nextflow[111]. The nf-core/sarek pipeline follows the GATK Best-Practices[112,113] for data preprocessing. Briefly, sequence reads (FASTQ) were aligned to the GRCh38 (hg38) reference genome with BWA-MEM[114], followed by duplicate marking and base quality score recalibration (BQSR)[112]. A wide range of quality control metrics was generated by several tools (FastQC[115], QualiMap[116], BCFtools[117], Samtools[118], and VCFtools[119]), whose output was integrated and visualized by MultiQC[120].

## Somatic SNV/indel calling

Somatic short mutation calling, including SNVs and Indels, was performed in matched tumor-normal mode using GATK4 Mutect2[121] and Strelka2[122]. To capture recurrent technical artifacts, the provided by GATK panel of normal (1000g_pon.hg38.vcf.gz) was also used as input in Mutect2. Then, all variants were further annotated for potential functional effects with VEP[123]. To reduce potential false positive variants, especially for post-treatment tumor samples with low purity, we have filtered out mutations detected by Mutect2 and Strelka2 following a tailored approach. Specifically, a somatic SNV was considered a positive call if it met the following stringent criteria: (1) Variant allele frequency (VAF): An SNV was called by both Mutect2 and Strelka2, with a VAF >2%, or by any caller independently with a VAF >5%. Any SNV from the aggregated list with VAF >2% was also identified if it appeared across at least two time points. (2) Total read depth and VAF in normal blood: the depth in all positive SNVs needed to be >30 reads, and the

VAF in matched normal samples needed to be <1% and the number of reads <5. (3) Population-specific variants: The selected SNVs were not in the list of population-based databases (i.e., dbSNP[124] and 1000G[125], and frequency exceeds 1%). As for the positive candidate Indels, besides the criteria mentioned above for SNVs, a total read depth requirement of >50 was set.

## Somatic Copy Number Alteration (SCNA) calling

CNVkit (0.9.6)[126] was used for somatic copy number alteration (SCNA) calling. Specifically, the circular binary segmentation (CBS) algorithm[127] was employed to infer copy number segments, and sample- and assay-specific systematic noises were removed by adjusting GC content and utilizing the normal reference samples. To estimate normal-cell DNA contamination, we used different tools, including THetA2[84], Control-FREEC[85], and PureCN[86], and the computationally-inferred-tumor purity was compared with the pathology-based tumor cellularity. PureCN showed the highest concordance and distribution of purity values among the tested tools and was thus selected and used for subsequent analyses. In addition, PureCN predicted the optimal ploidy of each sample in the likelihood score-based grid search. Finally, we performed allele-specific copy number analyses by combining the normalized coverage with SNP allele frequencies, which was also used to calculate the major and minor allele-specific integer copy numbers. Genomic Identification of Significant Targets in Cancer (GISTIC2.0) algorithm[128] was used to identify significantly amplified or deleted focal-level events (*q* values < 0.25, Amplification Threshold = 0.1, Deletion Threshold = −0.1, Broad Length Cutoff = 0.98, Arm Level Peel-Off = 1).

## Subclone deconvolution

Intratumoral heterogeneity, appearing as the characteristic of clonal architecture, enables tumors to adapt and acquire treatment resistance[129]. Herein, we inferred the clonality of a single sample based on cancer cell fraction (CCF, represented as a probability density distribution ∈ [0, 1]), which described the proportions of cancer cells harboring a mutation ($N_{mut}$). CCF was calculated as described previously[130] (3), and clonality was defined as a confidence interval of CCF overlapping 1.

$$CCF_{Nmut} = VAF \frac{1}{p} \left[ pCN_t + CN_n(1-p) \right] \qquad (3)$$

where p denotes the tumor purity, $CN_t$ is the tumor locus-specific copy number, and $CN_n$ is the normal locus specific copy number ($CN_n = 2$).

## Phylogenetic analysis

Since the distribution of CCF represented the independent estimates for somatic events, a series of somatic mutations were often shared by the same population of cancer cells/subclone. The multidimensional Dirichlet clustering algorithm (PhylogicNDT Clustering[58,131]) was employed to generate improved partitioning of the CCF and learn the underlying clonal structure via a Markov Chain Monte Carlo (MCMC) method from the data across multiple samples. Then, we inferred the phylogenetic trees within each patient using the BuildTree component of PhylogicNDT. The generated posterior distributions on cluster positions and mutation membership were used to calculate the ensemble of possible trees that supported the phylogenetic relationship of the detected cell populations. Lastly, we modeled subclonal growth rates within MCMC (PhylogicNDT GrowthKinetics). Growth acceleration was identified by comparing the growth rate to its parent and was considered significant if the growth rate was higher than that of the parent in >95% of MCMC iterations. Driver genes derived from The Cancer Gene Census (CGC) (https://cancer.sanger.ac.uk/census), and breast cancer-related or metabolic drivers were underlined on phylogenetic trees. All of the MCMCs were run with 10,000 iterations and burned-in 1000 samples.

## External validation

A series of HER2-negative breast cancer cohorts were obtained to validate findings from this study externally (see detailed in Supplementary Table. 2), including Oslo2 cohort[35] (42 patients with both RNA-seq and MS-based proteomics), South Korean NAC cohort[67] (86 patients treated with NAC, with longitudinal RNA-seq data (pre/on/post-NAC, $n = 144$)), and a scRNA-seq cohort[41] (a total of 21 untreated patients with single-cell RNA-seq). All samples from Oslo2 and scRNA-seq cohorts were previously untreated primary breast cancers, and the South Korean NAC cohort included both pre-treatment biopsies and post-treatment samples.

DepMap 21Q2 (Cancer dependency map) that facilitates the prioritization of therapeutic targets[132] and contains a score expressing how vital a particular gene is in terms of how lethal the knockout/knockdown of that gene is on a target cell line, was applied to assess biomarkers identified in this study. For instance, a highly negative dependency score implies that a cell line depends highly on that gene. We explored cancer genes in metabolism within DepMap for pan-cancer cell lines. In addition, PRISM (Profiling Relative Inhibition Simultaneously in Mixtures), which has demonstrated the feasibility of new approaches for pin-pointing small molecule sensitivities at large-scale[133], was used to identify potential anti-cancer, non-oncology drugs in metabolism.

## Cell lines

For the experiments we used four breast cancer cell lines, MDA-MB-231 (ATCC, HTB-26), BT-549 (ATCC, HTB-122), MCF7 (ATCC, HTB-22) and T47D (ATCC, HTB-133). MDA-MB-231 was cultured in DMEM (Thermo Fisher Scientific, 31966-047) supplemented with 10% FBS (Thermo Scientific, A3382001) and 1% NEAA (Thermo Fisher Scientific, 11140-050). MCF7 were cultured in MEM (Thermo Fisher Scientific, 11-095-080) supplemented with 10% FBS and 1% NEAA. BT-549 and MCF7 were cultured in RPMI (Thermo Scientific, 88365) supplemented with 10% FBS. 1% penicillin/streptomycin (Thermo Fisher Scientific, 15-140-122 was added to all media).

## Cell survival assays

Cell proliferation was assessed using the XTT assay (CyQUANT™ XTT Cell Viability Assay, ThermoFisher Scientific, X12223) following the manufacturer's guidelines. Cells were seeded at a density of 2000 cells per well in 96well plates and they were subsequently reverse transfected with siRNAs for 120 h. A 24-h treatment with 2 µM Staurosporine (Tocris, 1285) was used as a positive control. Following the addition of the diluted XTT substrate, the plates were incubated for 4 h at 37 °C and the emitted fluorescence was measured with a microplate reader (Tecan Infinite M1000 Pro) as the absorbance subtraction of 450 nm–660 nm (background signal).

## Cell apoptotic assays

Cell apoptosis was measured with the Invitrogen™ CellEvent™ Caspase-3/7 assay (ThermoFisher Scientific, C10423). Cells were seeded at a density of 2000 cells per well in 96well plates and they were subsequently reverse transfected with siRNAs for 72 h. A 24-h treatment with 2 µM Staurosporine (Tocris, 1285) was used as a positive control. Following the treatment, the cells were incubated at 37oC for 1 h with 5 µM of the Caspase 3/7 reagent. They were subsequently fixed using 4% formaldehyde (Sigma-Aldrich, F8775) for 10 min at room temperature and counterstained with Hoechst (Thermo Fisher Scientific, 62249). Images were acquired using an IN Cell Analyzer 2200 (GE Healthcare) and analysed using Cell Profiler and Fiji (Image J).

## Gene silencing

Commercially available SMARTpool ON-TARGET oligonucleotides targeting human uL18 (RPL5) (Catalog no. L-013611), ALDOA (Catalog no. L-010376), TPI1 (Catalog no. L-009776), and non-targeting siRNA (Catalog no. D-001810) were purchased by Horizon Discovery (Dharmacon). Cells were reverse transfected with 20 nM siRNA using Lipofectamine RNAiMAX reagent (Thermo fisher scientific, 13778150) according to the manufacturer's instructions.

## Tumor and T cell coculture

MDA-MB-231, MCF7, T47D, and BT549 (authenticated from ATCC) were labeled with red fluorescent CellTracker (5 µM, Invitrogen). For analysis of tumor cell killing. Tumor cells were plated at a concentration of $1 \times 10^4$ cells per well in 96-well flat u-bottom plates and incubated with the Caspase 3/7 green dye (Essen BioScience) to detect killing. The following day, CD3 bead-isolated T cells were added at a 2:1 ratio to the target cells, or 20:1 as a positive control. The number of killed target cells was monitored by 2-hourly fluorescence imaging over 24 hours using an IncuCyte Live Cell Analysis System (Essen BioScience). Dead cell frequency was quantified using IncuCyte software (Essen BioScience) and normalized to the number of dead cells remaining in the target cell-only control group and related to maximum killing (% killing = (overlap counts-red counts) / (overlap positive-red count) x 100). Alternatively, tumor cell confluence was calculated by red dye count of tumor and T cell coculture–tumor cell-only control.

## Databases

The following databases were used for gene (signatures), protein or cell line references: Immunological metagene signatures (https://criiatlas.org/), CORUM-3.0 (http://mips.helmholtz-muenchen.de/corum), BIOGRID-4.4.201 (https://downloads.thebiogrid.org/BioGRID), Cancer Gene Census (CGC) (https://cancer.sanger.ac.uk/census), ProteinAtlas (http://www.proteinatlas.org/humanproteome/druggable), and DepMap 21Q2 (https://depmap.org/portal/).

## Statistical analysis

The interaction effect between immune (I) and metabolic (M) phenotypes was assessed by prior LMEM (2) as the following equation:

$$I_{ij} \sim \beta 0 + \beta 1\, M_{ij} + \beta 2 S_{ij}( + \beta 3\, C_{ij}) + \gamma_j + \in_{ij} \qquad (4)$$

The tumor cellularity ($C_{ij}$) was adjusted when we ran LMEM using Korean validation cohort, where tumor-derived GEP was not available. A two-way ANOVA with interaction was used to test the metabolic GSVA score profile based on immune state switch (negative and positive) and treatment time (pre-treatment and on/post-treatment). Logistic regression with adjustment for confounder variables was applied to evaluate the relationship between immunometabolic phenotype and pCR status (pCR vs. residual disease), where odds ratio (OR) with a 95% confidence interval (CI) was calculated between groups. Survival curves were constructed using the Kaplan-Meier method and compared with the log-rank test. Multivariable Cox proportional hazard modeling was applied to estimate hazard ratios (HRs) and 95% CIs. Meanwhile, we integrated the Least Absolute Shrinkage and Selection Operator (LASSO) regression and bootstrapping algorithm (iteration = 10,000, nfold = 5) to find the best prognostic feature[134,135]. The time to event for DFS was defined as the days from breast cancer diagnosis until any relapses or death. Censoring time for DFS was defined as the time from inclusion to study until last follow-up occurring on or before 31 December 2020 for participants without events. Rank correlation analysis for immunometabolic phenotypes was conducted with Spearman's correlation.

$P$-values reported (two-sided) <0.05 were considered statistically significant. FDR was used for multiple correction testing. All analyses were performed using R (version 4.0.1).

## Reporting summary

Further information on research design is available in the Nature Portfolio Reporting Summary linked to this article.

## Data availability

The mass spectrometry proteomics raw data generated in this study have been deposited in the ProteomeXchange Consortium under accession number PXD039529 (URL: https://repository.jpostdb.org/entry/JPST001987). Source data can be downloaded from Figshare (https://doi.org/10.6084/m9.figshare.22687246). Source data are provided with this paper.

## Code availability

The code used to determine tumor cell-based metabolic phenotype (i.e. downregulated, neutral, upregulated) is available at https://github.com/WangKang-Leo/PureMeta. Codes are also archived at Zenodo (https://doi.org/10.5281/zenodo.10864368 (ref. 136)).

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

## Acknowledgements

This study was supported by grants from Region Stockholm, Karolinska Institutet including Cancer Research KI, the Swedish Research Council (2021-03061), the Swedish Cancer Society (21 1800 Pj 01 H), the Research Funds at Radiumhemmet (204063) and the Swedish Breast Cancer Association (Bröstcancerförbundet) to Theodoros Foukakis. Ioannis Zerdes is supported by the Region Stockholm (clinical post-doctorial appointment, FoUI-977295), the Swedish Society of Oncology postdoctoral grant, the Swedish Society of Medicine and the Iris, Stig och Gerry Castenbäcks foundation. Alexios Matikas is supported by the Swedish Cancer Society (Cancerfonden), The Swedish Breast Cancer Association (Bröstcancerförbundet), and The Research Funds at Radiumhemmet. Bulk tumors gene expression profiling in the PROMIX trial was financed by a grant from BioCARE to Niklas Loman. The funding sources had no role in the data analysis, interpretation, writing of the article, or decision to submit the article. The computations were enabled by resources provided by the Swedish National Infrastructure for Computing (SNIC) at Bianca partially funded by the Swedish Research Council through grant agreement no. 2018-05973.

## Author contributions

K.W.: conceptualization, data curation, formal analysis, methodology, validation, visualization, writing—original draft, and writing—review and editing. I.Z.: conceptualization, data curation, formal analysis, methodology, validation, visualization, writing—original draft, and writing—review and editing. H.J.: data curation, formal analysis, methodology, visualization, writing—original draft, and writing—review and editing. YZ.S.: methodology, validation, visualization, writing—original draft, and writing—review and editing. D.K.: methodology, validation, visualization, writing—revision draft, and writing—review and editing. E.S.: data curation, formal analysis, methodology, writing—original draft, and writing—review and editing. A.M.: data curation, formal analysis, methodology, visualization, writing—original draft, and writing—review and editing. X.L.: methodology, validation, visualization, writing—revision draft, and writing—review and editing. N.L.: methodology, validation, and review and editing. I.H.: methodology, validation, visualization, writing—original draft, and writing—review & editing. D.S.: methodology, validation, visualization, writing—original draft, and writing—review and editing. J.B. (Jonas Bergh): conceptualization, data curation, formal analysis, methodology, validation, visualization, writing—original draft, and writing—review and editing. J.B. (Jiri Bartek): methodology, validation, visualization, writing—revision draft, and writing—review and editing. T.H: conceptualization, data curation, formal analysis, methodology, validation, visualization, writing—original draft, and writing—review and editing. J.L.: conceptualization, data curation, formal analysis, methodology, validation, visualization, writing—original draft, and writing—review and editing. A.M.: conceptualization, data curation, formal analysis, methodology, validation, visualization, writing—original draft, and writing—review and editing. T.F.: conceptualization, data curation, formal analysis, methodology, validation, visualization, writing—original draft, and writing—review and editing.

## Funding

## Competing interests

Jonas Bergh reports grants from Amgen, AstraZeneca, Bayer, Merck, Pfizer, Roche, and Sanofi-Aventis outside the submitted work, as well as honoraria to Asklepios Medicine HB. In addition, Jonas Bergh is co-author on a chapter on prognostic and predictive factors in early, non-metastatic breast cancer in UpToDate; Jonas Bergh has also been offered stocks in Stratipath and offered to be consultant for that diagnostic company in early development. Niklas Loman reports honoraria for educational proposes and advisory boards received by his employer from Astra Zeneca, MSD, Roche, Gilead, and MSD; Janne Lehtio reports other support from Fenomark Diagnostics outside the submitted work; Janne Lehtio is involved in Cancer Core Europe BoB trial financed by Roche (not related to this work). Alexios Matikas reports other support from Veracyte and Roche outside the submitted work. Theodoros Foukakis reports institutional grants and personal fees from Novartis; institutional grants and personal fees from Pfizer; institutional grants from AstraZeneca; personal fees from Affibody, Exact Sciences, UpToDate and Veracyte; honoraria paid to his institution from Gilead, Pfizer, AstraZeneca and Roche outside the submitted work. No disclosures were reported by the other authors.
