## [Peer Review File · Nature Communications]

Longitudinal molecular profiling elucidates immunometabolism dynamics in breast cancerREVIEWER COMMENTS

Reviewer #1 (Remarks to the Author):

The authors present a high throughput analysis of longitudinal proteogenomic data and report on the metabolic changes and metabolic reprogramming within tumor cells and tumor microenvironment (TME) in breast cancer. Overall, the experiments and analysis are conducted elegantly, data are extensive and maybe useful to the community. Their analysis shows that TCA-cycle was elevated in cold tumors but decreased in hot tumors. Pair-wise longitudinal proteogenomic analysis revealed reasonable correlation between mRNA and protein expressions. They found that clones harboring mutations of metabolic enzymes ACSL3, CCNC, ATIC, and MED12 are more likely to have accelerated growth. Authors concluded that treatment outcomes depend on intrinsic metabolic state and its interaction with the TME. Overall, this is a well written manuscript but conclusions are mostly speculative due to the lack of functional follow-up studies.

The major criticism of this work is the lack of functional aspect of the features that have been identified from this comprehensive data analysis. It summarizes different data sets without any functional follow-up studies or a clear description of the novel aspects of these finding. Several identified metabolic enzymes, their mutation and interaction with TME, are already known to play a role in breast cancers. For example, mutation of isozymes of long-chain acyl-CoA synthetase has long been known in cancer and considered as a target for cancer therapy. While the data provide a resource for the field, the manuscript, in my opinion, falls short in firmly establishing a specific role of TCA or other metabolic enzymes in breast cancer.

I think supplementing these data with some functional analysis and validation with additional analysis, such as targeted proteomics of some selected metabolic enzymes, Western blot analysis or in-vitro studies, will significantly improve the quality of the manuscript.

Other comments:

-A clear definition of the terms used such as "metabolic reprogramming", "immunometabolic interplay", and "immunometabolic vulnerabilities" would help the readership understand the rational.

-While authors claim that TCA cycle enzymes were affected in tumors (up in cold and down in hot tumors), the graph (Fig. 3b) also shows that they were not significant in TNBC tumor cells. Can authors elaborate on this as well as other metabolic phenotypes they observed (e.g., lipid, amino acid, carbohydrate, etc.)

Reviewer #2 (Remarks to the Author):

The study investigates the immunometabolic remodeling that occurs during neoadjuvant chemotherapy of breast cancer. It describes the interaction between immune states and metabolic states and their dynamics during treatment. The study is original and presents significant results that establish the interaction between metabolic pathways and immune regulation in breast cancer. It divides the tumor into three immune subtypes, that possess significant predictive and prognostic value and associate gene and protein expression as well as genetic clonality to these immune states. results provide important data on immunometabolic processes occurring during chemotherapy treatment, with potential implications on resistance to therapy.

The study is very well designed and highly comprehensive. The manuscript is well written, very clear and coherently presented. The figures are well organized and nicely clarify the results. The Methods are clearly and thoroughly described. The analysis includes appropriate statistical and computational tools, accounting for confounding factors and controlling for tumor purity and subtype. Results were validated using external datasets.

Comments:

1. What are the associations between immune states and cell proliferation/tumor aggressiveness. Is the immune state independent of the proliferation state when calculating pCR and survival (Fig2e)? Proliferation state should be considered when correlating metabolic states with immune states (i.e. adjust for proliferation in LMEM when calculating prognostic effects).
2. The authors state that: "we identified that immunologically cold TNBC was associated with

downregulated lipid pathway " and hypothesized that "lipids, as essential components of immune cells, support the development of innate and adaptive immunity ". However, the metabolic state was calculated on the tumor cells and not the immune cells. Is it possible to apply the ISOpureR or other method to calculate an "immune cell" group similar to the calculation of the "tumor cell" only (Fig3a), and extract mRNA expression for this immune cell fraction? This will enable to differentiate between metabolic states occurring in the tumor and metabolic states occurring in the immune cells, and to better establish the interaction between immune states and metabolic states or the proliferative states of the immune cells themselves.

3. It is very interesting to understand the changes occurring throughout the treatment in individual patients. Following changes in individual patients over time can identify processes that are related with resistance or response. For example, the fraction of NK CD56 at surgery is a strong prognostic signature. How this marker is changed over time in individual patients? What accompanies this change, namely can we find any association with gene expression/protein/metabolic state that regulates its modulation?

4. Can we predict response according to the TME changes from T1 to T2? What would be the predictive power of the immune modulation relative to current predictions? Can we validate a clinically-relevant biomarker based on the TME findings (a specific metabolic signature)?

Minor comments:

1. Introduction, 2nd paragraph – rephrase the sentences for better fluency
2. Figure 1: please include the number of patients and samples used for each methodology within the scheme
3. Figure 2A – include the tumor subtype for each tumor in the clinical parameters

Reviewer #3 (Remarks to the Author):

In this study, authors performed mainly a longitudinal (2-3 time points) transcriptome study in HER2-negative breast cancer patients (n = 122). They classified all samples into immune hot/warm/cold subtypes based on transcriptome, and discussed their relationship to 7 metabolic gene signatures. They showed the dynamic changes of immune subtype and metabolic gene signatures upon neoadjuvant chemotherapy (NAC). They also performed proteomics (n = 29), snRNA-seq (n = 8) and WES (n = 20) on a subset of the patients on 2-3 time points to describe these multi-omics features on the immune subtypes and metabolic clusters. Most of the results are descriptive and associations, which is typical for multi-omics studies, but this manuscript lacks clear conclusions or a significant discovery.

Main concerns:

1. Authors state that they are performing proteogenomic analysis on breast cancer, but actually most of the analyses are based on their transcriptomic data. The number of samples profiled by both proteomics and genomics are roughly 50 samples in 20 patients, which is limited for a proteogenomic study. Therefore, they as expected didn't get any notable discoveries. Proteomic data is only used in some differential analysis, and WES is used in the last part, which have little add to the paper. They may profile more proteogenomic samples to match their title and make new discoveries on larger datasets. Otherwise, the title at least should be changed.
2. Authors should more precise to describe their cohort, because they are not studying all kinds of breast cancer but only HER2- breast cancer.
3. The transcriptome in HER2- breast cancer is extensively studied, and the hot/cold immune subtypes are also well studied. Authors should pay more attention to the longitudinal and NAC features of their dataset. But they didn't raise any new points on why patient response/not response to NAC.
4. In Fig. 2c hot tumor has a lower Macrophage, especially M2, which is opposite to the signature results as shown in Fig. 2a. This brings the major concern that most results in this study are based on signature activity that maybe can't be validated by orthogonal methods.
5. Results in Fig. 3 is based on gene signatures, authors have no substantial evidence and findings to state there is a critical interplay between tumor metabolism and immune state. More analyses on single cell data may help.

6. In Fig. 3c the prognostic value of metabolic phenotypes presented is limited, some are not even significant, such as amino acid $p = 0.07$.
7. The 7 metabolic gene sets used here are broad functionality, that consists many genes, such as lipid contains 766 genes, with such many genes as a signature, how the combined effect be representative should be discussed, a more detailed and specific metabolic gene sets if exists may help.
8. More comprehensive immune subtypes have been extensively studied in various cancer types and pan-cancer studies. The three simple immune class presented in this study is outdated. More sophisticated immune subtypes may lead to more interesting discoveries. For instance, CAF status is known to be related to immune response, which is not even mentioned in this part.
9. Authors may add number of samples sequenced by each tech in Fig. 1.
10. Add breast cancer subtype etc. information in Fig.2a. column label may help.
11. There are lots of abbreviations without explanation : for instance: BC, TNBC, HR+ in Fig. 2e and pCR, DFS etc..
12. Fig. 5c TME change is not clear, can't visually spot.
13. Fig. 5c why DFS is binary?
14. Typo: of which 7,357 proteins were quantified in each of the of the 53 tumors.
15. Whether GEP and protein data have batch effect and if any how they are addressed is not discussed. Low dimensional visualization on raw and normalized data should be shown.
16. Response to NAC is the one of the unique features in the dataset. Authors may focus more on this point. Some of the analysis, such as Fig. 4f and Fig. 5e can be performed between patients who response or not response to NAC. Tumor microenvironments form snRNA-seq may also help here. Maybe authors can dig more to find specific genes that potentially related to NAC response, and test it. Such as predict genes and related drugs could potentially reverse the response pattern, for example by using cmap dataset.
17. Whether the single cell immune proportion is highly correlated with the three immune subtypes form GEP?
18. Different immune cell types do not appear to be well separated in Fig.S13. Please show Fig.S13d painting by samples. Seems to me the batch effect in single cell data is not well addressed.
19. RNA velocity can be applied to show the tumor evolution upon NAC.
20. Are there any gene mutations more represented in NAC response tumors than non-responders, and whether the pattern is the same before and on/post treatment?
21. Is CNV based clonality change the same as mutation based one.
22. Some conclusions in the manuscript may need further evidence: such as: "suggesting that drug-resistant tumors may have a higher rate of glycolysis and be capable of escaping antitumor immunity." and "The disappearance of MC5 on/post-treatment in positive immune state change group could be explained by chemokines as CXCL12/14 that are known for the recruitment of T cells, and notch signaling that regulates mature of CD4/8+ T cells were enriched in MC5"etc.

Reviewer #4 (Remarks to the Author):

Having the longitudinal tumor samples from patients in the PROMIX trial as proteomic profiling (using 4 different platforms) would be an important and rich information resource for the scientific community. However, this manuscript requires several improvements. Several specific experimental comments to address include:

General Comments:

1. Generally, the manuscript is hard to follow in some sections. The text jumps between methods, results, and discussion without clear transitions. It would benefit from a more structured format guiding the reader through the methodology, findings, and implications step by step. Simplifying the presentation of results and using more narrative to describe the significance of findings could help.
2. It's not immediately clear what the main takeaways should be. The paper could benefit from a clearer statement of the primary conclusions and their significance in each section. The paper suggests that the findings could be useful as biomarkers of treatment efficacy but doesn't elaborate on how this would work in practice.

3. One of the critical aspects that seem underdeveloped in the manuscript is the integration of the various platforms used for proteomic and transcriptomic analyses. While the use of multiple platforms is commendable and has the potential to offer a comprehensive view of the TME, the authors have not sufficiently highlighted how these methods complement each other. It is essential that the authors delineate the synergistic benefits of combining these technologies. How each contributes to a better understanding of tumor metabolism and immune interactions. Moreover, the overarching message and contributions of the study remain obscured. I suggest trying to simplify the results and highlight the importance of each section.

Major:

1. There are several groups of data and groups analyzed, which is not clearly explained in every section. For example, Section "MS-based Proteomic Landscape of Immunometabolism Phenotype and Pathways". It is confusing why there is a new set of samples, and this happens in each section. The paper needs a better illustration of the sets used for each platform, and a more complete Figure 1 with this information could help. There are so many different analyses in this paper, and it is not well summarized.
2. The description of patient demographics utilized in this study is lacking. Even if this data is derived from previously published clinical trials like the PROMIX trial, the authors should incorporate a detailed table of the clinical features, groupings of clinical variables (clinical receptor status, etc.), and groups described in this manuscript from some of the analysis, survival data, subtypes, etc., as metadata for the analysis performed. This addition is essential for reproducibility and should be clearly indicated. The scientific community will need this data to use everything detailed in the data availability section. If it is already included somewhere, please specify it better in the manuscript.
3. The section "Tumor Metabolic Phenotype Interacts with Immune State" needs clarification on the sample groups analyzed. The text alternates between bioinformatics approaches, which complicates comprehension. For instance, while introducing an integrated approach classifying seven metabolic pathways into three states, the authors fail to identify these pathways or justify their focus.
4. The hypothesis regarding lipids and immunity is not articulated clearly. The authors should revisit this hypothesis in the conclusion, detailing how the results support it.
5. A statement regarding the failure to extract TC GEP due to the absence of RNA-seq data raises concerns. The authors assert that certain metabolic-pathway-based subtypes are negatively correlated with immune states. However, without distinguishing lipid pathway subtypes from immune cells, this statement seems speculative. The use of tumor purity as a variable in the LMEM could introduce bias. Moreover, tumor purity assessment via ESTIMATE may not always be accurate in bulk RNA analysis and could lead to misinterpretations if this is used as a variable in a supervised learning analysis. The authors don't know if accounting for purity in the model is causing a boost in the tumor signal, and if that is the case, why do you want to remove the signal coming from the TME? Some tumors have less tumor cell content and higher immune cell content, but this does not always mean lower purity, especially after treatment. A pathological cut-off for high tumor cellularity may be a more accurate method for sample selection than a deconvolution method for tumor purity in this particular case.
6. In the section "Overview of transcriptome response to NAC" the authors employed a linear mixed-effects model (LMEM). Can the authors explain why they used a linear mixed-effect model for doing these analyses instead of other methods like DESeq2? What about the batch effect? Or how do they handle normal cell contamination? What is the random effect? It is not explained in the methodology. Could the authors explain why they used an LMEM instead of other methods if no random effect was used in the model (or at least I missed this info in the methodology). Please clarify.
7. Figure 4 d-e: lack of statistics. You make statements in the text for Figure 4d-3 without numbers or statistics. In the same section, the authors claim that "hot tumors associated with increased proliferation (bulk MKI67 protein) were characterized by a massive increase in glucose consumption, also known as the Warburg effect". Supplemental Fig. 8d. However, this figure did not reach significance. The authors again speculate that might be derived from immune cells. As for Figure 6e, there are no medians or statistics, so this figure is hard to interpret. Please clarify.
8. Supplemental Fig. 14a-b: Please add medians and statistics to this figure.
9. Supplemental Fig. 15a: Please explain how you calculated the subtypes. And the authors should have this subtype as supplementary data. The PAM50 subtypes are not included or explained in

the methodology (single or bulk RNA).

Minor:

10. The supplemental figures contain a significant amount of data that is not discussed in the text. It is advisable for the authors to omit any panels not integral to the manuscript's understanding, as they contribute to confusion.

11. In Figure 2A, why do hot tumors in TNBC present higher levels of HER2?

12. In Figure 2c, the author mentions an mIHC panel but is now explained which are the markers used in the text or figure legend.

13. Section "Longitudinal Pairwise Analyses on Immunometabolism" . Figure 5d. Are these groups balanced? How many samples are in each group?

RESPONSE TO REVIEWERS' COMMENTS

We thank the reviewers for the critical assessment of our manuscript and the valuable comments. We have addressed all the comments and performed revisions in the manuscript according to the suggestions. Please find hereunder a point-by-point reply to the reviewers' comments, with the comment appearing in regular text and our reply following in **bold** text. The revised manuscript version has been uploaded in two forms, one clean and one where all the changes to the manuscript are marked with the **red** words.

Please note that, at the request of reviewer #3, the title has been changed to **Longitudinal molecular profiling elucidates immunometabolism dynamics in breast cancer**.

Reviewer #1 (Remarks to the Author):

The authors present a high throughput analysis of longitudinal proteogenomic data and report on the metabolic changes and metabolic reprogramming within tumor cells and tumor microenvironment (TME) in breast cancer. Overall, the experiments and analysis are conducted elegantly, data are extensive and maybe useful to the community. Their analysis shows that TCA-cycle was elevated in cold tumors but decreased in hot tumors. Pair-wise longitudinal proteogenomic analysis revealed reasonable correlation between mRNA and protein expressions. They found that clones harboring mutations of metabolic enzymes ACSL3, CCNC, ATIC, and MED12 are more likely to have accelerated growth. Authors concluded that treatment outcomes depend on intrinsic metabolic state and its interaction with the TME. Overall, this is a well written manuscript but conclusions are mostly speculative due to the lack of functional follow-up studies.

We thank the reviewer for this careful consideration, comments, and questions. You can find our responses to specific comments hereunder.

1. The major criticism of this work is the lack of functional aspect of the features that have been identified from this comprehensive data analysis. It summarizes different data sets without any functional follow-up studies or a clear description of the novel aspects of these finding. Several identified metabolic enzymes, their mutation and interaction with TME, are already known to play a role in breast cancers. For example, mutation of isozymes of long-chain acyl-CoA synthetase has long been known in cancer and considered as a target for cancer therapy. While the data provide a resource for the field, the manuscript, in my opinion, falls short in firmly establishing a specific role of TCA or other metabolic enzymes in breast cancer. I think supplementing these data with some functional analysis and validation with additional analysis, such as targeted proteomics of some selected metabolic enzymes, Western blot analysis or in-vitro studies, will significantly improve the quality of the manuscript.

We agree that our study relied mostly on correlative analyses for the immunometabolic interplay using various multi-omics data. Therefore, we performed additional *in vitro*/functional studies in order to further validate our previously described findings. Based on the multi-omics data analyses, we demonstrated that various metabolic-related genes/proteins, were upregulated in cold tumors (i.e. *FASN*, *ALDOA*, *HMGCS1*, *ACACA*, seen in Fig. 4f) or presented within on/post-treatment tumors with negative immune state change (i.e. *RPL5*, *GAPDH*, *TPI1*, *DCXR*, seen in Fig. 6g). Among these genes, *RPL5* and *TPI1* represented also potential therapy targets based on DepMap 21Q2 CRISPR breast cancer cell line screen (Supplementary Figure 16). Therefore, we wanted to select established human breast cancer cell lines as our experimental set based on the baseline expression of these metabolic-related genes (i.e. highly expressed genes *RPL5*, *TPI1*, *ALDOA*). By using transcriptomic data (see figure below), we observed that these genes were most commonly upregulated in the following breast cancer cell lines: MDA-MB-231 & BT549 (basal-like), MCF7 & T47D (luminal). By using siRNA technology, we successfully performed knockdown of the three genes in all cell lines. Upon knockdown of the metabolic-related genes, we observed and decreased cell viability (evaluated by XTT assay) and increased apoptosis (evaluated by caspase 3/7 assay) of the tumor cells compared to the respective controls. Next, in order to evaluate any changes and/or crosstalk between tumor and immune cells, we performed co-culturing of tumor cells (siRNA-knockdown and control) with T-cells for the siRNA-knockdown cell lines. Accordingly, we observed the higher tumor cell killing percentage within metabolic gene knockdown cell lines compared with their controls. The updated results, figures (Figure 7 and Supplementary Figure19) and Methods description have been now added as following (page 17, line 19 to page 18, line 19 of the revised manuscript):

***In Vitro* Validation of Immunometabolic Targets**

Based on the aforementioned analyses of the bulk GEP, proteomic data and snRNA-seq, we demonstrated that various metabolic-related genes were upregulated in cold tumors (i.e. *FASN*, *ALDOA*, *HMGCS1*, *ACACA*, seen in Fig. 4f) or presented in on/post-treatment tumors with negative immune state change (i.e. *RPL5*, *GAPDH*, *TPI1*, *DCXR*, seen in Fig. 6g). Therefore, to further substantiate these findings and gain functional insights into the immunometabolic interplay, we performed *in vitro* studies targeting these genes. In order to establish our experimental set of HER2-negative human breast cancer cell lines, we used publicly available cell line transcriptomic data (Supplemental Fig. 19a), and observed that baseline expression of three of these metabolic-related genes (i.e. *RPL5*, *TPI1*, *ALDOA*) was most commonly upregulated in the following breast cancer cell lines: MDA-MB-231 & BT549 (basal-like subtype), MCF7 and T47D (luminal subtype). By using siRNA transient transfection technology, we successfully performed knockdown of the three genes in all 4 cell lines (Supplemental Fig. 19b). Upon knockdown of the metabolic-related genes, we observed a decreased cell viability (evaluated by XTT cytotoxicity

assay, **Fig.7a-b**) and increased apoptosis (evaluated by caspase 3/7 assay, **Fig.7c-d** and **Supplemental Fig. 19c**) of the tumor cells with the knocked-down genes compared to the control cells.

Given the importance of the anti-tumor activity of T cells in the TME to eliminate tumor cells, we next conducted co-cultures of T-cells with the aforementioned cancer cell lines (both control and gene-silenced) and assessed for tumor killing over the course of a 24-hour live cell-imaging. Upon knockdown of the metabolic-related genes, direct tumor cell killing was observed for most cell lines and target genes when cultured with T-cells, especially for *ALDOA* in both luminal and basal-like cells, and *TP11* in luminal cells (**Fig.7e**). Furthermore, tumor cell growth was arrested when cultured with T cells and upon gene silencing, as demonstrated by a decreased confluence compared to control cells (**Supplemental Fig. 19d**). Taken together, our in vitro data support the hypothesis that targeting metabolic genes may lead to immune cell-mediated tumor cell killing and tumor growth inhibition, thereby provide insights for future studies in this field.

Fig7: a,b XTT assay of two luminal (MCF7, T47D) and two basal (MDA-MB-231, BT-549) breast cancer cell lines following a. the knock-down of RPL5, ALDOA or TPI1 or b. treatment with Staurosporine. All growth assays were performed twice in sextuplicate (N = 12, knock-down) or triplicates (N=6, Staurosporine). Data are shown as mean \pm SD, ANOVA t-test (treatments compared to siCtrl or DMSO), **** p < 0.001, ** p < 0.01, * p < 0.05, ns: non-significant. The percentage of viable cells is shown inside the bards. **c** Cell apoptosis is shown as mean intensity (arbitrary units) of Caspase 3/7 following a 72-hour knockdown of the same genes in the cell lines mentioned in (a). All apoptotic assays were performed twice with a total of

1000-2000 cells counted per experimental condition. Data are shown as mean \pm SD, Students t-test (treatments compared to siCtr), **** $p < 0.001$, ns: non-significant. Scale bar = 10 μ m. **d.** Representative immunofluorescence images used for the analysis presented in (c). **e.** Tumor cell killing percentage following T-cell co-culturing in control cell lines and upon knock-down of RPL5, ALDOA or TP11 for 24hours and 2-hours time-lapse using live cell imaging. Cumulative data (n=4) are shown as mean \pm SEM, experiments were repeated twice with three technical replicates and eight biological replicates. Statistical analyses were performed using Dunnett's multiple comparisons test presenting adjusted p-values, * $p < 0.05$.

Supplementary Fig 19: **a.** Heatmap of baseline transcriptomic expression of different metabolic-related genes (columns) in various breast cancer cell lines (rows), the breast cancer cell lines selected for the *in vitro* studies are highlighted within the black-outlined boxes. **b.** Relative quantitation of *RPL5*, *ALDOA* and *TPI1* mRNA levels following their siRNA-mediated depletion in two luminal (MCF7, T47D) and two basal (MDAMB231, BT549) breast cancer cell lines. All qPCR assays were performed three independent times in triplicates (N = 9). Data are shown as mean \pm SD, Students t-test (treatments compared to siCtr), **** p < 0.001, * p < 0.05. **c.** Cell apoptosis is shown as arbitrary units of Caspase 3/7 integrated intensity (immunofluorescence) following treatment of various breast cancer cell lines with 2 μ M Staurosporine for 24 hours. The apoptotic assay was performed twice with a total of 1000-2000 cells counted for each experimental condition. Data are shown as mean \pm SD, Students t-test (treatments compared to DMSO), **** p < 0.001. **d.** Tumor cell confluence after following T-cell co-culture in control cell lines and upon knock-down of *RPL5*, *ALDOA* or *TPI1* for 24hours and 2-hours time-lapse using live cell imaging. Cumulative data (n=4) are shown as mean \pm SEM, experiments were repeated twice with three technical replicates and eight biological replicates. Statistical analyses were performed using Dunnett's multiple comparisons test presenting adjusted p-values, *p < 0.05.

Page 37, line 15 to page 39, line 9 in the revised manuscript:

Method

Cell lines

For the experiments we used four breast cancer cell lines, MDA-MB-231 (ATCC, HTB-26), BT-549 (ATCC, HTB-122), MCF7 (ATCC, HTB-22) and T47D (ATCC, HTB-133). MDA-MB-231 was cultured in DMEM (Thermo Fisher Scientific, 31966-047) supplemented with 10% FBS (Thermo Scientific, A3382001) and 1% NEAA (Thermo Fisher Scientific, 11140-050). MCF7 were cultured in MEM (Thermo Fisher Scientific, 11-095-080) supplemented with 10% FBS and 1% NEAA. BT-549 and MCF7 were cultured in RPMI (Thermo Scientific, 88365) supplemented with 10% FBS. 1% penicillin/streptomycin (Thermo Fisher Scientific, 15-140-122 was added to all media.

Cell Survival Assays

Cell proliferation was assessed using the XTT assay (CyQUANT™ XTT Cell Viability Assay, ThermoFisher Scientific, X12223) following the manufacturer's guidelines. Cells were seeded at a density of 2000 cells per well in 96well plates and they were subsequently reverse transfected with siRNAs for 120 hours. A 24-hour treatment with 2 μ M Staurosporine (Tocris, 1285) was used as a positive control. Following the addition of the diluted XTT substrate, the plates were incubated for 4 hours at 37°C and the emitted fluorescence was measured with a microplate reader (Tecan Infinite M1000 Pro) as the absorbance subtraction of 450 nm - 660 nm (background signal).

Cell Apoptotic Assays

Cell apoptosis was measured with the Invitrogen™ CellEvent™ Caspase-3/7 assay (ThermoFisher Scientific, C10423). Cells were seeded at a density of 2000 cells per

well in 96well plates and they were subsequently reverse transfected with siRNAs for 72 hours. A 24-hour treatment with 2µM Staurosporine (Tocris, 1285) was used as a positive control. Following the treatment, the cells were incubated at 37oC for 1 hour with 5µM of the Caspase 3/7 reagent. They were subsequently fixed using 4% formaldehyde (Sigma-Aldrich, F8775) for 10 min at room temperature and counterstained with Hoechst (Thermo Fisher Scientific, 62249). Images were acquired using an IN Cell Analyzer 2200 (GE Healthcare) and analysed using Cell Profiler and Fiji (Image J).

Gene silencing

Commercially available SMARTpool ON-TARGET oligonucleotides targeting human uL18 (RPL5) (L-013611), ALDOA (L-010376), TPI1 (L-009776), and non-targeting siRNA (D-001810) were purchased by Horizon Discovery (Dharmacon). Cells were reverse transfected with 20 nM siRNA using Lipofectamine RNAiMAX reagent (Thermo fisher scientific, 13778150) according to the manufacturer's instructions.

Tumor and T cell coculture

MDA-MB-231, MCF7, T47D, and BT549 (authenticated from ATCC) were labeled with red fluorescent CellTracker (5 µM, Invitrogen). For analysis of tumor cell killing. Tumor cells were plated at a concentration of 1 x 10⁴ cells per well in 96-well flat u-bottom plates and incubated with the Caspase 3/7 green dye (Essen BioScience) to detect killing. The following day, CD3 bead-isolated T cells were added at a 2:1 ratio to the target cells, or 20:1 as a positive control. The number of killed target cells was monitored by 2-hourly fluorescence imaging over 24 hours using an IncuCyte Live Cell Analysis System (Essen BioScience). Dead cell frequency was quantified using IncuCyte software (Essen BioScience) and normalized to the number of dead cells remaining in the target cell-only control group and related to maximum killing (% killing = (overlap counts-red counts) / (overlap positive-red count) x 100). Alternatively, tumor cell confluence was calculated by red dye count of tumor and T cell coculture – tumor cell-only control.

Other comments:

1. A clear definition of the terms used such as “metabolic reprogramming”, “immunometabolic interplay”, and “immunometabolic vulnerabilities” would help the readership understand the rationale.

Per the reviewer's request and in order to seamlessly integrate and convey the significance of our findings, we now use consistent and clear definitions when they first appeared in main text:

Metabolic reprogramming allows tumors to acquire metabolic properties that support cell survival, evasion of immune surveillance, and hyperplastic growth⁹, which not only meets specific demands for increased energy, biomass, redox maintenance, and cellular communication, but also interacts with the complex TME¹⁰⁻¹². **(Page 3, line 14 to 15)**

Moreover, immunometabolic interplay exists in TME, where complex metabolic

networks of tumor, stromal and immune cells are dictated by cell-intrinsic and environmental factors¹⁷. (page 3, line 18 to 20)

As for “immunometabolic vulnerabilities”, we replaced it using immunometabolic drivers. (page 2, line 12)

2. While authors claim that TCA cycle enzymes were affected in tumors (up in cold and down in hot tumors), the graph (Fig. 3b) also shows that they were not significant in TNBC tumor cells. Can authors elaborate on this as well as other metabolic phenotypes they observed (e.g., lipid, amino acid, carbohydrate, etc.)

To better present our statistical model on the interaction of tumor metabolic phenotype and immune state, we added Fig3c. Fig3b the reviewer mentioned only involved univariate comparison evaluated by Chi-Square test or Fisher’s exact test, which visualized the rough trends of the composition differences. However, we made a conclusion based on the linear mixed-effects models (LMEM) as below (page 39, line 13 to 15 in the revised manuscript):

$$I_{ij} \sim \beta_0 + \beta_1 M_{ij} + \beta_2 S_{ij} (+ \beta_3 C_{ij}) + \gamma_j + \epsilon_{ij}, \quad (4)$$

The tumor cellularity (C_{ij}) was adjusted when we ran LMEM using Korean validation cohort, where tumor-derived GEP was not available.

The model was established, adjusting for subtype (and purity) with a random term from patient. Fig3c was added to present details of model parameters and significance. Besides tumor TCA cycle, lipid, amino acid, vitamin/cofactor are significantly correlated with immune state.

$$I_{ij} \sim \beta_0 + \beta_1 M_{ij} + \beta_2 S_{ij} + \gamma_j + \epsilon_{ij}$$

Fig.3c Funkyheatmap depicting the coefficient, random term, residual variance and p value of LEME, which was conducted within all samples to identify interaction effects between immune state (I) and tumor metabolic phenotypes (M), adjusting for the breast cancer subtype (S).

In addition, we also did the same analyses in the Korean validation dataset, where tumor cell derived GEP are not available, and tumor cellularity was adjusted in model.

a

b

c

Phenotype	N of upregulated	N of downregulated	Adjusted OR	P value	Adjusted OR with 95% CI
Lipid	10	14	1.39	0.08	0.5 - 1.5
Amino acid	23	17	0.71	0.02	0.5 - 1.5
Carbohydrate	10	8	1.26	0.26	0.5 - 1.5
TCA cycle	26	17	0.99	0.97	0.5 - 1.5
Energy	4	5	0.78	0.4	0.5 - 1.5
Nucleotide	9	6	1.22	0.4	0.5 - 1.5
Vitamin	3	4	0.85	0.65	0.5 - 1.5

Supplementary Fig. 7 Bulk gene expression profiling based metabolic phenotypes interact with immune subtypes in Korean NAC cohort⁶. a Percentage stacked bar chart showed distribution of bulk gene expression based metabolic phenotype in seven pathways on different immune state, where P values were derived from Chi-Square test or Fisher's exact test. b Funkyheatmap depicting the coefficient, random term, residual variance and p value of LEME, which was conducted within all samples to identify interaction effects between immune state (I) and bulk tumor metabolic phenotypes (M), adjusting for the breast cancer subtype (S) and cellularity

(C). c Forest plot depicting association between bulk metabolic phenotype and pCR. LMEM, linear mixed-effects model; pCR, pathologic complete response.

Given that only TCA cycle phenotype was well validated (Supplementary Fig7), we indicated that upregulated tumor cell-derived TCA cycle is associated with cold tumors. The relevant paragraph now reads (page 10, line 11 to 22 in the revised manuscript): The reproducibility of the interaction between metabolic pathway-based subtype and immune state was externally validated by the expression profiles of the Korean cohort (Supplemental Fig. 7a-c). Although we failed to extract TC GEP due to lack of RNA-seq data of post-treatment samples that reached pCR, the tumor cellularity was added into LMEM (Supplemental Fig. 7b) and multivariate logistic regression. Metabolic-pathway based phenotype like TCA cycle (coefficient, -0.22; P=0.003) and nucleotide (coefficient, -0.26; P=0.03) were negatively correlated with immune states (Supplemental Fig. 7b), but upregulated vitamin/co-factors was associated with hot immune state (coefficient, 0.37; P=0.03). Moreover, patients with pre-treatment upregulated amino acid phenotype were less likely to reach pCR compared to downregulated group (multivariable-adjusted OR=0.71, 95% CI, 0.54 to 0.94, P=0.02) (Supplemental Fig. 7c).

Overall, here we uncovered a critical interplay between tumor cell GEP-based metabolic phenotype (i.e. TCA cycle) and immune state subtype, and highlighted the prognostic role of cellular metabolism.

Reviewer #2 (Remarks to the Author):

The study investigates the immunometabolic remodeling that occurs during neoadjuvant chemotherapy of breast cancer. It describes the interaction between immune states and metabolic states and their dynamics during treatment. The study is original and presents significant results that establish the interaction between metabolic pathways and immune regulation in breast cancer. It divides the tumor into three immune subtypes, that possess significant predictive and prognostic value and associate gene and protein expression as well as genetic clonality to these immune states. results provide important data on immunometabolic processes occurring during chemotherapy treatment, with potential implications on resistance to therapy.

The study is very well designed and highly comprehensive. The manuscript is well written, very clear and coherently presented. The figures are well organized and nicely clarify the results. Methods are clearly and thoroughly described. The analysis includes appropriate statistical and computational tools, accounting for confounding factors and controlling for tumor purity and subtype. Results were validated using external datasets.

Comments:

1. What are the associations between immune states and cell proliferation/tumor aggressiveness. Is the immune state independent of the proliferation state when

calculating pCR and survival (Fig2e)? Proliferation state should be considered when correlating metabolic states with immune states (i.e. adjust for proliferation in LMEM when calculating prognostic effects).

We thank the reviewer for this insightful comment and we agree that cell proliferation may correlate with immune cell states. In order to appropriately adjust for proliferation in LMEM, we first compared mRNA MKI67 expression between immune states as in updated **Supplementary Fig1.c:**

However, Kruskal-Wallis test as well as univariate logistic regression (OR, 1.21, 95% CI, 0.49 to 3.01, P value, 0.81) indicated that MKI67 did not correlate with immune state. This result was further supported in tumor IHC KI67 percentage (see Supplementary Fig9e). Therefore, we didn't add mRNA MKI67 into LMEM (Immune state ~ metabolic phenotype).

Besides, we found that mRNA MKI67 expression level at baseline was associated DFS (univariate HR, 3.52, 95% CI, 1.70 to 7.30, P value, 0.001), rather than pCR status (univariate OR, 0.95, 95% CI, 0.73 to 1.24, P value, 0.70). Accordingly, we adjusted for mRNA KI67 in the Cox regression model for the DFS endpoint as shown in the Fig.2e below, added also in updated manuscript.

The following text was added to the revised manuscript (page 7, lines 10 to 12): “Regarding the association between immune state and long-term DFS, we observed that pre-treatment warm tumors tended to have inferior DFS than cold tumors (pre-treatment: multivariable HR=2.18; 95% CI, 0.92 to 5.15, P=0.08)”

2. The authors state that: “we identified that immunologically cold TNBC was associated with downregulated lipid pathway” and hypothesized that “lipids, as essential components of immune cells, support the development of innate and adaptive immunity “. However, the metabolic state was calculated on the tumor cells and not the immune cells. Is it possible to apply the ISOpureR or other method to calculate an “immune cell” group similar to the calculation of the “tumor cell” only (Fig3a), and extract mRNA expression for this immune cell fraction? This will enable to differentiate between metabolic states occurring in the tumor and metabolic states occurring in the immune cells, and to better establish the interaction between immune states and metabolic states or the proliferative states of the immune cells themselves.

We agree with the reviewer’s comment that our metabolic states (presented in Fig. 3) were established only using gene expression profiling (GEP) of tumor cells. When extracting tumor cell GEP, we ran ISOpureR package with bulk tumor GEP, and normal breast tissue GEP as control. Although deconvolution tools could calculate the immune cell proportion based on bulk GEP data, it is still quite challenging to extract individual immune cell GEP matrix from bulk tumor GEP. In order to address this important issue and infer the various metabolic states corresponding to different immune cells, we have performed

analyses using the snRNA-seq data from the PROMIX trial (presented in Supplementary Figure 17), depicting the distribution of different metabolism profiles according to the cellular compartment (tumor, stromal, immune, normal) and cell type (T-cells, B-cells, macrophages etc). More importantly, we refined metabolic activity analyses on immune and stromal cells using the breast cancer single-cell atlas dataset (Sunny Z et al, Nature Genetics 2021), presented in Fig.6j-m.

Fig.6j UMAP dimensionality reduction diagram showing immune and stromal cell from HER2-negative subset of breast cancer single cell atlas (GSE176078) by sampleID, subtype, and immune state (n of sample=21, n of cell=54,652). Cell type was previously well annotated⁴⁰ as follows: B cells Memory (memory B cells (CD79A, MS4A1, CD27), naive B cells (CD79A, MS4A1, IGHD), plasmablasts (IGKC and IGLC2), CD8+ T cells (CD3, CD8), CD4+ T cells (CD3, CD4), NK cells (KLRC1, KLRB1, NKG7, AREG), cycling T cells (CD3, MKI67), NKT cells (KLRC1, KLRB1, NKG7, FCGR3A), macrophage (CD86), monocyte (CD127), cycling myeloid (KI67), DCs (CLEC9A or CD1C), endothelial (PECAM1, CD34 and VWF), MSC iCAF-like CAFs (ALDH1A1, KLF4 and LEPR), myCAF-like CAFs (ACTA2 (α SMA), TAGLN, FAP and COL1A1), PVL (ACTA 2, PDGFRB and MCAM). Metabolic pathways enriched in genes with highest contribution to the metabolic heterogeneities among immune cells (k) and stromal cells (l). m. Represented metabolic pathways score was compared between immune and stromal cells. TME, tumor micro-environment; FC, fold change; FDR, false discovery rate; CAF, cancer associated fibroblast; MSC,

mesenchymal stem cells; iCAFs, inflammatory-like CAFs; DCs, dendritic cell; PVL, perivascular-like.

Accordingly, the following paragraph was rewritten (page 16, line 14 to page 17, line 13 in the revised manuscript): “As complement to inherent limitations of the used snRNA-seq from PROMIX trial method⁴² for depicting immune and stromal cells (seen in Supplementary results, Supplemental Fig. 17), we conducted additional analyses mainly on metabolic profiles within immune and stromal cells using HER2-negative subset of breast cancer single-cell atlas dataset (GSE176078) (Fig.6j)⁴⁰. Interestingly, although breast epithelial cells (Supplemental Fig. 18a), immune cells (Fig.6k), or stromal cells (Fig.6l) predominantly depended on OXPHOS, metabolic flexibility and variation within the TME were identified (Supplemental Data7). Specifically, in epithelial cells, glycolysis was the most critical metabolic pathway besides OXPHOS, and was enriched in basal-like and cycling cancer cells that were highly proliferative (all GSEA FDR < 0.05) (Supplemental Fig. 18a and Supplementary Data 7). Likewise, glycolysis was also enriched in immune effector cells including memory B cells, CD8+ T cells, cycling T cells, monocyte and cycling myeloid (Fig.6k and Supplementary Data 7), which was revealed by a series of nuanced models⁴³⁻⁵² investigating the metabolism of T cell expansion and CD8+ effector differentiation. Cancer-associated fibroblasts (CAFs) (myofibroblast-like CAFs and inflammatory-like CAFs) and endothelial cells, serving as major components of tumor stroma and ECM, showed metabolic plasticity and shared similar metabolic activity (OXPHOS, glycolysis, glutathione, cytochrome P450) (Fig.6l and Supplementary Data 7). The 21 tumors were further classified into immunologically cold (n=13) and hot (n=8) based on mean CD8+ T cell proportion as cut-off value (Supplemental Fig. 18b). Then we calculated represented metabolic pathway GSVA score using pseudo-bulk gene profiles for immune and stromal cells for each sample, which were compared between hot and cold tumors (Fig.6m). Immune effector cells from hot tumors harbored higher metabolic activity than counterpart cells from cold tumors, including pyruvate (CD4+ T cells and NK cells), glycolysis (NK cells), citric acid cycle (CD8+ T cells and NK cells), and fatty acid metabolism (CD4+/CD8+ T cells) (Fig.6m). Conversely, amino acid (glutathione) metabolism signature score derived from CD4+ T cells was higher in cold tumors compared with hot tumors (Fig.6m), and we observed similar tendencies for cycling T-cells and memory B cells (both P=0.1) (Fig.6m).”

3. It is very interesting to understand the changes occurring throughout the treatment in individual patients. Following changes in individual patients over time can identify processes that are related with resistance or response. For example, the fraction of NK CD56 at surgery is a strong prognostic signature. How this marker is changed over time in individual patients? What accompanies this change, namely can we find any association with gene expression/protein/metabolic state that regulates its modulation?

We agree with the reviewer that exploring immune fraction changes occurring throughout the treatment in individual patients would be of interest. For this

reason, we have now evaluated the dynamic change of CD56 from baseline to on- and post-treatment samples.

We demonstrated increased NK CD56 bright cell signature levels over treatment time; only the scores at surgery correlated with disease recurrence but not with pCR. Of note, we observed enriched metabolic pathways (GSEA) in patients with low NK gene signature score.

These analyses have been added as Supplemental Fig.2d-f in the revised manuscript, and we have the following text to the revised manuscript:

“Specifically, we observed that difference in CD56^{bright} NK cells signature score between disease-free patients and those with relapse became more and more pronounced during NAC (Supplementary Fig. 2d), but no difference was found between pCR and non-pCR group (Supplementary Fig. 2e). Gene set enrichment analysis (GSEA) suggested that a series of metabolic pathways (i.e. glycolysis, fatty acid, oxidative phosphorylation (OXPHOS)) were enriched in patients with low NK cells signature score (Supplementary Fig. 2f).” (page 7, line 18 to 24)

Supplementary Fig. 2 d,e, NK CD56^{bright} cell gene signature score comparisons between patients with and without disease recurrence/pCR by timepoints. P values were derived from wilcoxon rank sum test. **f**. Gene set enrichment analysis (GSEA) based on differential genes between high and low NK CD56^{bright} cell gene signature scores.

Furthermore, we have explored the transcriptomic response to NAC over three timepoints, to reveal four distinct tumor biology related clusters (immune, metabolism, proliferation, extracellular matrix) (seen in Supplement results, Supplement Fig.11). More dedicated longitudinal analyses in immunometabolism are shown in Fig5.

4. Can we predict response according to the TME changes from T1 to T2? What would be the predictive power of the immune modulation relative to current predictions? Can we validate a clinically-relevant biomarker based on the TME findings (a specific metabolic signature)?

We have demonstrated that a positive TME change (conversion from cold or warm tumors pre-treatment to hot tumors post-treatment) was associated with improved pCR, as shown in Figure 5c. Per the reviewer's suggestion we have now performed regression analysis for the association between immune state (and combined with metabolic state) changes from T1 to T2 timepoints and pCR. This effect was independent from other clinicopathological parameters including tumor size, lymph node status and IHC subtype as stated in the following addition: "Patients with positive immune state change (OR=1.2, 95% CI, 1 to 1.45; P=0.05) (Fig.5d) were more likely to achieve pCR after adjusting for tumor size, lymph node status and breast cancer subtype. The same strategies were also applied to define tumor-cell-GEP based metabolic phenotype profile (i.e. positive change: patients maintained downregulated metabolic phenotype under NAC, or with changed metabolic phenotype (i.e. from upregulated/neutral to downregulated); all others are defined as negative change. Accordingly, we revealed that positive changes of TCA cycle (OR=1.28, 95% CI, 1.03 to 1.58; P=0.03) and nucleotide (OR=1.41, 95% CI, 1.09 to 1.82; P=0.01) metabolisms were independently associated with increased pCR. Similarly, patients with positive energy metabolism change showed a trend towards better treatment response (OR=1.36, 95% CI, 0.96 to 1.94; P=0.09). In addition, we evaluated the correlation between integrated immunometabolism profiles (Group1-4 shown in Fig.5e) and radiologic response (response group, N=28; no response group, N=41) after two treatment cycles. We found that patients with both positive immune state and metabolic phenotype profile were more likely to respond to NAC compared with other groups (Fig. 5e and Supplementary Table 7)." (page 13, line 11 to 24)

For better visualization and explanation of our results, we have now added forest plot with respective OR in the updated Figure 5d

Fig5. d Forest plot depicting multivariable the logistic regression model adjusting for tumor size, lymph node status, and IHC subtype that assess the association between immunometabolism profiles and treatment response (pCR).

Minor comments:

1. Introduction, 2nd paragraph – rephrase the sentences for better fluency

The text indicated by the reviewer has been rewritten for better fluency in the revised manuscript which now reads (page 3, line 14 to 25): “Metabolic reprogramming allows tumors to acquire metabolic properties that support cell survival, evasion of immune surveillance, and hyperplastic growth⁹, which not only meets specific demands for increased energy, biomass, redox maintenance, and cellular communication, but also interacts with the complex TME¹⁰⁻¹². Previous studies unraveled the plasticity of cancer metabolism in vitro and identified alterations in several metabolic enzymes, fluxes and mediators¹³⁻¹⁶. Moreover, immunometabolic interplay exists in TME, where complex metabolic networks of tumor, stromal and immune cells are dictated by cell-intrinsic and environmental factors¹⁷. This diverse milieu of immune, tumor and stromal cells creates a complex and dynamic ecosystem that can be influenced by cancer type and treatment^{18,19}. There is emerging interest in the cancer immunometabolic phenotype, which can have clinical implications^{17,20}, as a source of prognostic biomarkers²¹⁻²³ and for therapeutic targeting²⁴⁻²⁶. Yet, the immunometabolic remodeling that occurs during treatment is not well understood. Neoadjuvant studies provide a unique window to assess predictive biomarkers and response to therapy in vivo²⁷.”

2. Figure 1: please include the number of patients and samples used for each methodology within the scheme

We have now added the number of patients and samples analyzed according to each method used in the updated **Figure 1**, per the reviewer's suggestion.

Fig1: Overview of the study design and multi-omics data collection. a. Pre/on/post-treatment breast cancer tissue were collected from a phase II PROMIX trial (ClinicalTrials.gov identifier NCT00957125). Enrolled patients with locally advanced (tumor size>20 mm) HER2-negative breast cancer, who were scheduled to receive six cycles of NAC with a combination of epirubicin and docetaxel. Bevacizumab was added during cycles 3–6 for those patients who did not achieve a clinical complete response (cCR) after the second cycle of NAC. For details on clinicopathologic characteristics, see **Supplemental Table1**. b. Heatmap showing longitudinally in-depth MS-based proteomics, transcriptomics, genetic and mflHC/IF profiling from PROMIX trial. c. UpSet plot showing multi-omics data intersection. HER2, Human epidermal growth factor receptor-2; mflHC, multiplex fluorescent immunohistochemistry; MS, mass spectrometry.

3. Figure 2A – include the tumor subtype for each tumor in the clinical parameters

We have now included the tumor subtype (PAM50) for each tumor in the updated **Figure 2A**, per the reviewer's suggestion.

Reviewer #3 (Remarks to the Author):

In this study, authors performed mainly a longitudinal (2-3 time points) transcriptome study in HER2-negative breast cancer patients (n = 122). They classified all samples into immune hot/warm/cold subtypes based on transcriptome, and discussed their relationship to 7 metabolic gene signatures. They showed the dynamic changes of immune subtype and metabolic gene signatures upon neoadjuvant chemotherapy (NAC). They also performed proteomics (n = 29), snRNA-seq (n = 8) and WES (n = 20) on a subset of the patients on 2-3 time points to describe these multi-omics features on the immune subtypes and metabolic clusters. Most of the results are descriptive and associations, which is typical for multi-omics studies, but this manuscript lacks clear conclusions or a significant discovery.

Main concerns:

1. Authors state that they are performing proteogenomic analysis on breast cancer, but actually most of the analyses are based on their transcriptomic data. The number of samples profiled by both proteomics and genomics are roughly 50 samples in 20 patients, which is limited for a proteogenomic study. Therefore, they as expected

didn't get any notable discoveries. Proteomic data is only used in some differential analysis, and WES is used in the last part, which have little add to the paper. They may profile more proteogenomic samples to match their title and make new discoveries on larger datasets. Otherwise, the title at least should be changed.

We agree with the reviewer that most of the analyses are based on the transcriptomic data in the whole cohort and supported by proteomic and single cell data in a subset of patients. Given that this study was conducted more than a decade ago, we were restricted by material availability. Therefore, in accordance with the reviewer's suggestion, we have now changed the article's title to the more descriptive and representative one: "Longitudinal molecular profiling elucidates immunometabolism dynamics in breast cancer". Besides, we also updated the Fig1, to provide better visualization of our partially overlapping multi-omics data. Please refer to our response to reviewer #2 (minor Q2).

Fig1: Overview of the study design and multi-omics data collection. a. Pre/on/post-treatment breast cancer tissue were collected from a phase II PROMIX trial (ClinicalTrials.gov identifier NCT00957125). Enrolled patients with locally advanced (tumor size>20 mm) HER2-negative breast cancer, who were scheduled to receive six cycles of NAC with a combination of epirubicin and docetaxel. Bevacizumab was added during cycles 3–6 for those patients who did not achieve a clinical complete response (cCR) after the second cycle of NAC. For details on clinicopathologic characteristics, see **Supplemental Table1**. b. Heatmap showing

longitudinally in-depth MS-based proteomics, transcriptomics, genetic and mflHC/IF profiling from PROMIX trial. c. UpSet plot showing multi-omics data intersection. HER2, Human epidermal growth factor receptor-2; mflHC, multiplex fluorescent immunohistochemistry; MS, mass spectrometry.

Although the data included in this study were limited for a typical proteogenomic analyses, we highlighted the advantages from each data layer at the beginning of the results: “The intersection of the proteomics data with GEP and WES is shown in **Fig1c**. Herein, multi-omics data enabled comprehensive analyses on the correlation of immunometabolic phenotype and treatment response/long-term survival (GEP), the interaction of immune state and tumor metabolism (GEP), potential metabolic targets that modulated TME (paired GEP and proteomics), metabolic characteristics of breast epithelial, immune and stromal cells per immune state (snRNA-seq), tumor and immune state co-evolution under NAC (WES and GEP).” **(page 5, line 12 to 18)**

2. Authors should more precise to describe their cohort, because they are not studying all kinds of breast cancer but only HER2- breast cancer.

We have now added following **Supplemental Table1, to better describe our main cohort.**

Supplementary Table1. Clinicopathologic characteristics of BC patients from PROMIX trial.

Factor	N (%)
Age (mean (SD))	50.00 (9.9)
Menopausal status	
Premenopausal	90 (60.0)
Premenopausal (0-5 years)	17 (11.3)
Premenopausal (more than 5 years)	42 (28.0)
Missing	
Histologic Type	
IDC	108 (72.0)
ILC	22 (14.7)
Other	15 (10.0)
Missing	5 (3.3)
Histologic Grade	
I	4 (2.7)
II	46 (30.7)
III	33 (22.0)
Missing	67 (44.7)
KI67 percentage (mean (SD))	36.0 (24.8)
ER status	
Negative	41 (27.3)
Positive	107 (71.3)
Missing	2 (1.3)
PR status	
Negative	64 (42.7)
Positive	84 (56.0)
Missing	2 (1.3)
Subtype	
LumA	48 (32.0)
LumB	62 (41.3)
TNBC	38 (25.3)
Missing	2 (1.3)
Clinical Stage	
0	4 (2.7)
I	35 (23.3)
II	62 (41.3)
III	39 (26.0)
Missing	10 (6.7)
Response after two cycles of NAC	
Complete response	5 (3.3)
Partial response	64 (42.7)
Stable disease	74 (49.3)
Progressive disease	4 (2.7)
Missing	3 (2.0)
pCR status	
No	129 (86.0)
Yes	20 (13.3)
Missing	1 (0.7)
Recurrence	
No	127 (84.7)
Yes	23 (15.3)
DFS, month	76.80 (33.97)

Abbreviations: SD, standard deviation; IDC, invasive ductal carcinoma; ILC, invasive lobular carcinoma; IHC, immunohistochemistry; NAC, neoadjuvant chemotherapy; ER, estrogen receptor; PR, progesterone receptor; TNBC, triple negative breast cancer; pCR, pathologic complete response; DFS, disease-free survival.

We also only focused on the study of HER2-negative breast cancer, and repeated and refined analyses using HER2-negative subset of the validation datasets (seen in updated Fig.6, Supplemental Fig3, Supplemental Fig7, Supplemental Fig10, Supplemental Fig3 and Supplemental Table2). Accordingly, the methods section (page 36, line 22 to page 37, line 5) now reads: “A series of HER2-negative breast cancer cohorts were obtained to validate findings from this study externally (see detailed in Supplementary Table2), including Oslo2 cohort³⁵ (42 patients with both RNA-seq and MS-based proteomics), South Korean NAC cohort⁶³ (86 patients treated with NAC, with longitudinal RNA-seq data (pre/on/post-NAC, n=144)), and a scRNA-seq cohort⁴⁰ (a total of 21 untreated patients with single-cell RNA-seq). All samples from Oslo2 and scRNA-seq cohorts

were previously untreated primary breast cancers, and the South Korean NAC cohort included both pre-treatment biopsies and post-treatment samples.”

3. The transcriptome in HER2- breast cancer is extensively studied, and the hot/cold immune subtypes are also well studied. Authors should pay more attention to the longitudinal and NAC features of their dataset. But they didn't raise any new points on why patient response/not response to NAC.

We have analyzed dynamic transcriptomic profiles under NAC, identifying four biological clusters (immune response (C1), metabolism (C2), extracellular matrix (ECM) (C3), and tumor proliferation (C4)). Please see details in Supplementary Fig11 and Supplementary Results.

In this work, we demonstrate that a positive immune/metabolic state change is associated with improved pCR, as shown in Figure 5c. We have now added the respective forest plot with the odds ratio and confidence intervals for the association of longitudinal TME changes and likelihood for pCR in the updated Fig. 5d.

Fig5. d Forest plot depicting multivariable the logistic regression model adjusting for tumor size, lymph node status, and IHC subtype that assess the association between immunometabolism profiles and treatment response (pCR).

We now extended analyses of CD56^{bright} NK cells gene signature (in updated Supplementary Fig. 2 d,e) that is a strong prognostic factor for long-term survival, please refer also to our response to the reviewer #2 (Q3&4).

Supplementary Fig. 2 d,e, NK CD56^{bright} cell gene signature score comparisons between patients with and without disease recurrence/pCR by timepoints. P values were derived from wilcoxon rank sum test. **f**. Gene set enrichment analysis (GSEA) based on differential genes between high and low NK CD56^{bright} cell gene signature scores.

Besides, we also demonstrate that combined immunometabolic profiles also correlate with treatment response (Fig. 5e). We identified individual biomarker genes/protein that showed differential expression between on- and pre-treatment in patients with different treatment responses (shown in Fig.5f-g). Furthermore, we observed changeable metabolic epithelia cell clusters composition and metabolic pathways score for patients with positive and negative TME changes (shown in Fig.6c,i).

4. In Fig. 2c hot tumor has a lower Macrophage, especially M2, which is opposite to the signature results as shown in Fig. 2a. This brings the major concern that most results in this study are based on signature activity that maybe can't be validated by orthogonal methods.

We thank the reviewer for this comment. Based on the data presented in Fig. 2a, there is an upregulation of the macrophage signature in the hot tumors. This signature contains signals from both M1 and M2 macrophage-related genes -as shown in Figure 2a- without separating the subtype of origin. However, as presented in the Supplementary Table 3 on a detailed description of the data accompanying Figure 2, the proportion of the mean macrophage M2 cell fraction was significantly lower in hot compared to cold tumors (0.23 versus 0.37, respectively), based on signature activity and shows an opposite pattern for the macrophage M1 cell fraction. The M2 cell fraction gradient from

cold to hot tumors was in line with the multiplex IHC results -presented in Figure 2c- even in a small subset of patients. Furthermore, the observed pattern of enrichment of M1 macrophages in hot tumors and vice versa for M2 ones, is in accordance with the previous reports on the differential role of anti-tumoral (M1) and pro-tumoral (M2) macrophages (De Palma M et al. Cancer Cell 2013).

5. Results in Fig. 3 is based on gene signatures, authors have no substantial evidence and findings to state there is a critical interplay between tumor metabolism and immune state. More analyses on single cell data may help.

We thank the reviewer for this crucial point. Since Fig3b only showed univariate comparison of metabolic phenotype between immune states, we investigated the interaction of tumor metabolic and immune state based on the linear mixed-effects models (LMEM) as below (also seen in our response to Reviewer #1, other comment 2):

$$I_{ij} = \beta_0 + \beta_1 M_{ij} + \beta_2 S_{ij} (+ \beta_3 C_{ij}) + \gamma_j + \epsilon_{ij}, \quad (4)$$

The tumor cellularity (C_{ij}) was adjusted when we ran LMEM using Korean validation cohort, where tumor-derived GEP was not available.

The model was established, adjusting for subtype (and purity) with a random term (varying across patients). Fig3c below was added to present details of model parameters and significance. We demonstrated tumor TCA cycle, lipid, amino acid, vitamin/cofactor are significantly negatively correlated with immune state in PROMIX, and only TCA-cycle was validated in the Korean RNA-seq cohort (Supplementary FigureS7).

$$I_{ij} \sim \beta_0 + \beta_1 M_{ij} + \beta_2 S_{ij} + \gamma_j + \epsilon_{ij}$$

Fig.3c Funkyheatmap depicting the coefficient, random term, residual variance and p value of LEME, which was conducted within all samples to identify interaction effects between immune state (I) and tumor metabolic phenotypes (M), adjusting for the breast cancer subtype (S).

In addition, we have now performed additional *in vitro* experiments in order to further explore the interplay between immune states and tumor metabolism. For detailed description of our findings please refer to our response to reviewer #1 Q1.

6. In Fig. 3c the prognostic value of metabolic phenotypes presented is limited, some are not even significant, such as amino acid $p = 0.07$.

The relatively low number of patients in our clinical study could have affected the statistical analysis which deemed to be non-significant for the correlation of metabolic phenotypes with pCR, now as presented in Fig. 3d. Given the number of correlations between immunometabolism phenotypes and pCR/DFS, we summarized the results in Supplementary Table5 as following:

Supplementary Table6. Multivariate HRs or ORs (95% CIs) for DFS or pCR by immunometabolic subtypes.

Group	Multivariable-adjusted HR/OR (95% CI)	P value
DFS^a Upregulated vs. downregulated/Hot vs. Cold		
TME	1.32 [0.60, 2.90]	0.49
Amino Acid (TC)	1.11 [0.51, 2.39]	0.79
Amino Acid (Bulk)	1.14 [0.51, 2.58]	0.75
Lipid (TC)	1.49 [0.48, 4.62]	0.49
Lipid (Bulk)	2.83 [0.58, 13.71]	0.20
Carbohydrate (TC)	2.62 [1.07, 6.44]	0.04
Carbohydrate (Bulk)	1.84 [0.64, 5.31]	0.26
TCA cycle (TC)	2.89 [1.16, 7.21]	0.02
TCA cycle (Bulk)	0.96 [0.36, 2.58]	0.94
Energy (TC)	1.54 [0.35, 6.65]	0.57
Energy (Bulk)	1.16 [0.22, 6.12]	0.87
Nucleotide (TC)	1.17 [0.42, 3.31]	0.76
Nucleotide (Bulk)	1.00 [0.83, 1.21]	0.99
Vitamin/co-factor (TC)	1.34 [0.48, 3.76]	0.57
Vitamin/co-factor (Bulk)	1.03 [0.82, 1.28]	0.81
pCR^b Upregulated vs. downregulated/Hot vs. Cold		
TME	1.33 [1.15, 1.54]	<.001
Amino Acid (TC)	0.87 [0.76, 1.01]	0.07
Amino Acid (Bulk)	1.02 [0.92, 1.14]	0.69
Lipid (TC)	0.94 [0.77, 1.13]	0.50
Lipid (Bulk)	0.77 [0.59, 1.00]	0.06
Carbohydrate (TC)	0.95 [0.77, 1.17]	0.61
Carbohydrate (Bulk)	0.93 [0.77, 1.12]	0.46
TCA cycle (TC)	0.87 [0.74, 1.03]	0.10
TCA cycle (Bulk)	0.87 [0.71, 1.06]	0.16
Energy (TC)	0.90 [0.70, 1.15]	0.41
Energy (Bulk)	0.83 [0.65, 1.05]	0.14
Nucleotide (TC)	0.77 [0.61, 0.98]	0.04
Nucleotide (Bulk)	3.40 [0.41, 28.44]	0.26
Vitamin/co-factor (TC)	0.89 [0.75, 1.05]	0.18
Vitamin/co-factor (Bulk)	2.85 [0.50, 16.29]	0.24

^aStratified by breast cancer subtype (luminal and TN) due to PH assumption violation and adjusted for tumor size (<2cm, 2-5cm, >5cm) and lymph node status (metastasis, no metastasis).

^bAdjusted for breast cancer subtype (luminal, TN), tumor size (<2cm, 2-5cm, >5cm) and lymph node status (metastasis, no metastasis).

7. The 7 metabolic gene sets used here are broad functionality, that consists many genes, such as lipid contains 766 genes, with such many genes as a signature, how the combined effect be representative should be discussed, a more detailed and specific metabolic gene sets if exists may help.

We agree with the reviewer that the 7 metabolic states represent a broad functionality and some include a large number of genes. According to the reviewer's request, we have now validated the findings using well-established KEGG metabolic signatures (<https://www.genome.jp/kegg/pathway.html#metabolism>) in **Supplementary Fig. 6**, demonstrating a high correlation with our initially reported results.

Supplementary Fig. 6 Comparison of representative KEGG metabolic phenotype among three metabolic phenotypes in e amino acid, f lipid, g carbohydrate, h TCA cycle, i nucleotide, j energy, k vitamin/co-factor. P values were derived from Kruskal–Wallis tests.

The corresponding paragraph now reads (page 9, line 12 to 15): “Furthermore, representative metabolic protein abundance (Supplemental Fig. 6a-d) as well as KEGG metabolism signatures (Supplemental Fig. 6e-k) were compared between groups (all $P \leq 0.1$), which also showed a good concordance with tumor-cell based metabolic group.”

8. More comprehensive immune subtypes have been extensively studied in various cancer types and pan-cancer studies. The three simple immune class presented in

this study is outdated. More sophisticated immune subtypes may lead to more interesting discoveries. For instance, CAF status is known to be related to immune response, which is not even mentioned in this part.

We agree with the reviewer that more comprehensive immune subtypes have been previously reported. However, we wanted to start our analysis/approach by capturing a broader and rough immune landscape categorization in order to more easily explore its correlation with metabolic states. Nevertheless, we already calculated the most popular pan-cancer immune subtype (i.e. IFN- γ dominant, Inflammatory, TGF- β dominant, lymphocyte depleted, Wound healing) in the PROMIX cohort (seen in Fig2a and Supplemental Table3), which showed predominance of IFN- γ dominant and Inflammatory subtype in our cohort.

We now have further explored the TME-metabolism correlation for different immune/stromal components (including CAFs) using breast cancer single-cell atlas dataset (presented in Supplemental Fig18 and updated Fig.6j-m). Metabolic activity was directly compared within immune and stromal cell from hot and cold tumors. Please refer the Reviewer #2 (Q2) for more details.

Supplementary Fig. 18 Metabolic heterogeneity in TME analyzed based on the scRNA-seq cohort⁸. Metabolic pathways enriched in genes with highest contribution to the metabolic heterogeneities among malignant cells (a). b. The cell proportion of hot and cold tumors. The single epithelial cell subpopulation was previously annotated⁹ (Sunny Z. Wu et al, Supplementary Table 4. scSubtype gene lists).

Immune and Stromal Cells from GSE176078 (BRCA single cell atlas)

Fig6 j UMAP dimensionality reduction diagram showing immune and stromal cell from HER2-negative subset of breast cancer single cell atlas (GSE176078) by sampleID, subtype, and immune state (n of sample=21, n of cell=54,652). Cell type was previously well annotated⁴⁰ as follows: B cells Memory (memory B cells (CD79A, MS4A1, CD27), naive B cells (CD79A, MS4A1, IGHD), plasmablasts (IGKC and IGLC2), CD8+ T cells (CD3, CD8), CD4+ T cells (CD3, CD4), NK cells (KLRC1, KLRB1, NKG7, AREG), cycling T cells (CD3, MKI67), NKT cells (KLRC1, KLRB1, NKG7, FCGR3A), macrophage (CD86), monocyte (CD127), cycling myeloid (KI67), DCs (CLEC9A or CD1C), endothelial (PECAM1, CD34 and VWF), MSC iCAF-like CAFs (ALDH1A1, KLF4 and LEPR), myCAF-like CAFs (ACTA2 (α SMA), TAGLN, FAP and COL1A1), PVL (ACTA 2, PDGFRB and MCAM). Metabolic pathways enriched in genes with highest contribution to the metabolic heterogeneities among immune cells (k) and stromal cells (l). m. Represented metabolic pathways score was compared between immune and stromal cells. TME, tumor micro-environment; FC, fold change; FDR, false discovery rate; CAF, cancer associated fibroblast; MSC, mesenchymal stem cells; iCAFs, inflammatory-like CAFs; DCs, dendritic cell; PVL, perivascular-like.

Accordingly, we re-write the paragraph as following (page 16, line 14 to page 17, line 13 in the revised manuscript): “As complement to inherent limitations of the used snRNA-seq from PROMIX trial method⁴² for depicting immune and stromal

cells (seen in Supplementary results, Supplemental Fig. 17), we conducted additional analyses mainly on metabolic profiles within immune and stromal cells using HER2-negative subset of breast cancer single-cell atlas dataset (GSE176078) (Fig.6j)⁴⁰. Interestingly, although breast epithelial cells (Supplemental Fig. 18a), immune cells (Fig.6k), or stromal cells (Fig.6l) predominantly depended on OXPHOS, metabolic flexibility and variation within the TME were identified (Supplemental Data7). Specifically, in epithelial cells, glycolysis was the most critical metabolic pathway besides OXPHOS, and was enriched in basal-like and cycling cancer cells that were highly proliferative (all GSEA FDR < 0.05) (Supplemental Fig. 18a and Supplementary Data 7). Likewise, glycolysis was also enriched in immune effector cells including memory B cells, CD8+ T cells, cycling T cells, monocyte and cycling myeloid (Fig.6k and Supplementary Data 7), which was revealed by a series of nuanced models⁴³⁻⁵² investigating the metabolism of T cell expansion and CD8+ effector differentiation. Cancer-associated fibroblasts (CAFs) (myofibroblast-like CAFs and inflammatory-like CAFs) and endothelial cells, serving as major components of tumor stroma and ECM, showed metabolic plasticity and shared similar metabolic activity (OXPHOS, glycolysis, glutathione, cytochrome P450) (Fig.6l and Supplementary Data 7). The 21 tumors were further classified into immunologically cold (n=13) and hot (n=8) based on mean CD8+ T cell proportion as cut-off value (Supplemental Fig. 18b). Then we calculated represented metabolic pathway GSVA score using pseudo-bulk gene profiles for immune and stromal cells for each sample, which were compared between hot and cold tumors (Fig.6m). Immune effector cells from hot tumors harbored higher metabolic activity than counterpart cells from cold tumors, including pyruvate (CD4+ T cells and NK cells), glycolysis (NK cells), citric acid cycle (CD8+ T cells and NK cells), and fatty acid metabolism (CD4+/CD8+ T cells) (Fig.6m). Conversely, amino acid (glutathione) metabolism signature score derived from CD4+ T cells was higher in cold tumors compared with hot tumors (Fig.6m), and we observed similar tendencies for cycling T-cells and memory B cells (both P=0.1) (Fig.6m).”

9. Authors may add number of samples sequenced by each tech in Fig. 1.

We have now added the number of patients and samples analyzed according to each method used in the updated Fig.1, per the reviewer’s suggestion. Please refer also to our response to the reviewer #2 (minor comment 2).

10. Add breast cancer subtype etc. information in Fig.2a. column label may help.

Breast cancer subtype has now been added as a column label in the updated Fig. 2a in the revised manuscript, per the reviewer’s suggestion. Please refer also to our response to the reviewer #2.

11. There are lots of abbreviations without explanation : for instance: BC、 TNBC, HR+ in Fig. 2e and pCR, DFS etc

We have now explained the abbreviations throughout the text. Since both hormone receptor and hazard ratio have the same abbreviation “HR”, we replaced hormone receptor with estrogen receptor.

12. Fig. 5c TME change is not clear, can't visually spot.

We agree with the reviewer that in the current Fig. 5c the TME change is not clear, given that we presented a selection of immune cell components as rows. In order to improve the visualization and show a clearer TME change, we have now increased the number of rows (same with fig2a), as presented in the updated Fig. 5c in the revised manuscript.

13. Fig. 5c why DFS is binary?

In order to improve the visualization and interpretation of the results presented in Fig. 5c, we have now replaced DFS (in the updated figure) with the term “recurrence/death”, which is presented as a binary variable.

14. Typo: of which 7,357 proteins were quantified in each of the of the 53 tumors

The typographical error has now been corrected

15. Whether GEP and protein data have batch effect and if any how they are addressed is not discussed. Low dimensional visualization on raw and normalized data should be shown.

We agree with the reviewer regarding this important comment on batch effect. Regarding the GEP, the lack of batch effect has been previously reported by Kimbung et al (Int J Cancer. 2018 Feb 1;142(3):618-628; doi: 10.1002/ijc.31070, Supplementary Figure 1) in the first publication of the GEP data from the PROMIX trial. Regarding proteomic data, we now present QC analyses based on TMT values in order to investigate the presence of any batch effects. We show raw and normalized protein abundance across samples according to the TMT set, and normalized profiles were normally distributed. Euclidean distance heatmap did not find any significant outlier samples. Principal component analysis (PCA) indicated that the variation in proteomic data mainly derives from subtype. The respective plots are presented in the **Supplemental Fig8** in the revised manuscript.

(A)

(B)

(B)

(C)

Supplementary Fig. 8 Quality control of mass spectrometry (MS) data. Boxplot show unnormalized (a) and normalized (b) protein abundance by TMT set. c. Heatmap depicting euclidean distance of samples calculated by mass spectrometry data. d. Principal component analysis by treatment timepoint, subtype, TMT set and TME tage, respectively.

We have now added this description into the revised manuscript: “Protein abundance data were generated based on a subset of pre/on-treatment samples (N=53) using mass spectrometry-based proteomics. No significant batch effect between six TMT sets was detected (Supplemental Fig. 8)”. (page 11, line 1 to 3)

16. Response to NAC is the one of the unique features in the dataset. Authors may focus more on this point. Some of the analysis, such as Fig. 4f and Fig. 5e can be performed between patients who response or not response to NAC. Tumor microenvironments form snRNA-seq may also help here. Maybe authors can dig more to find specific genes that potentially related to NAC response, and test it. Such as predict genes and related drugs could potentially reverse the response pattern, for example by using cmap dataset

We have previously addressed this question and reported the results in the same cohort (Kim et al Cell 2018), showing that various transcriptional profiles were acquired by reprogramming in response to chemotherapy (defined as clonal extinction, similarly to pCR). We therefore focused more biomarker genes associated with the treatment response that were simultaneously identified by gene expression profiles and proteomics. We already previously addressed this point using DepMap “Metabolic druggable proteome upregulated in cold tumors such as FASN, whose inhibitor (cerulenin) is a potential candidate for inhibiting tumor growth and simultaneously boosting TME function (Supplemental Fig. 9g). Likewise, other potential drug targets with experimental evidence like ACACA (inhibitor: KD-023), ALDOA and HMGCS1 were also identified (Supplemental Fig. 9h-j).” (page 12, line 13 to 15). More importantly, we conducted in vitro experiments to validate metabolic enzymes and their interplay with immune cells (seen in our response to reviewer #1, Q1).

17. Whether the single cell immune proportion is highly correlated with the three immune subtypes form GEP?

We have shown the expected difference of single cell immune proportion within CD45+ cells between three immune sates (Supplementary Fig.14f), where pooled hot tumors have higher compositions of CD4/8+ T cells and B cells than cold tumors.

We have now performed an additional analysis in order to demonstrate if the single cell immune proportion is correlated with the three bulk GEP-based immune subtypes. Despite the limited number of available samples with immune cell snRNA-seq information (n=16), we could demonstrate a trend towards to higher T cell proportion in hot tumors, shown in the figure below.

18. Different immune cell types do not appear to be well separated in Fig.S13. Please show Fig.S13d painting by samples. Seems to me the batch effect in single cell data is not well addressed

The incomplete separation of the immune cell types based on the single cell data is attributed to the inherent limitations of the used snRNA-seq method for depicting immune and stromal cells (Ruli et al, Nature Communications 2017), which makes immune cell analyses difficult (Method part). We employed the same strategy (package harmony, <https://github.com/immunogenomics/harmony>) as epithelial cell analyses (Fig6) to remove batch effect within immune cells, and updated the Suppl. Fig 13d, painting by samples, per the reviewer's suggestion. In addition, we use breast cancer single cell atlas to explore metabolic heterogeneity within immune and stromal cells. Please refer the Reviewer2 (Q2) for more details.

19. RNA velocity can be applied to show the tumor evolution upon NAC

We fully agree that RNA velocity analyses could imply current and future cell states under NAC. RNA velocity leverages the relative ratio between intronic (unspliced, nuclear) and exonic (spliced, cytoplasmic) mRNAs in scRNA-seq data (Chenglong et al, PNAS 2019), however, our single-nuclei RNA-seq has limited power to calculate RNA velocity (i.e. does not cover cytoplasmic mRNAs).

Therefore, we employed Monocle3 (1.3.4) to conduct breast epithelial cell trajectory analysis instead. We demonstrated pseudo-time order existed in our metabolic-based epithelial clusters. The respective method and results are presented in the Fig.6f and added as text which reads:

“Single-cell trajectory analysis within metabolic epithelial cell clusters was conducted using Monocle3 (1.3.4)¹⁰⁴, which uses an algorithm to learn the sequence of gene expression changes each cell.” (page 33, line 21 to 23)

“Furthermore, the transition of MC2 through the other metabolic clusters was strongly supported by the trajectory analysis (Fig. 6f). Pseudo-time ordering demonstrated an ordered, progressive, stepwise transition from normal breast epithelial cells (MC1) to malignant hypoxic and glycolytic phenotype (MC2) (Fig. 6f)” (page 15, line 17 to 20) in the updated manuscript.

Fig6. f UMAP of metabolic breast epithelial cell clusters, colored by pseudotime, calculated using Monocle3 (1.3.4).

20. Are there any gene mutations more represented in NAC response tumors than non-responders, and whether the pattern is the same before and on/post treatment?

We thank the reviewer for this remark. However, we have previously addressed this question and reported the results in the same cohort (Kim et al Cell 2018), showing that resistant genotypes were pre-existing and adaptively selected by neoadjuvant chemotherapy. By applying single-cell DNA and bulk exome sequencing methods, we profiled longitudinal samples during neoadjuvant chemotherapy and revealed the different mutations arising in patients with clonal persistence (corresponding to non-pCR).

21. Is CNV based clonality change the same as mutation based one

We have now performed an additional CNV-based clonality analysis in order to evaluate if it changes in the same way as the mutation-based one. We demonstrated that CNV burden significantly and consistently decreased within patients with tumor clonal extinction, but it shows comparable level cross treatment timepoint in patients in the clonal persistent group. These results are presented in the Supplemental Fig19 and added as text which reads: “Somatic copy number alteration (SCNA) profiles were in agreement

with the changes of mutation number in each clonal evolution group, respectively (Supplemental Fig. 20f-g)” (page 19, line 18 to 20 in the revised manuscript).

Supplementary Fig. 20 f,g The barplot shows the number of copy number alterations within longitudinal samples on clonal persistence and extinction group.

22. Some conclusions in the manuscript may need further evidence: such as: “suggesting that drug-resistant tumors may have a higher rate of glycolysis and be capable of escaping antitumor immunity.” and “The disappearance of MC5 on/post-treatment in positive immune state change group could be explained by chemokines as CXCL12/14 that are known for the recruitment of T cells, and notch signaling that regulates mature of CD4/8+ T cells were enriched in MC5”etc.

In accordance with the reviewer’s suggestion, we deleted the following sentence “suggesting that drug-resistant tumors may have a higher rate of glycolysis and be capable of escaping antitumor immunity.”, and toned down the explanatory sentences such as “The disappearance of MC5 on/post-treatment in positive immune state change group could be explained by highly expressed chemokines, which directed the migration of immune cells into tumor tissue⁴¹”. (page 15, line 24 to page 16, line 1)

Reviewer #4 (Remarks to the Author):

Having the longitudinal tumor samples from patients in the PROMIX trial as proteomic profiling (using 4 different platforms) would be an important and rich information resource for the scientific community. However, this manuscript requires several improvements. Several specific experimental comments to address include:

General Comments:

1. Generally, the manuscript is hard to follow in some sections. The text jumps between methods, results, and discussion without clear transitions. It would benefit from a more structured format guiding the reader through the methodology, findings, and implications step by step. Simplifying the presentation of results and using more narrative to describe the significance of findings could help.

We have extensively restructured the manuscript and throughout the text have

added methods before results to improve the flow and comprehension. In addition, we have moved supportive paragraphs to supplemental results.

2. It's not immediately clear what the main takeaways should be. The paper could benefit from a clearer statement of the primary conclusions and their significance in each section. The paper suggests that the findings could be useful as biomarkers of treatment efficacy but doesn't elaborate on how this would work in practice.

We have now extensively edited the manuscript text in order to improve its flow and present our data, methodology and results in a clearer and more structured and comprehensive way. We now added "introduction" and "conclusion" paragraphs for each result part, in order to highlight our aims of analyses and the most important findings. As for useful biomarkers, for example, we now extended the analyses on CD56^{bright} NK cells gene signature that is a strong prognostic factor for long-term survival. Please also refer to the updated Supplementary Fig.2 and our response to Reviewer #2 (Q3).

3. One of the critical aspects that seem underdeveloped in the manuscript is the integration of the various platforms used for proteomic and transcriptomic analyses. While the use of multiple platforms is commendable and has the potential to offer a comprehensive view of the TME, the authors have not sufficiently highlighted how these methods complement each other. It is essential that the authors delineate the synergistic benefits of combining these technologies. How each contributes to a better understanding of tumor metabolism and immune interactions. Moreover, the overarching message and contributions of the study remain obscured. I suggest trying to simplify the results and highlight the importance of each section.

We now updated Fig1 with multi-omics data availability for each patient and data intersection (also seen in our response to Reviewer #2, minor comment 2), and added a summarizing paragraph: "The intersection of the proteomics data with GEP and WES is shown in Fig1c. Herein, multi-omics data enabled comprehensive analyses on the correlation of immunometabolic phenotype and treatment response/long-term survival (GEP), the interaction of immune state and tumor metabolism (GEP), potential metabolic targets that modulated TME (paired GEP and proteomics), metabolic characteristics of breast epithelial, immune and stromal cells per immune state (snRNA-seq), tumor and immune state co-evolution under NAC (WES and GEP)." (page 5, line 12 to 18)

Major:

1. There are several groups of data and groups analyzed, which is not clearly explained in every section. For example, Section "MS-based Proteomic Landscape of Immunometabolism Phenotype and Pathways". It is confusing why there is a new set of samples, and this happens in each section. The paper needs a better illustration of the sets used for each platform, and a more complete Figure 1 with this information could help. There are so many different analyses in this paper, and it is not well summarized.

In accordance with the reviewer's suggestion, we have updated the Fig1, to provide better visualization of our partially overlapping multi-omics data.

Fig1: Overview of the study design and multi-omics data collection. a. Pre/on/post-treatment breast cancer tissue were collected from a phase II PROMIX trial (ClinicalTrials.gov identifier NCT00957125). Enrolled patients with locally advanced (tumor size>20 mm) HER2-negative breast cancer, who were scheduled to receive six cycles of NAC with a combination of epirubicin and docetaxel. Bevacizumab was added during cycles 3–6 for those patients who did not achieve a clinical complete response (cCR) after the second cycle of NAC. For details on clinicopathologic characteristics, see **Supplemental Table1**. b. Heatmap showing longitudinally in-depth MS-based proteomics, transcriptomics, genetic and mflHC/IF profiling from PROMIX trial. c. UpSet plot showing multi-omics data intersection. HER2, Human epidermal growth factor receptor-2; mflHC, multiplex fluorescent immunohistochemistry; MS, mass spectrometry.

2. The description of patient demographics utilized in this study is lacking. Even if this data is derived from previously published clinical trials like the PROMIX trial, the authors should incorporate a detailed table of the clinical features, groupings of clinical variables (clinical receptor status, etc.), and groups described in this manuscript from some of the analysis, survival data, subtypes, etc., as metadata for the analysis performed. This addition is essential for reproducibility and should be clearly indicated. The scientific community will need this data to use everything detailed in the data availability section. If it is already included somewhere, please specify it better in the manuscript.

We have now added supplemental table 1 where we present patient demographics, tumor clinicopathologic features, on/post-treatment response and long-term survival. Due to patient consent and confidentiality agreements, the meta data from clinical trial can be made available for validation purposes by contacting the corresponding author.

Supplementary Table1. Clinicopathologic characteristics of BC patients from PROMIX trial.

Factor	N (%)
Age (mean (SD))	50.00 (9.9)
Menopausal status	
Premenopausal	90 (60.0)
Premenopausal (0-5 years)	17 (11.3)
Premenopausal (more than 5 years)	42 (28.0)
Missing	
Histologic Type	
IDC	108 (72.0)
ILC	22 (14.7)
Other	15 (10.0)
Missing	5 (3.3)
Histologic Grade	
I	4 (2.7)
II	46 (30.7)
III	33 (22.0)
Missing	67 (44.7)
KI67 percentage (mean (SD))	36.0 (24.8)
ER status	
Negative	41 (27.3)
Positive	107 (71.3)
Missing	2 (1.3)
PR status	
Negative	64 (42.7)
Positive	84 (56.0)
Missing	2 (1.3)
Subtype	
LumA	48 (32.0)
LumB	62 (41.3)
TNBC	38 (25.3)
Missing	2 (1.3)
Clinical Stage	
0	4 (2.7)
I	35 (23.3)
II	62 (41.3)
III	39 (26.0)
Missing	10 (6.7)
Response after two cycles of NAC	
Complete response	5 (3.3)
Partial response	64 (42.7)
Stable disease	74 (49.3)
Progressive disease	4 (2.7)
Missing	3 (2.0)
pCR status	
No	129 (86.0)
Yes	20 (13.3)
Missing	1 (0.7)
Recurrence	
No	127 (84.7)
Yes	23 (15.3)
DFS, month	76.80 (33.97)

Abbreviations: SD, standard deviation; IDC, invasive ductal carcinoma; ILC, invasive lobular carcinoma; IHC, immunohistochemistry; NAC, neoadjuvant chemotherapy; ER, estrogen receptor; PR, progesterone receptor; TNBC, triple negative breast cancer; pCR, pathologic complete response; DFS, disease-free survival.

3. The section “Tumor Metabolic Phenotype Interacts with Immune State” needs clarification on the sample groups analyzed. The text alternates between

bioinformatics approaches, which complicates comprehension. For instance, while introducing an integrated approach classifying seven metabolic pathways into three states, the authors fail to identify these pathways or justify their focus.

We understand the metabolic phenotype represent a broad functionality, and now we already validated the phenotype in related pathways as follows. Please refer to our response to reviewer #3 (Q7).

To ensure computational reproducibility of our analyses in Fig3, we now made a R package “PureMeta” (<https://github.com/WangKang-Leo/PureMeta>) involving used integrated bioinformatic pipeline with a tutorial. Accordingly, we changed the code availability part as “The code used to determine tumor cell-based metabolic phenotype (i.e. downregulated, neutral, upregulated) is available at <https://github.com/WangKang-Leo/PureMeta>.”(page 41, line 12 to 13)

4. The hypothesis regarding lipids and immunity is not articulated clearly. The authors should revisit this hypothesis in the conclusion, detailing how the results support it.

We agree that it is crucial to revisit this finding in the conclusion part. We now demonstrate that tumor cell metabolic phenotype (only TCA cycle) interacts

with immune state based on our revised results (Fig.3c and Supplementary Fig.8b), and we deleted some over-speculative statements on lipids. Please refer our response to Reviewer #1 (other comment 2). We also addressed this point into the discussion part: “Among those important metabolic modules, we revealed that tumors with upregulated TCA cycle were more likely to be immunologically cold. When observing closely protein(lipid)-protein(immune) interactions (Fig.4d), immune effector cells (CD4/8+ T cell) from hot tumors harboring higher fat acid signature score compared with cold tumors (Fig.6m), and drivers (ACSL3 and CCNC) in lipid metabolism (Fig.8c) associated with an accelerated growth rate of relevant subclones, we also demonstrated that lipid metabolism plays a crucial role on immune cells”. (page 22, line 11 to 16)

5. A statement regarding the failure to extract TC GEP due to the absence of RNA-seq data raises concerns. The authors assert that certain metabolic-pathway-based subtypes are negatively correlated with immune states. However, without distinguishing lipid pathway subtypes from immune cells, this statement seems speculative. The use of tumor purity as a variable in the LMEM could introduce bias. Moreover, tumor purity assessment via ESTIMATE may not always be accurate in bulk RNA analysis and could lead to misinterpretations if this is used as a variable in a supervised learning analysis. The authors don't know if accounting for purity in the model is causing a boost in the tumor signal, and if that is the case, why do you want to remove the signal coming from the TME? Some tumors have less tumor cell content and higher immune cell content, but this does not always mean lower purity, especially after treatment. A pathological cut-off for high tumor cellularity may be a more accurate method for sample selection than a deconvolution method for tumor purity in this particular case.

We thank the reviewer for proposing this key issue on modeling and tumor purity. We applied LMEM twice: (1) Correlating tumor-cell/bulk derived metabolic phenotype with immune state. (2) Correlating gene expression profiles with treatment timepoint. Given that tumor purity varied across treatment timepoints (from high to low), we extracted tumor cell based GEP to generate metabolic phenotype in LMEM (1) and adjusted for tumor purity in LMEM (2).

A statement “failure to extract TC GEP due to the absence of RNA-seq data” refers to a validation dataset, the Korean cohort, which doesn't include data of non-cancerous breast tissue. Besides, we cannot combine public normal breast GEP with this dataset, because the batch effect and group information are same/overlapped here. We now address this point in the updated Supplementary Fig.7 by adding cellularity term for the validation dataset as following:

$$I_{ij} \sim \beta_0 + \beta_1 M_{ij} + \beta_2 S_{ij} (+ \beta_3 C_{ij}) + \gamma_j + \epsilon_{ij} \quad (4)$$

The tumor cellularity (C_{ij}) was adjusted when we ran LMEM using Korean validation cohort, where tumor-derived GEP was not available.

a

b

c

Phenotype	N of upregulated	N of downregulated	Adjusted OR	P value
Lipid	10	14	1.39	0.08
Amino acid	23	17	0.71	0.02
Carbohydrate	10	8	1.26	0.26
TCA cycle	26	17	0.99	0.97
Energy	4	5	0.78	0.4
Nucleotide	9	6	1.22	0.4
Vitamin	3	4	0.85	0.65

Adjusted OR with 95% CI

Supplementary Fig. 7 Bulk gene expression profiling based metabolic phenotypes interact with immune subtypes in Korean NAC cohort6. a Percentage stacked bar chart showed distribution of bulk gene expression based metabolic phenotype in seven pathways on different immune state, where P values were derived from Chi-Square test or Fisher's exact test. b Funkyheatmap depicting the coefficient, random term, residual variance and p value of LEME, which was conducted within all samples to identify interaction effects between immune state (I) and bulk tumor metabolic phenotypes (M), adjusting for the breast cancer subtype (S) and cellularity

(C). c Forest plot depicting association between bulk metabolic phenotype and pCR. LMEM, linear mixed-effects model; pCR, pathologic complete response.

In this model, as expected, we found that tumor cellularity was negatively correlated with immune state (Seen in source data (Supplemental Fig.7b)).

In the PROMIX cohort, the reason why we have to use ESTIMATE-derived purity is that only RNA-seq derived purity is available for all samples with immune/metabolic phenotype. Importantly, our background data indicated that ESTIMATE-derived purity was highly correlated with cellularity (ground truth, assessed by pathologists) ($r=0.71$) and DNA-based purity (from PureCN) ($r=0.58$) and as following:

6. In the section “Overview of transcriptome response to NAC” the authors employed a linear mixed-effects model (LMEM). Can the authors explain why they used a linear mixed-effect model for doing these analyses instead of other methods like DESeq2? What about the batch effect? Or how do they handle normal cell contamination? What is the random effect? It is not explained in the methodology. Could the authors explain why they used an LMEM instead of other methods if no random effect was used in the model (or at least I missed this info in the methodology). Please clarify.

We apologize for this potential misunderstanding. No potential batch effect within gene expression microarray data from PROMIX trial was identified by Kimbung et al (Int J Cancer. 2018 Feb 1;142(3):618-628; doi: 10.1002/ijc.31070, Supplementary Figure 1) (also see reviewer3 Q15).

The reason why we employed LMEM rather than using DESeq2 or edgeR pipelines is because pathway signature scores is not independent (i.e. two or three biopsies come from the same tumor). Besides, partially overlapping samples across three time points (updated Figure1) result in missing/excluded profiles if we conduct pair-wise analyses using conventional methods. Thereby, LMEM enables us to input all available samples.

We previously ran the LMEM through R package “lme4” as follows:

```
lmem<-lmer(as.numeric(score)~as.factor(time)+as.numeric(purity)+as.factor
```

(subtype)+(1|patientsID), data=mydata), which treats patient as a random effect term (random intercept) and adjusts for “normal cell contamination” using ESTIMATE-derived purity.

To avoid making readers confused, we re-write the equation of LMEM as following:

$$y_{ij} = \beta_0 + \beta_1 P_{ij} + \beta_2 S_{ij} + \beta_3 T_{ij} + \gamma_j + \epsilon_{ij} \quad (2)$$

where samples (i) were contributed by the patient (j), β_0 is the overall intercept, β_1 is the purity effect on PS, β_2 estimates GEP due to IHC subtype, β_3 describes the treatment time effects on GEP, γ_j is the intercept that is allowed to vary across patients (random effect term), and ϵ_{ij} is the residual variations, respectively.

7. Figure 4 d-e: lack of statistics. You make statements in the text for Figure 4d-3 without numbers or statistics. In the same section, the authors claim that “hot tumors associated with increased proliferation (bulk MKI67 protein) were characterized by a massive increase in glucose consumption, also known as the Warburg effect”. Supplemental Fig. 8d. However, this figure did not reach significance. The authors again speculate that might be derived from immune cells. As for Figure 6e, there are no medians or statistics, so this figure is hard to interpret. Please clarify.

We thank the reviewer for pointing out this omission. We have now added box plots and compared mean protein abundance of each module between cold and warm/hot tumors in Fig6e.

Fig. 4e Visualization of average proteome quantification of the three immune subtypes/states (from Fig. 2a) in the correlation network. Boxplot showing difference of mean protein abundance of each module between cold and warm/hot tumors. Statistical significance (P value) was determined using Wilcoxon signed-rank test.

We also add pair-wise statistical comparison in previous Fig.8d (now is Fig.9d).

The paragraph (page 11, line 22 to page 12, line 4 in the revised manuscript) now reads: “Hot/warm tumors were likely to have higher mean protein abundance in amino acid ($P=0.03$, Fig. 4e) and nucleotide ($P=0.07$, Fig. 4e) metabolism compared with cold tumors, probably owing to extra nutrition demands from functional immune cells. Interestingly, though hot tumors were associated with increased proliferation (bulk MKI67 protein) (Supplemental Fig. 9d), MKI67 percentage according to IHC that only counts tumor cells was not different between the three immune states at baseline (Supplemental Fig. 9e). Therefore, we speculated that this upregulated proliferation/nucleotide signaling in hot tumors observed from proteomics might be derived from immune cells”

In Figure 6e, we now marked median value (red) in the stacked violin plots, and logFC derived from differential signature score analysis for each metabolic epithelial cell cluster was added into figure legend as follows:

Fig. 6e The metabolic characteristics of each breast epithelial cell cluster (MC) were analyzed and quantified based on metabolic signature scores: MC1 (glycerolipid metabolism, $\log_2FC=7.4$; cancer antigen presentation, $\log_2FC=2.5$), MC2 (purine biosynthesis, $\log_2FC=2.5$; folate one carbon, $\log_2FC=2.7$; cell cycle, $\log_2FC=3.0$; glycolysis, $\log_2FC=3.2$; hypoxia, $\log_2FC=3.1$; oxidative phosphorylation, $\log_2FC=3.0$;

citric acid cycle, $\log_2FC=2.7$), MC3 (citric acid cycle, $\log_2FC=0.8$; pantothenate and CoA, $\log_2FC=1.6$; pyruvate metabolism, $\log_2FC=1.8$), MC4 (steroid hormone metabolism, $\log_2FC=2.5$; glutathione metabolism, $\log_2FC=0.8$; retinol metabolism, $\log_2FC=2.7$), MC5 (chemokines, $\log_2FC=1.7$; notch signaling, $\log_2FC=0.8$). All $FDR < 0.05$.

8. Supplemental Fig. 14a-b: Please add medians and statistics to this figure.

Here we add medians (dark red) for violin plots, and mark highest group with $\log_2(\text{fold change})$ that was calculated via pseudo-bulking and compared by Wilcoxon-rank test.

9. Supplemental Fig. 15a: Please explain how you calculated the subtypes. And the authors should have this subtype as supplementary data. The PAM50 subtypes are not included or explained in the methodology (single or bulk RNA).

We now added the following text to the methodology part for bulk PAM50: “PAM50 subtype for each sample using Genefu package⁸⁶, taking the official centroids with traditional scaling of the gene expressions as input.”(page 29, line 3 to 5).

The Supplemental Fig. 15a that we moved to Fig.6j-m was generated using well annotated single cell dataset (GSE176078), and we showed marker genes used for cell classifications in figure legend:

Fig.6j UMAP dimensionality reduction diagram showing immune and stromal cell from HER2-negative subset of breast cancer single cell atlas (GSE176078) by sampleID, subtype, and immune state (n of sample=21, n of cell=54,652). Cell type was previously well annotated⁴⁰ as follows: B cells Memory (memory B cells (CD79A, MS4A1, CD27), naive B cells (CD79A, MS4A1, IGHD), plasmablasts (IGKC and IGLC2), CD8+ T cells (CD3, CD8), CD4+ T cells (CD3, CD4), NK cells (KLRC1, KLRB1, NKG7, AREG), cycling T cells (CD3, MKI67), NKT cells (KLRC1, KLRB1, NKG7, FCGR3A), macrophage (CD86), monocyte (CD127), cycling myeloid (KI67), DCs (CLEC9A or CD1C), endothelial (PECAM1, CD34 and VWF), MSC iCAF-like CAFs (ALDH1A1, KLF4 and LEPR), myCAF-like CAFs (ACTA2 (α SMA), TAGLN, FAP and COL1A1), PVL (ACTA 2, PDGFRB and MCAM).

Minor:

10. The supplemental figures contain a significant amount of data that is not discussed in the text. It is advisable for the authors to omit any panels not integral to the manuscript's understanding, as they contribute to confusion.

We agree that this study already included a large number of supplementary figures. However, even more supporting materials have been added in order to properly respond to reviewers' comments.

11. In Figure 2A, why do hot tumors in TNBC present higher levels of HER2?

PROMIX trial enrolled patients who were HER2 non-amplified but had some expression of HER2. Figure 2A showed relative gene expression signature score across samples, and mRNA ERBB2 variation within patients still exists even if they represent clinically HER2-negative breast cancer.

12. In Figure 2c, the author mentions an mIHC panel but is not explained which are the markers used in the text or figure legend.

We thank reviewer for pointing out this omission. We now added mIHC panel details into figure legend as following:

Fig2: c Representative mIHC images (stained for lymphocytic, macrophage and epithelial markers, i.e CD4, CD8, CD20, CD163, CD68, FoxP3 and Cytokeratin), for three immune states, and immune cells density (number of positive cells normalized to tissue area) between immune states. Statistical significance (P value) was determined using Kruskal-Wallis test.

13. Section "Longitudinal Pairwise Analyses on Immunometabolism". Figure 5d. Are these groups balanced? How many samples are in each group?

We now add the table (Supplementary Table6) that summarizes the baseline clinicopathologic characteristics within 69 patients used for longitudinal pairwise analyses on immunometabolism. We demonstrated that patients with positive TME change were more likely to have TNBC than those with negative TME change. Of note, we have already adjusted for tumor stage and subtype when correlating TME change with pCR status in fig5d,e.

Supplementary Table6. Baseline clinicopathologic characteristics of 69 BC patients divided by dynamic TME change.

	Negative TME change	Positive TME change	P value
N	24	45	
Age (mean (SD))	51.53 (8.95)	47.18 (10.48)	0.09
Histologic grade (%)			
I	0 (0.0)	1 (2.2)	0.1
II	10 (41.7)	9 (20.0)	
III	3 (12.5)	16 (35.6)	
Missing	11 (45.8)	19 (42.2)	
Subtype (%)			
LumA	12 (50.0)	8 (17.8)	0.004
LumB	9 (37.5)	18 (40.0)	
TNBC	2 (8.3)	19 (42.2)	
Missing	1 (4.2)	0 (0.0)	
Clinical Stage (%)			
0	1 (4.2)	1 (2.2)	0.07
I	3 (12.5)	18 (40.0)	
II	15 (62.5)	14 (31.1)	
III	5 (20.8)	10 (22.2)	
Missing	0 (0.0)	2 (4.4)	
pCR (%)			
No	23 (95.8)	35 (77.8)	0.1
Yes	1 (4.2)	10 (22.2)	
Radiologic response after 2 cycles (%)			
No	19	5	0.03
Yes	22	23	

Theodoros Foukakis, MD, PhD
 Associate Professor of Oncology, Senior Consultant
 Breast Center, Theme Cancer Karolinska University Hospital Solna, Karolinska
 Comprehensive Cancer Center &
 Department of Oncology-Pathology, Karolinska Institutet, Stockholm, Sweden
 e-mail: theodoros.foukakis@ki.se

REVIEWERS' COMMENTS

Reviewer #1 (Remarks to the Author):

The manuscript presents a comprehensive longitudinal studies to report on the metabolic changes and metabolic reprogramming within tumor cells and tumor microenvironment (TME) in breast cancer. This is the revision of my earlier review comments. I am grateful that authors have taken my comments seriously and taken every possible experimental steps to address my comments. The manuscript has also been significantly revised/re-written for clarity and remove ambiguous/speculative texts. Authors have addressed all my concerns satisfactorily. I am happy to recommend for its acceptance.

Reviewer #2 (Remarks to the Author):

The authors dramatically improved the manuscript and referred to all comments and suggestions. No further comments.

Reviewer #3 (Remarks to the Author):

The authors have almost addressed my previous concerns. Based on the newly updated manuscript, I have a few minor suggestions:

1. The authors are only analyzing HER2- breast cancer, title could be more specific.
2. I'm not convinced that the immune subtypes the authors identified from GEP is well aligned with the single cell data. In Q17, the author stated "we could demonstrate a trend towards to higher T cell proportion in hot tumors, shown in the figure below." However, this is all driven by one outlier sample. The other samples didn't show any difference.

Reviewer #4 (Remarks to the Author):

I have now had the opportunity to review the revised manuscript titled "Longitudinal molecular profiling elucidates immunometabolism dynamics in breast cancer" by Kang Wang et al. The authors have thoroughly and satisfactorily addressed the comments and suggestions I raised in my initial review.

The revisions made to the manuscript have significantly improved its clarity, depth, and overall quality. Specifically, I appreciate the effort put into clarifying methodology and adding more data analysis and clinical data. These changes have enhanced the manuscript and addressed the previously raised concerns.

The manuscript has been strengthened and is now more robust in light of these revisions. The authors have demonstrated commitment to enhancing their work.

Based on these considerations, I recommend the manuscript be accepted for publication.

Thank you for the opportunity to review this manuscript again.

RESPONSE TO REVIEWERS' COMMENTS

Reviewer #3 (Remarks to the Author):

The authors have almost addressed my previous concerns. Based on the newly updated manuscript, I have a few minor suggestions:

1. The authors are only analyzing HER2- breast cancer, title could be more specific. **We agree that only HER2-negative breast cancer was included in this study, however, HER2-negative breast cancer is still a heterogenous subgroup that represents about 80% of the diagnosed cases. We thus suggest that the title remains “Longitudinal molecular profiling elucidates immunometabolism dynamics in breast cancer”.**

2. I'm not convinced that the immune subtypes the authors identified from GEP is well aligned with the single cell data. In Q17, the author stated “we could demonstrate a trend towards to higher T cell proportion in hot tumors, shown in the figure below.” However, this is all driven by one outlier sample. The other samples didn't show any difference.

We employed the Wilcoxon rank test, a statistical model known for its resilience to outliers and extreme values, ensuring the reliability of our analysis.

Theodoros Foukakis, MD, PhD
Associate Professor of Oncology, Senior Consultant
Breast Center, Theme Cancer Karolinska University Hospital Solna, Karolinska
Comprehensive Cancer Center &
Department of Oncology-Pathology, Karolinska Institutet, Stockholm, Sweden
e-mail: theodoros.foukakis@ki.se